

SciPost Phys. Lect.Notes 18 (2020)

# Lecture notes on Generalised Hydrodynamics

**Benjamin Doyon**

Department of Mathematics, King's College London, Strand, London WC2R 2LS, U.K.

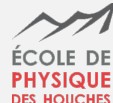

*Part of the Integrability in Atomic and Condensed Matter Physics
Session 111 of the Les Houches School, August 2018
published in the Les Houches Lecture Notes Series*

## Abstract

These are lecture notes for a series of lectures given at the Les Houches Summer School on Integrability in Atomic and Condensed Matter Physics, 30 July to 24 August 2018. The same series of lectures has also been given at the Tokyo Institute of Technology, October 2019. I overview in a pedagogical fashion the main aspects of the theory of generalised hydrodynamics, a hydrodynamic theory for quantum and classical many-body integrable systems. Only very basic knowledge of hydrodynamics and integrable systems is assumed.

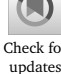

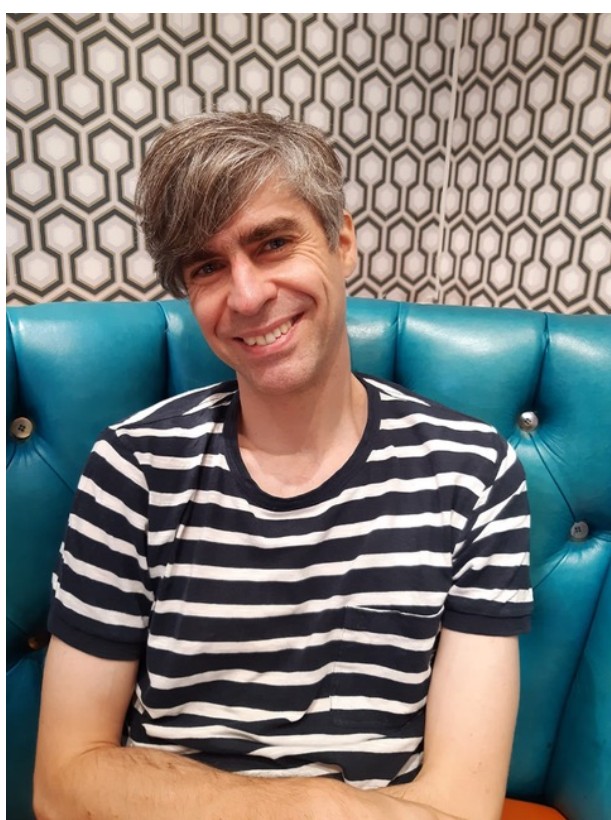

## Chapter 1   Introduction

These are lecture notes for the Les Houches summer school on Integrability in Atomic and Condensed Matter Physics, 30 July to 24 August 2018. Their purpose is to provide an introduction to the theory of generalised hydrodynamics (GHD). This theory has been developed primarily in the past three years or so, sparked by two papers that appeared simultaneously in preprint in May 2016 [1, 2], where it is proposed using alternative, complementary arguments. It is a theory based both on the empirically well-established fundamental principles of hydrodynamics, and on some basic structures associated to integrability, abstracted from the Bethe ansatz. It provides a framework for studying a wide family of integrable systems far from equilibrium. Integrability is here understood in a large sense, and includes classical and quantum gases, chains, field theory models, and even (conjecturally) certain cellular automata; no specific algebraic of geometric constructions of advanced integrability theory appears to be needed for the formulation of generalised hydrodynamics. I expect generalised hydrodynamics to be even applicable, at low densities, to non-integrable one-dimensional many-body systems with short range interactions whose excitations can be described by quasiparticles.

Non-equilibrium phenomena, often defined as those where time-reversal invariance is broken or where entropy is produced, constitute one of the most active and high profile areas of studies in modern science [3,4]. It is a vast subject, as unlike the characterisation of equilibrium states, it is hard to imagine a full account of all situations that are far from equilibrium, which must include life itself! A foremost question is that of establishing a general framework and uncovering a set of principles that may be considered as underpinning at least some wide family of non-equilibrium states and their dynamics. In theoretical physics, important advances have been made in these directions. I believe two related frameworks have emerged as being of particular relevance: hydrodynamics, and large deviation theory.

Hydrodynamics, the main topic of these notes, is of course extremely old, and the subject of many textbooks. It is a theory for inhomogeneous, dynamic many-body states, where there exist nonzero currents or flows – of particles, charge, energy, spin, etc. – between various parts of the system; see for instance [5–7]. It describes what happens when variations occur over large distances and long times, answering at some level, in this regime, the questions of how non-equilibrium flows evolve and fluctuate, and what guide their shape and strength. Modern developments have shown how its basic principles can be extended or deepened to account for many of the sometimes exotic systems and phenomena that are of current interest. Excluding developments directly related to the subject of these notes – reviewed below – one may cite advances in high-energy physics and field theory [8–12] and in strongly correlated systems [13,14], as well as the large amount of progress done in the context of electron flows in graphene-based systems [15–18], which has had experimental evidence [19–21].

Large deviation theory, on the other hand, was developed in the 1960's and 1970's as an overarching theory for equilibrium thermodynamics, and has been suggested to provide the right principles for a generalisation to non-equilibrium physics, see e.g. [22]. It is a theory that can be formulated in very general terms, describing the fluctuations of "macroscopic" quantities. It has been especially successful in non-equilibrium physics. For instance, it gave rise to universal fluctuation relations [23–29] (see the review [30]), which encode, in system-independent symmetry relations, properties of driving potentials into fluctuations of quantities transported over long times. It is also at the basis of macroscopic fluctuation theory [31–34], which establishes a deep connection between the hydrodynamics of diffusive transport, and fluctuations of non-equilibrium currents.

In recent years, there has been a large amount of interest in understanding the non-equilibrium physics of *integrable* systems, where a macroscopic number of conservation laws exist. One goal was to use the mathematical structure of integrability in an attempt to obtain exact, model-dependent results that might help uncover new general principles of non-equilibrium physics, and verify established ones. However, going beyond this, it quickly became clear that integrability offers much more. First, integrable models are actually realised in experiments. For instance, it was discovered [35–37] that the Lieb-Liniger model, solved by Lieb and Liniger using the Bethe ansatz in 1963 [38], describes cold atomic gases constrained to (quasi-)one-dimensional tubes. Such experiments have attracted a lot of attention in the past 15 years, for their high manipulability and the access they give us to the physics of many-body quantum dynamics, see the book [39]. Second, integrability affects non-equilibrium physics in fundamental ways, not seen in equilibrium states. This was observed for instance in the seminal cold-atomic experiment on the so-called quantum Newton's cradle [40], where clouds of Rubidium atoms, constrained to tubes, collide without thermalising, in contrast to what is expected of ordinary, non-integrable gases. That is, in one dimension, despite the fact that real systems are not integrable, the large number of conserved quantities of integrable models still appears to play a fundamental role in the non-equilibrium dynamics.

The lack of thermalisation in integrable models was theoretically addressed by studying simpler setups, referred to as *quantum quenches*, where integrable quantum systems are made

(on paper or in computers) to undergo homogeneous evolution from non-stationary states far from the ground state. The fundamental concept of generalised Gibbs ensembles emerged [41–44], and was seen experimentally [45] and developed in a large amount of works in the integrability community, see the reviews [4,46]. However, despite these advances in the non-equilibrium physics of integrability, there was no simple and efficient framework to describe their inhomogeneous dynamics such as in the quantum Newton's cradle experiment, nor to understand their large deviations. The extensive amount of ballistic transport afforded by the conservation laws of integrable models rendered conventional theories inapplicable. It is in this respect that generalised hydrodynamics offers new directions.

Generalised hydrodynamics [1, 2, 47–49] is an extension of hydrodynamics to integrable systems, constructed on generalised Gibbs ensembles instead of Gibbs ensembles. It was originally used in order to solve one of the most iconic problems of non-equilibrium physics, that of evaluating exact currents in states that are steady – that do not change with time – yet far from equilibrium. Non-equilibrium steady states have been studied for a long time. Perhaps the earliest appearance in theoretical physics is the Riemann problem of hydrodynamics [50]. In more general contexts it is sometimes referred to as the "partitioning protocol" [51–53], see the reviews [54, 55], as well as [56] for a more general discussion of transport in quantum models. In this protocol, a physical system is partitioned into two halves, which are, initially, independently thermalised into different equilibrium states. The halves are then connected, and let to evolve according to the Hamiltonian evolution of the physical system under consideration. After a long time, if ballistic transport is allowed by the dynamics, a current develops as a consequence of the initial imbalance, where quantities are transported, without diffusing, from one side to the other. This protocol, based on Hamiltonian dynamics, has the advantage of being directly amenable to study in a wide variety of systems, including classical and quantum gases, lattice models and field theories. A large number of exact predictions and even rigorous results have been obtained: for the classical harmonic lattice [51], quantum models with free fermionic and bosonic descriptions [57–65], and conformal field theories in one dimension [53,66] and in higher dimensions [10,67,68]. In *interacting, integrable, non-conformal* quantum models, the full solution was provided in [1] for field theories, with the examples of the sinh-Gordon and Lieb-Liniger models, and in [2] for quantum chains, with the example of the XXZ anisotropic Heisenberg chain. These papers introduced generalised hydrodynamics, and solved its Riemann problem.

From the original papers, the architecture of generalised hydrodynamics was relatively clear, and has been developed in later works. The theory appears to be extremely flexible, being applicable to *classical and quantum* models of various types, such as chains, gases and field theories. From the perspective of integrable systems, it provides, in its current form, an extension of the techniques of the thermodynamic Bethe ansatz [69–71], uncovering new structures within it, and unifies a wide range of models, including in particular soliton gases [72–76] and the hard rod gas [7,77,78], where similar hydrodynamic theories had been partially developed at a high level of mathematical rigour. From the perspective of cold atom physics, it provides the first complete and efficient framework for one-dimensional cold atomic gases at large wavelengths, able to reproduce the main effects seen in the quantum Newton's cradle experiment [79], and verified experimentally [80]. From the perspective of non-equilibrium physics, it is a powerful framework for inhomogeneous dynamics of quantum and classical systems with extensive ballistic transport. I also hope that it will form the basis for their large deviation theory, via a counterpart to macroscopic fluctuation theory.

These lecture notes are divided into three chapters. In Chapter 2, the rudiments of hydrodynamics are explained. No advanced prior knowledge of hydrodynamics is assumed. In this chapter, the standard, fundamental precepts of hydrodynamics are put into a perspective that makes them easily applicable, at least in principle, to integrable models. A variety of aspects of

hydrodynamics are reviewed, including the Riemann problem, normal modes, Onsager-type relations and linear fluctuating hydrodynamics. In Chapter 3, the machinery of the thermodynamic Bethe ansatz is explained. This is of course based on the Bethe ansatz, which the students will have seen, at least in part, in the first few lectures of other courses in this school. However, I provide an intuitive, physical point of view which does not rely on it, sufficient for my purposes. For clarity, I base it on the scattering picture of classical particle systems. Little knowledge of the mathematics of integrability is required in order to understand the equations themselves – although it helps in order to understand more technically where they come from. In Chapter 4, the concepts introduced in the previous two chapters are combined into generalised hydrodynamics. I explain the basic equations of the hydrodynamic theory, the solution to the Riemann problem as well as a variety of other topics.

I note that the division of topics taken here is not that corresponding to the historical development of the subject. Neither do I abide, for some of the topics, by the usual distinction between thermodynamics and hydrodynamics. The division is instead based on a more conceptual logic. For instance, the Drude weight, flux Jacobian and velocities of the fluid normal modes are conventionally seen as part of Euler hydrodynamics, but their exposition is here provided within thermodynamics. This is more accurate, as, although these objects are naturally involved in the hydrodynamic description, no hydrodynamic assumption is actually needed for their basic definition and in order to derive their main properties: only thermodynamic averages are required.

Many important topics stemming from, or connected to, generalised hydrodynamics have been omitted. I note in particular, the following (incomplete list of) extra topics, some rather well developed, other necessitating further investigation: the study of the zero-entropy subspace of the thermodynamic states in quantum integrable models with Fermi seas [81,82], the integrability structure of the GHD equations themselves [75,83,84], thermalisation and the breaking of the GHD equations [85–93], the spectrum of fluctuations in transport phenomena and correlation functions of order parameter [94,95], the phenomenon of superdiffusion in quantum spin chains [96–99], properties of quantum entanglement in inhomogeneous, non-stationary situations [82,100–104], the hydrodynamics of integrable relativistic quantum field theory [1,105], and the formal derivation of the elements of GHD [106–116].

Before plunging into the details of hydrodynamics, the thermodynamic Bethe ansatz technology, and generalised hydrodynamics, it may be useful to discuss the nature of the GHD description of one of the most important experiments in low-dimensional many-body systems, that of the quantum Newton's cradle [40] discussed above. This description will already expose much of the intuition behind GHD. As an invitation, I present it in the following section; the later chapters do not depend on it.

## 1.1 Invitation: the hydrodynamics of a quantum Newton's cradle experiment

A full theoretical understanding of an experiment is always a very difficult endeavour, and GHD alone cannot do this. However, it is able to provide a description, valid for all interaction strengths, of large scale phenomena where the emergent physics arise. It is in this sense that GHD solves the quantum Newton's cradle experiment: it arguably gives the theoretical underpinning for the most striking features of its non-equilibrium physics. Clearly, there has been many theoretical studies of the quantum Newton's cradle experiment, and a variety of methods can be applied under various approximations and in various parameter regimes. Discussing these methods and their relations to GHD, or the particular physics of cold atomic gases, would bring me too far from the main goal of these notes. I refer the reader, for instance, to the book [39] for the quasicondensate regime and related questions, and to the discussions and references in the papers [79,80,117] where the GHD of cold atomic gases is discussed. I believe it is fair to say that more research is necessary in order to fully clarify the

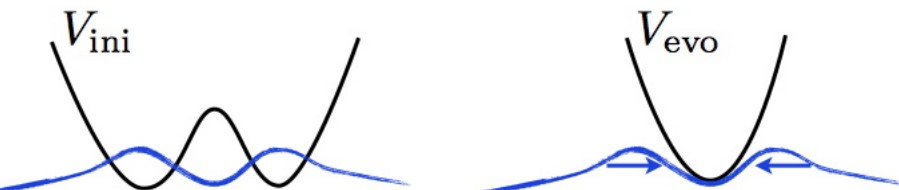

Figure 1: A sudden change of the shape of the longitudinal potential induces a large-scale motion in the gas.

relation between standard methods and GHD in cold-atomic setups.

In the quantum Newton's cradle experiment, a gas of rubidium atoms, confined to lie within a one-dimensional "tube", is first brought to low enough temperatures so that quantumness becomes relevant. The details of how such particles are confined to one dimension – be it using lasers or magnetic fields from nearby wires – is not relevant for the present purpose. It suffices to say that the transverse confining leads to the presence of states (representing transverse motion) whose energies are sufficiently higher than the temperature, and thus can be neglected. The gas – which, say, contains about 2000 rubidium atoms – is also confined to lie within a small region of one-dimensional space with a longitudinal external potential (again, with laser or magnetic fields). The longitudinal potential allows atoms to move along the tube, in such a way that the system behaves as a one-dimensional gas. Of course, in order for the system to be nontrivial, the gas density should be of the order of the inverse typical one-dimensional scattering length, something which depends on the transverse trapping potential. In order for a hydrodynamic theory to be applicable, the longitudinal potential should also vary only on large enough length scales.

Then, the gas is made to move in some way. The most intuitive way is simply to imagine that the initial shape of the one-dimensional confining potential is suddenly changed. For instance, the shape may initially be a double well, where the gas has relaxed and is concentrated on two separate regions of space, and then it may be suddenly changed (but this could also be smooth, instead of sudden) to a single well, forcing the two blobs of gas to start moving. See Fig. 1. The question to answer is: how can we describe the motion of the gas after the sudden change of longitudinal potential?

A microscopic, theoretical formulation of the problem, which is a very good approximation of real systems, is in fact well known. The atoms are described by the Lieb-Liniger model, a model of Galilean particles of Bose statistics with point-like interactions. Its homogeneous Hamiltonian is (setting the mass of the particles to 1)

$$H = -\sum_{n=1}^{N} \frac{1}{2} \partial_{x_n}^2 + g \sum_{n<m} \delta(x_n - x_m),$$  (1)

where $g$ is the interaction strength (which, as said, depends on the transverse trap), and $N$ is the number of particles in the gas. The external longitudinal potential is simply modelled as interacting with the local particle density $\mathfrak{n}(x) = \sum_m \delta(x - x_m)$. We may assume that the initial state is an equilibrium, thermal state within the inhomogeneous potential, at some temperature $\beta^{-1}$, so that averages of any local observables are given by

$$\langle o(x) \rangle = \frac{\text{Tr}(\rho_{\text{ini}} o(x))}{\text{Tr}\rho_{\text{ini}}}, \quad \rho_{\text{ini}} = \exp\left[-\beta\left(H + \int dy\, V_{\text{ini}}(y)\mathfrak{n}(y)\right)\right].$$  (2)

Then the evolution occurs with the Hamiltonian in a different inhomogeneous potential,

$$\langle o(x,t) \rangle = \langle e^{iH_{\text{evo}}t} o(x) e^{-iH_{\text{evo}}t} \rangle, \quad H_{\text{evo}} = H + \int dy\, V_{\text{evo}}(y)\mathfrak{n}(y).$$  (3)

For instance, a quantity that is directly observable experimentally is the atom density, $\langle \mathfrak{n}(x,t) \rangle$.

Technically, this is a well-posed problem, and in principle, it is simply a matter of solving it by the standard methods of quantum mechanics. However, doing this by "brute force" on the computer would be rather limiting, as perhaps a maximum of 30 particles could be reached with current technology; we couldn't have confidence that we are extracting the correct large-scale physics that is expected to arise at 2000 particles. Instead, we may note that the model (1) is in fact integrable: this means in particular that there are a number of advanced techniques available in order to solve it. For instance, one can obtain its eigenvectors and eigenvalues by the Bethe ansatz. But this is not that helpful, as the operators appearing in the exponentials in (2) and (3) are *not* integrable, because of the presence of the inhomogeneous potentials. It is, after all, these operators that we actually have to diagonalise. We may use the Bethe ansatz basis of the homogeneous Hamiltonian, and this allows us to reach perhaps 50 particles or a bit more. But this is still relatively far from 2000, and requires extensive computer resources.

Instead of thinking of how to solve the microscopic model, we may think of the physics we expect. The first intuition is that, if two blobs of a gas – or two regions of a gas where a significantly higher density is present – are made to collide against each other, then one would expect, because of the interactions, the particles to exchange momenta in complicated ways, and the momenta of particles to re-distribute so as to re-thermalise. We would expect to be able to use the Navier-Stokes equation (specialised to one dimension), or even its specialisation without viscosity, the Euler equation – after all, these equations automatically take into account the redistribution of microscopic momenta, and should describe the motion of fluids at large scales (see Section 2.1). In particular, anyone who is well acquainted with standard hydrodynamics would expect shocks to form at the collision, at which entropy increases. In any case, shortly, the independent blobs would stop their forward displacements, the gas would display a complicated large-scale motion where the original blobs would not be discernible anymore, and eventually it would reach some relatively steady configuration with respect to the evolution potential $V_{\mathrm{evo}}$.

The problem, of course, is that this is not what is seen in experiments. Instead, the blobs collide, form a new blob momentarily in the centre, and then remerge from it, slightly modified, to continue their motion up, and then down, the slopes of the evolution potential. The process repeats for a great many cycles, until other effects, such as atom losses or "prethermalisation", become relevant. This means that, not only the standard techniques are unable to solve the problem from its microscopic formulation, but the physics of standard hydrodynamics, which is the one we thought should describe the large-scale motion, is simply incorrect. It is to be noted that if the transverse potential is not as tight, and states representing transverse motion are within the range of energies available, then the intuitive picture of standard hydrodynamics is seen to be correct. Something happens in one dimension, because of integrability.

The authors of the original quantum Newton's cradle experiment correctly identified the phenomenon: that of a Newton's cradle. Recall that this is the toy in which a number of metallic beads are held by strings, and disposed horizontally so as to lightly touch each other when at equilibrium. One of the outer bead is held up, and then let to descend upon its neighbour. As it hits it, it exchanges its momentum with it, itself exchanging its momentum with the next neighbour, and so on. No bead seems to move, except for the other outer bead, which, imparted with the momentum, starts moving up. The beads are to be likened with the atoms of rubidium in the gas. The statement is that, because of the integrability of the model, although the atoms do indeed exchange their momenta in complicated ways, the motion of these individual momenta is relatively simple, and occurs on large scales. The blobs we see emerging after the first collision are not formed of the original atoms, but carry the original momenta.

Although this is an over-simplified picture, it extracts the correct physics. But, in order to

go from such a picture to actual equations, putting together the inhomogeneous potentials – which break integrability – and the simplification of momentum exchanges due to integrability, one needs to do a bit of work. The presentation I propose in these notes is in fact based on applying fundamental principles of hydrodynamics, and, satisfyingly, the Newton's cradle picture of large-scale momentum motion comes out a posteriori.

Let me simply provide here the generalised hydrodynamic solution to the above problem. It appears as a set of integro-differential equations. The main object, in the most physically transparent formulation, is the density $\rho_p(p, x, t)$ of "quasi-particles" that carry momentum $p$ at space-time point $x, t$. The quantity $\rho_p(p, x, t)$ is a density per unit momentum and per unit space. Like in the Newton's cradle, the momenta are smoothly carried around, but instead of a single momentum, we have a continuum. These are not the real particles – which move around and exchange their momenta in complicated ways – but intermediate objects following the flows of momenta. The connection with real particles is simple however, for instance the density of particles at space-time point $x, t$ is

$$\langle \mathfrak{n}(x, t) \rangle = \int dp \, \rho_p(p, x, t). \tag{4}$$

The initial state representing (2) is fixed by solving a set of integral equations for two functions: the sought-after initial quasi-particle density $\rho_p(p, x, 0)$, and an auxiliary function $\epsilon(p, x)$ called the pseudoenergy. The equations are as follows:

$$2\pi(1 + e^{\epsilon(p,x)})\rho_p(p, x, 0) = 1 + \int dq \, \varphi(p - q)\rho_p(q, x, 0)$$

$$\epsilon(p, x) = \beta\left(\frac{p^2}{2} + V_{\text{ini}}(x)\right) - \int \frac{dq}{2\pi} \, \varphi(p - q)\log(1 + e^{-\epsilon(q,x)}). \tag{5}$$

In the latter equation, the function $\varphi(p - q)$ appears: this encodes the scattering of particles with momenta $p$ and $q$. In the Lieb-Liniger model, it takes the form

$$\varphi(p - q) = \frac{2g}{(p - q)^2 + g^2}. \tag{6}$$

Eq. (6) is in fact simple to obtain from a two-body scattering problem; while Eqs. (5) follow from a "local density approximation" and the thermodynamic Bethe ansatz.

Once Eqs. (5) are solved, say numerically by recursion (a process that is usually very efficient), the function $\rho_p(p, x, t)$ is obtained for other times $t$ by solving the integro-differential equation

$$\partial_t \rho_p + \partial_x\left[v^{\text{eff}}\rho_p\right] - \partial_x V_{\text{evo}}(x) \, \partial_p \rho_p = 0. \tag{7}$$

Here all functions whose arguments we do not write are to be evaluated at $p, x, t$. In this equation, a new function $v^{\text{eff}}(p, x, t)$ appears. This is obtained by solving yet another integral equation:

$$v^{\text{eff}}(p, x, t) = p + \int dq \, \varphi(p - q)\rho_p(q, x, t)(v^{\text{eff}}(q, x, t) - v^{\text{eff}}(p, x, t)). \tag{8}$$

Eqs. (7) with (8) are the hydrodynamic representation of the microscopic evolution (3). Eq. (7) has a clear physical interpretation: it is a dynamical equation for a density, with quasi-particles moving at velocities $v^{\text{eff}}(p, x, t)$ and subjected to an acceleration $-\partial_x V_{\text{evo}}(x)$. The nontrivial aspect of this equation is the so-called effective velocity $v^{\text{eff}}(p, x, t)$, which encodes all interaction effects.

In these notes, the meaning of Eqs. (7), (8) as Euler-scale hydrodynamic equations will be explained, and Eqs. (4)-(8) will be put within a wide framework able to access many physically relevant quantities in a large variety of integrable many-body models.

**Acknowledgements.** The development of generalised hydrodynamics in the past few years has been extremely fast, with contributions by many researchers in the fields of integrable systems, condensed matter theory and statistical mechanics. Likewise, my understanding as overviewed in these notes has benefited immensely from discussions with collaborators and colleagues, including: Alvise Bastianello, Denis Bernard, Bruno Bertini, M. Joe Bhaseen, Olalla Castro-Alvaredo, Jean-Sébastien Caux, Jacopo De Nardis, Jérôme Dubail, Joseph Durnin, Krzysztof Gawedski, Jason Myers, Herbert Spohn, Romain Vasseur, Jacopo Viti, Takato Yoshimura, and many others. Particular thanks go to Gabriele Perfetto for finding many typos in the first version, to Alvise Bastianello and Jacopo Viti for identifying missed references, to Herbert Spohn for pointing out conceptual imprecisions, and to Alessandro De Martino and Andrew Lucas for suggesting references on electron fluids. A large part of these notes was written while at the Les Houches Summer School on Integrability in Atomic and Condensed Matter Physics. I also thank University of Roma Tre, as well as École Normale Supérieure de Lyon and the Tokyo Institute of Technology for support and hospitality during invited professorships, where lectures on the topic were presented and parts of the notes were written. This work was supported by a Royal Society Leverhulme Trust Senior Research Fellowship, "Emergent hydrodynamics in integrable systems: non-equilibrium theory", ref. SRF\R1\180103, and an EPSRC standard grant, "Entanglement Measures, Twist Fields, and Partition Functions in Quantum Field Theory", ref. EP/P006132/1.

# Chapter 2    Rudiments of hydrodynamics

Hydrodynamics is an extremely old theory. It has been very widely studied, with great success. Although there does not exist a mathematically rigorous proof of its validity in realistic, interacting gases, the principles of hydrodynamics can be thought of as being rather well established. Yet, surprisingly, it is still the subject of many modern studies in a wide variety of fields, including high-energy, statistical and condensed matter physics. This is for a good reason: it is an extremely powerful emergent theory for describing complex many-body systems. It states that the complex behaviours of the many, many particles with small-range interactions in a gas can be simplified drastically to a much smaller set of equations than those for the individual trajectories of each particle. These equations are particularly well suited for studying non-equilibrium, dynamical, inhomogeneous phenomena of many-body systems.

In a pictorial representation of time scales, see Fig. 2, hydrodynamics lies somewhere below thermodynamics, and above the Boltzmann equations. The thinking goes as follows. At very short time scales, the individual particles of the gas propagate ballistically, reversibly, between collisions. This is the microscopic regime, which can always be used to describe the gas. Then after many collisions, some mixing occurs. At this stage, one reverts to an approximate description where, instead of individual particles' trajectories, one uses the coarser density of particles in the single-particle phase space. This description is valid only after this coarse graining has actually occurred. This leads to the Boltzmann equation[1], with contains a collision integral that accounts for the change of phase space densities due to collisions. This is famously an irreversible dynamics, the passage from reversible to irreversible being attributed to the coarsening and arguments about microscopic phase space volumes occupied by coarse states. Collisions – or the collision integral in the Boltzmann equation – lead to relaxation,

---

[1]Intermediate steps leading to the Boltzmann equation are given by the so-called BBGKY hierarchy.

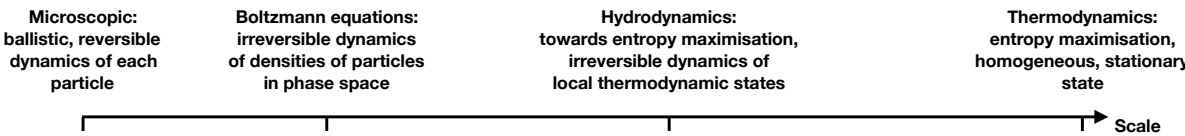

Figure 2: Time scales in gases with their various theoretical descriptions.

whereby the system tends to maximise entropy. Entropy maximisation occurs at large scales compared to the microscopic scales, but can still occur at small scales compared to laboratory scales. Thus we divide the system into "fluid cells", where each cell, small on laboratory scales, is considered thermodynamically large, and is considered to have (nearly) reached a state in which entropy has been maximised. These are local thermodynamic states, and local entropy maximisation is often referred to as the reaching of a local thermodynamic equilibrium (although in integrable systems this nomenclature is not entirely accurate). There are usually much fewer available entropy-maximised thermodynamic states than there are possible distribution of particles in momentum space. In a conventional one-dimensional Galilean gas, for instance, an entropy-maximised thermodynamic state is described by a temperature, a chemical potential and a Galilean boost: three numbers instead of an infinity. This change in the degrees of freedom used to describe the local states – from densities in single-particle phase space to the degrees of freedom of entropy-maximised thermodynamic states – is one of the most important assumptions of hydrodynamics. Hydrodynamics is a derivative expansion, and as such there are various scales within it, obtained by various scaling limits. Unless one takes the Euler scaling limit, which is the lowest order in derivatives, hydrodynamics is generically irreversible: Navier-Stokes type of terms, which are at the second order in derivatives, lead to diffusion. By diffusion or other mechanisms, the weak state modulations in space eventually disappear, and at the largest time scales, one recovers thermodynamics.

Of course, this picture is clear for a classical gas, but similar principles are expected to hold for classical and quantum gases, lattice models, and field theories. In constructing generalised hydrodynamics of integrable systems, we simply follow these basic principles. I will now make them more precise, in a general one-dimensional setting that includes conventional and integrable systems.

The most fundamental objects in hydrodynamics are the conservation laws afforded by the model under consideration. These are at the basis of the manifold of entropy-maximised thermodynamic states, and as a consequence give the dynamical degrees of freedom governed by hydrodynamics.

In this Chapter, I will describe the main aspects of thermodynamics and hydrodynamics that we need in order to fully understand the theory of generalised hydrodynamics. I start with a review of very basic hydrodynamics from textbooks: the Euler and Navier-Stokes equations.

## 2.1 Euler and Navier-Stokes equations

Before extracting the general principles, it is useful to remind ourselves of the usual equations of hydrodynamics for Galilean fluids (in one dimension of space). They take the following form:

$$\partial_t \rho(x,t) + \partial_x [v(x,t)\rho(x,t)] = 0$$
$$\partial_t v(x,t) + v(x,t)\partial_x v(x,t) = \frac{1}{\rho(x,t)} \big[ -\partial_x \mathrm{P}(\rho(x,t)) + \zeta \partial_x^2 v(x,t) \big]. \tag{9}$$

In general there is also an equation for the local entropy, but in isentropic fluids – a good approximation in many situations – the above are enough. The dynamical variables are $\rho(x,t)$, the local mass density (for simplicity we assume a unit mass $m=1$), and $v(x,t)$, the local

velocity field. In the second equation, $P(\rho)$ is the pressure, purely a function of the mass density, and $\zeta$, the bulk viscosity (in one dimension, there is no shear viscosity). The pressure and bulk viscosity are the only model-dependent quantities, which must be determined from microscopic calculations. Once they are known, Eqs. (9) describe the time evolution of the fluid. That is, these equations stipulate that there is a large reduction of the number of degrees of freedom, from the $\approx 10^{23}$ trajectories of the individual particles, to two space-time functions, $\rho(x, t)$ and $v(x, t)$. This large reduction of the number of degrees of freedom is at the basis of hydrodynamics.

Let us look at Eqs. (9) in more details. The first equation is clearly that of mass conservation. The second equation is the one-dimensional Navier-Stokes equation, and with the second-derivative term omitted, it is the Euler equation. It says that the convective derivative $\partial_t + v(x, t)\partial_x$ of the velocity fields is controlled by the pressure and viscosity. These have natural interpretations: the pressure variations give a "thermodynamic force" modifying the velocity, and the viscosity acts as a friction force. In fact, the second equation in (9) can also be recast into a conservation form, that of momentum. Defining the momentum field $p = v\rho$ and its current $j_p = P + v^2\rho - \zeta\partial_x v$, the equations now take the form

$$
\begin{aligned}
\partial_t \rho(x, t) + \partial_x p(x, t) &= 0 \\
\partial_t p(x, t) + \partial_x j_p(x, t) &= 0.
\end{aligned}
\tag{10}
$$

The reduction of the relevant dynamical degrees of freedom to those governed by conservation laws is the fundamental principle of hydrodynamics. Armed with this principle, let us look at the general situation.

## 2.2 Maximal entropy states and thermodynamics

Consider a homogeneous many-body one-dimensional system of infinite length, with short-range interactions. We assume it is isolated from any external environment. Suppose the system admits a conserved total energy, which we will denote $H$ (in models with Hamiltonian dynamics, this is the Hamiltonian), and a set of conserved charges $Q_i$ (for $i$ in some index set), which can be written as integrals of densities satisfying conservation laws[2]:

$$
Q_i = \int \mathrm{d}x\, q_i(x, t), \qquad \partial_t q_i(x, t) + \partial_x j_i(x, t) = 0, \qquad \partial_t Q_i = 0.
\tag{11}
$$

In particular, the Hamiltonian $H$ is one of these. Here, densities are either local – supported on finite regions – or quasi-local – supported on infinite regions but with an "envelope" that decays sufficiently fast (see the review [118]).

The first question that we ask is: if the system starts in some arbitrary, generic, homogeneous state, what happens to a typical finite region after long enough times? Physically, we expect such finite regions to "relax" to some state, the rest of the infinite system playing the role of a bath. By ergodicity, the density matrix, or state distribution, $\rho$ that describes[3] all local or quasi-local observables $o$,

$$
\langle o \rangle = \mathrm{Tr}[\rho o],
\tag{12}
$$

will *maximise entropy*, $S = -\mathrm{Tr}[\rho \log \rho]$. As averages of conserved densities cannot change, entropy maximisation is with respect to the available conservation laws. Constraints for the

---

[2]For simplicity, in quantum systems, we will assume that the charges commute with each other. Here and below I use a continuous-space notation for convenience; similar equations hold for chains.

[3]Here and below I use the trace notation Tr proper to quantum mechanics for convenience; in general, this represents some appropriate a priori measure on phase space.

conserved quantities $Q_i$ and for the normalisation of the distribution $\rho$ can be expressed using Lagrange parameters $\beta^i$ and $\alpha$, and entropy maximisation is:

$$\delta \operatorname{Tr}\Big[\rho\big(\log\rho + \sum_i \beta^i Q_i + \alpha\big)\Big] = 0 \;\Rightarrow\; \operatorname{Tr}\Big[\delta\rho\big(\log\rho + 1 + \alpha + \sum_i \beta^i Q_i\big)\Big] = 0. \qquad (13)$$

Hence the **maximal entropy states** are of the Gibbs form,

$$\rho \propto e^{-\sum_i \beta^i Q_i}. \qquad (14)$$

The question as to *what charges $Q_i$ occur* can also be given a rough answer using physical intuition, and in fact we have already answered it. With local interactions, finite regions that are far apart should be independent, and thus the entropy should be additive. Therefore, it can only be related, in (13), to *extensive conserved quantities*, of the form (11) with local or quasi-local densities.

The $\beta^i$'s form a system of coordinates in the manifold of maximal entropy states. They are the only parameters encoding information about the initial, generic state after relaxation has occurred. These are the "generalised inverse temperatures", or "generalised chemical potentials", which I will simply refer to as the Lagrange parameters. Eq. (14) means that, formally, the classical probability distribution, or the quantum density matrix, is proportional to the exponential $e^{-\sum_i \beta^i Q_i}$. Crucially, the states (14) have averages that are invariant under translations in space and time (homogeneity and stationarity), and that factorise as local observables are brought far from each other (clustering).

The form (14) is of course quite formal, and one needs to define the state more precisely. If the series $\sum_i \beta^i Q_i$ actually truncates and the charges $Q_i$ are local, there is a variety of ways to make it rigorous in the context of $C^*$ algebras, for instance directly as an infinite-volume limit, via the Kubo-Martin-Schwinger (KMS) relation (in the quantum case) or the Dobrushin-Lanford-Ruelle (DLR) equations (in the classical case), or by a precise notion of entropy maximisation, see [119, 120] for discussions. A general formulation, based on tangents to a manifold of states and which account for quasi-local charges, is developed in [121]. In all cases, one basic property is that averages $\langle\cdots\rangle_{\underline{\beta}}$ evaluated in maximal entropy states satisfy

$$-\frac{\partial}{\partial\beta^i}\langle o\rangle_{\underline{\beta}} = \int dx\, \langle o q_i(x,0)\rangle^{\mathrm{c}}_{\underline{\beta}} \qquad (15)$$

for any observable $o$, where the upperscript c indicates that we must take the connected correlation function $\langle o_1 o_2\rangle^{\mathrm{c}}_{\underline{\beta}} = \langle o_1 o_2\rangle_{\underline{\beta}} - \langle o_1\rangle_{\underline{\beta}}\langle o_2\rangle_{\underline{\beta}}$ (note that it doesn't matter if $q_i(x,0)$ is on the left or right of $o$ even in the quantum case, as it is assumed to commute with (14)). This equation has the geometric interpretation that the conserved charge $Q_i$ lies in the tangent space of the manifold of maximal entropy states. The finiteness of (15) requires at least fast enough algebraic clustering[4].

Maximal entropy states admit a number of properties that will be essential in what follows. It turns out that in order to establish these, Eq. (15) is sufficient as a definition of the $\beta^i$'s; there is no need for the formal expression (14). In particular, the general structure is the same independently of the microscopic setup (quantum or classical, deterministic or stochastic)!

Let me denote by

$$\mathsf{q}_i = \langle q_i(0,0)\rangle_{\underline{\beta}}, \quad \mathsf{j}_i = \langle j_i(0,0)\rangle_{\underline{\beta}} \qquad (16)$$

the average densities and currents, as functions of $\underline{\beta}$. First, (15) implies the symmetry

$$\frac{\partial \mathsf{q}_j}{\partial \beta^i} = \frac{\partial \mathsf{q}_i}{\partial \beta^j}. \qquad (17)$$

---

[4]As one-dimensional systems with local interaction cannot display thermal phase transitions, clustering is exponential if the series in (14) truncates and all charges are local [122].

This in turn implies the existence of a **free energy density** f such that

$$q_i = \frac{\partial f}{\partial \beta^i}. \tag{18}$$

Clearly, its formal expression is $f = \lim_{L \to \infty} L^{-1} \log \text{Tr}\left[e^{-\sum_i \beta^i Q_i}\right]$, where $L$ is the length of the system. However, we did not actually need this in order to define f; only (15) was used. Now consider the inner product

$$(o_1, o_2) = \int dx \, \langle o_1(0,0) o_2(x,0) \rangle^c_{\underline{\beta}} \tag{19}$$

on the space of local and quasi-local observables, here taken to be hermitian (or real) for simplicity[5]. Eq. (17) simply expresses the symmetry of this inner product on the subspace of conserved densities. From this, it is convenient to introduce the symmetric **static covariance matrix** $C_{ij}$,

$$C_{ij} = (q_i, q_j) = -\frac{\partial^2 f}{\partial \beta^i \partial \beta^j} = -\frac{\partial q_i}{\partial \beta^j}, \quad C_{ij} = C_{ji}. \tag{20}$$

This inner product is positive semidefinite, since

$$\int dx \, \langle o(0,0) o(x,0) \rangle^c_{\underline{\beta}} = \lim_{L \to \infty} \frac{1}{L} \langle Q_o^2 \rangle_{\underline{\beta}} \geq 0, \quad Q_o = \int_0^L dx \left[ o(x,0) - \langle o(0,0) \rangle_{\underline{\beta}} \right]. \tag{21}$$

Thus C is positive, and the set $\{Q_i\}$ should be chosen such that it is in fact strictly positive. As a consequence, f is convex.

Second, note that there are as many parameters $\beta^i$ as there are conserved densities $q_i(x,t)$. In fact, by convexity of f, the coordinate map $\underline{\beta} \mapsto \underline{q}$, from (an appropriate space of) Lagrange parameters to averages of conserved densities in maximal entropy states, is a bijection. That is, the set of averages can be used to fully characterise the state. As a consequence, all averages of local or quasi-local observables in maximal entropy states can be seen as functions of $\underline{q}$. For the average currents $j_i$, we may thus write

$$j_i = j_i(\underline{q}). \tag{22}$$

The dependence of the average currents on the average densities are what we will refer to as the **equations of state** of the model. These are model dependent functions.

Third, the equations of state satisfy quite surprising relations. Indeed, there is a symmetry mimicking (17), which is a ballistic analogue of the *Onsager reciprocity relations*. Seeing the average currents as functions of $\underline{\beta}$, these are (see [1,123]; their most general version, and an in-depth discussion, is provided in [124])

$$\frac{\partial j_j}{\partial \beta^i} = \frac{\partial j_i}{\partial \beta^j}. \tag{23}$$

One quick way of deriving this is as follows. The conservation laws in (11) implies the existence of **height fields** $\varphi_i(x,t)$ of total differentials

$$d\varphi_i(x,t) = q_i(x,t)dx - j_i(x,t)dt. \tag{24}$$

---

[5]For generic observables in quantum systems, the order of the observables in (19) now matters. In order for the formulae discussed below at the diffusive level to hold, an appropriate symmetrisation of the product of observables must be taken, such as $(o_1(0,0)o_2(x,0) + o_2(x,0)o_1(0,0))/2$ or more involved expressions; see for instance the review [56].

Then $\langle q_i(x,t)j_j(0,0)\rangle^c_{\underline{\beta}} = -\langle(\partial_x\varphi_i)(x,t)(\partial_t\varphi_j)(0,0)\rangle^c_{\underline{\beta}} = \langle\varphi_i(x,t)(\partial_x\partial_t\varphi_j)(0,0)\rangle^c_{\underline{\beta}}$ where we used space-translation invariance, $= -\langle(\partial_t\varphi_i)(x,t)(\bar{\partial}_x\varphi_j)(0,0)\rangle^c_{\underline{\beta}}$ using time-translation invariance, $= \langle j_i(x,t)q_j(0,0)\rangle^c_{\underline{\beta}}$. Clustering has been used in order for height field correlation functions to be well defined. Using (15), this implies (23). As a consequence, there must exist a **free energy flux** g such that

$$j_i = \frac{\partial g}{\partial \beta^j}. \tag{25}$$

The free energy flux g is, however, not obviously convex.

Fourth, seeing $j_i$ as functions of $\underline{q}$ as per (22), one defines the **flux Jacobian**

$$A_i^{\ j} = \frac{\partial j_i}{\partial q_j}. \tag{26}$$

This matrix will play an important role in describing hydrodynamics below. The symmetry (23) expresses that of a matrix built out of C and A, conventionally denoted by

$$B_{ij} = (j_i, q_j) = -\frac{\partial^2 g}{\partial\beta^i\partial\beta^j} = -\frac{\partial j_i}{\partial\beta^j} = \sum_k A_i^{\ k}C_{kj}, \quad B_{ij} = B_{ji}, \tag{27}$$

where the last equality of the first statement follows from the chain rule of differentiation. This symmetry can then be expressed as

$$AC = CA^{\mathrm{T}} \quad (\text{that is, } \sum_k A_i^{\ k}C_{kj} = \sum_k C_{ik}A_j^{\ k}). \tag{28}$$

Let us denote by $C^{ij}$ the *inverse* of the positive matrix $C_{ij}$, that is $\sum_k C^{ik}C_{kj} = \sum_k C_{ik}C^{kj} = \delta^i_j$. Writing an arbitrary conserved density as a linear combination $\sum_{i,j} v_i C^{ij} q_j(x,t)$, which can always be done, the inner product (19), with (20), induces an inner product on the coefficient vectors $\underline{v}$ (with components $v_i$):

$$\langle\underline{v},\underline{w}\rangle = (\underline{v}\cdot C^{-1}\underline{q}, \underline{w}\cdot C^{-1}\underline{q}) = \underline{v}\cdot C^{-1}\underline{w} = \sum_{ij} v_i C^{ij} w_j \tag{29}$$

where the first equality is the definition of $\langle\cdot,\cdot\rangle$ on vectors. The matrix A naturally acts on such vectors. Thanks to (28), under this inner product, A is symmetric: $\langle\underline{v}, A\underline{w}\rangle = \underline{v}\cdot C^{-1}A\underline{w} = \underline{v}\cdot A^{\mathrm{T}}C^{-1}\underline{w} = A\underline{v}\cdot C^{-1}\underline{w} = \langle A\underline{v},\underline{w}\rangle$. Therefore, A is diagonalisable and has real eigenvalues. The eigenvectors will be interpreted below as the **normal modes** of hydrodynamics, and the eigenvalues, as their associated **effective velocities** (or "generalised sound velocities"). Parametrising eigenvectors $h_{j;\ell}$ and eigenvalues $v_\ell^{\mathrm{eff}}$ by an index $\ell$,

$$\sum_j A_i^{\ j} h_{j;\ell} = v_\ell^{\mathrm{eff}} h_{i;\ell}. \tag{30}$$

Finally, as an application of the above structure, let us evaluate the **Drude weight**. This is a quantity which, if nonzero, represents the fact that there is *ballistic transport* in the model. Although it is a transport quantity, the Kubo formula relates it to a quantity that is purely a property of the thermodynamic state: the time-averaged, space-integrated current two-point functions. With many conservation laws, the Drude weight is in fact a matrix, and the Kubo formula is

$$D_{ij} = \lim_{t\to\infty} \frac{1}{2t} \int_{-t}^{t} ds \int dx \, \langle j_i(x,s)j_j(0,0)\rangle^c = \lim_{t\to\infty} \frac{1}{2t} \int_{-t}^{t} ds \, (j_i(s), j_j(0)), \tag{31}$$

where in the second equality I use the inner product (19), and the arguments of the fields are times. Time evolution can be expected (and in some cases proven) to be unitary on the Hilbert space induced by $(\cdot,\cdot)$. By the ergodic theorem, the result is a projection on the kernel of the time evolution, which should be identified with the subspace of conserved quantities, of which $q_i$ form a basis:

$$\mathsf{D}_{ij} = (j_i, \mathbb{P}j_j) = (\mathbb{P}j_i, j_j) = (\mathbb{P}j_i, \mathbb{P}j_j)\,, \tag{32}$$

where the projection acts as

$$\mathbb{P}o = \sum_{j,k} q_j \mathsf{C}^{jk}(q_k, o)\,. \tag{33}$$

Thus [125],

$$\mathsf{D}_{ij} = \sum_{kl} (j_i, q_k) \mathsf{C}^{kl}(q_l, j_j)\,. \tag{34}$$

This is an example of a *hydrodynamic projection* formula, here giving a more complete version of the so-called Mazur bound for the Drude weight [126–128]. Using the matrix (27), we therefore obtain various representations of the Drude weight,

$$\mathsf{D} = \mathsf{B}\mathsf{C}^{-1}\mathsf{B}^{\mathsf{T}} = \mathsf{A}\mathsf{C}\mathsf{A}^{\mathsf{T}} = \mathsf{A}^2\mathsf{C}\,. \tag{35}$$

It is worth emphasising that, although the matrices A, B and D have interpretations related to transport and to hydrodynamics, they are *purely properties of the thermodynamic state* – no hydrodynamic approximation is made. The equations they satisfy follow solely from statistical mechanics as encoded in (15).

**Remark 2.1.** *Gibbs states, which take the form (14), are usually associated with equilibrium. Equilibrium can be conventionally defined by the requirement of time-reversal invariance. In conventional hydrodynamics, the only charges that do not possess this invariance are the momentum charges (there is a single one in one dimension), which can be gotten rid of by going to the co-moving frame. Thus (14) is indeed at equilibrium, in this frame. In general, especially in integrable systems, however, this is not the case, and (14) are, generically, truly non-equilibrium states.*

**Remark 2.2.** *The notation using up and down indices points to the understanding of $\mathsf{C}_{ij}$ as a "metric" in the space of conserved densities, with which indices can be raised and lowered. Under this notation, the conserved densities are "vectors" and the Lagrange parameters, "covectors". This makes all equations more transparently consistent. Using this notation, one might in fact write $\mathsf{B}_{ij} = \mathsf{A}_{ij}$. As far as I am aware, this notation was first introduced in [123].*

**Example 2.3.** *In a conventional, non-integrable, one-dimensional Galilean gas, there are three conserved quantities: the total mass of all particles $Q_0 = mN$ where $N$ is the total number of particles and $m$ their mass (assuming a single specie), the total momentum $Q_1 = P$ and the total energy $Q_2 = H$ (identified with the Hamiltonian function or operator if the dynamics is Hamiltonian). In general, the states take the form $\mathrm{e}^{-\beta(H-\mu N-\nu P)}$ where $\beta$ is the inverse temperature, $\mu$ the chemical potential, and $\nu$ the Galilean boost parameter. This is a conventional thermalised Gibbs state, up to a Galilean boost which can be taken away by changing the laboratory frame. The case considered in Section 2.1 was that where we neglect the effects of $Q_2$ (that is, the temperature is kept fixed), which is justified in many real situations.*

*Galilean invariance is the statement that the mass current is exactly equal to the density of momentum, this being true at the level of observables, $j_0 = q_1$. Taking averages, this gives $\mathsf{A}_0^{\ 0} = 0, \mathsf{A}_0^{\ 1} = 1, \mathsf{A}_0^{\ 2} = 0$. This part of the equations of state is quite simple. On the other hand the average current of momentum is, by definition, simply related to the pressure $\mathsf{P} = \mathsf{P}(\mathsf{q}_0, \mathsf{q}_2)$ of the gas in this state, $\mathsf{j}_1 = \mathsf{P} + \mathsf{q}_1^2/\mathsf{q}_0$: the pressure is the current of momentum in the fluid's rest frame,*

Macroscopic    Mesoscopic (fluid cells)    Microscopic

$(x', t')$

$e^{-\sum_i \beta^i(x',t')Q_i}$

$e^{-\sum_i \beta^i(x,t)Q_i}$

$(x, t)$

Figure 3: The separation of scales at the basis of the hydrodynamic approximation.

*and the second term is a consequence of Galilean invariance. Thus $A_1^0 = \partial_0 P - q_1^2/q_0^2, A_1^1 = 2q_1/q_0, A_1^2 = \partial_2 P$. Finally, by Galilean invariance, the energy current also is expressible in terms of the pressure, $j_2 = (3q_1/2q_0)P + q_1^3/q_0^2$. One can easily work out the remaining matrix elements of $A$. Thus the equations of state are completely determined by the way the pressure $P$ depends on the mass and energy densities $q_0, q_2$. The symmetry relations (20) and (28) give nontrivial, pressure-dependent constraints on the static covariant matrix $C$, which encode the fact that there is an underlying short-range microscopic model.*

## 2.3 Local entropy maximisation and Euler hydrodynamics

Consider some initial state, in a many-body system as discussed in Section 2.2, whose averages we denote $\langle \cdots \rangle$. This state will generically be inhomogeneous and dynamical (non-stationary): average values of observables depend on the position, and change as time passes. Now consider some average $\langle o(x,t) \rangle$ of a local observable at $x$ evolved up to time $t$. In the quantum mechanical setting, we are thinking about the Heisenberg picture; for a deterministic classical model, the evolution is obtained by the Hamiltonian flow on the Poisson manifold; for a stochastic model, it is the stochastic dynamics in the classical case, or the Lindbladian dynamics in the quantum case. The basic assumption of hydrodynamics is that of *local entropy maximisation*: to a good approximation, this average can be evaluated as the average of $o(0, 0)$ in a maximal entropy state with $x, t$-dependent Lagrange parameters,

$$\langle o(x,t) \rangle \approx \langle o(0,0) \rangle_{\underline{\beta}(x,t)}. \tag{36}$$

By homogeneity and stationarity of the maximal entropy state $\langle \cdots \rangle_{\underline{\beta}(x,t)}$ [*not* of the state $\langle \cdots \rangle$], we can position the observable at any point on the right-hand side, and here we chose $(0, 0)$. The state $\langle \cdots \rangle_{\underline{\beta}(x,t)}$, that appears in the hydrodynamic approximation, itself depends on the position in space-time where the observable lies on the left-hand side of (36), via the space-time dependence of the Lagrange parameters $\underline{\beta}(x, t)$. But crucially, it does not depend on the observable itself. The same state describes any local or quasi-local observable at $x, t$.

This is the separation of scales between the macroscopic, mesoscopic (fluid cells) and microscopic at the basis of hydrodynamics. See Fig. 3.

The validity of this approximation is hard to establish in nontrivial systems. An exception is the hard rod gas, where a version of it has been proven rigorously for a wide family of initial states and in an appropriate long-time limit [77]. In general, we expect the approximation to become exact in the limit where typical lengths in space and time over which variations

of local averages occur are infinitely large. This can happen for various reasons: it may be dynamical, developing over (infinitely) long times, or it may be that the initial state is chosen appropriately. Infinitely large variation lengths in space and time do not simply imply that the state is homogeneous and stationary: the limit can still be nontrivial, as we may simultaneously take the position $x, t$ of the observable to be infinitely far in space-time. This is what I will refer to as the Euler scaling limit. For instance, the initial state may be like a maximal entropy state but with modulated Lagrange parameters, $e^{-\sum_i \int dy\, \beta^i(y,0)q_i(y)}$, and we may take the simultaneous limit where this modulation occurs on large distances and the point $(x, t)$ is scaled. In this case, we expect the limit to give, for almost all values of $x, t$, the result of the above approximation for some $\beta^i(x, t)$'s:

$$\lim_{\lambda \to \infty} \frac{\text{Tr}\left(e^{-\sum_i \int dy\, \beta^i(\lambda^{-1}y,0)q_i(y)} o(\lambda x, \lambda t)\right)}{\text{Tr}\left(e^{-\sum_i \int dy\, \beta^i(\lambda^{-1}y,0)q_i(y)}\right)} = \langle o(0,0)\rangle_{\underline{\beta}(x,t)}. \tag{37}$$

This is just an example; states with modulated Lagrange parameters are not the only inhomogeneous states that one can construct, and are not necessarily more natural than other states. Without the Euler scaling limit, the approximation is not exact, but may be a good approximation. It is difficult to precisely characterise the error made in a universal way, although certainly the inverse of typical variation lengths as well as the particles' scattering lengths will be involved.

Once local entropy maximisation is assumed, one can derive, from the microscopic dynamics, the Euler equations for the model under consideration. The arguments goes as follows.

First, consider the conservation laws in (11) in their integral form, over some contour, say $[0, X] \times [0, T]$:

$$\int_0^X dx\,(q_i(x,T) - q_i(x,0)) + \int_0^T dt\,(j_i(X,t) - j_i(0,t)) = 0. \tag{38}$$

Evaluate these within the state $\langle \cdots \rangle$. Using the local entropy maximisation assumption, we get

$$\int_0^X dx\,(\mathsf{q}_i(x,T) - \mathsf{q}_i(x,0)) + \int_0^T dt\,(\mathsf{j}_i(X,t) - \mathsf{j}_i(0,t)) = 0, \tag{39}$$

where I use the notation

$$\mathsf{q}_i(x,t) = \langle q_i(0,0)\rangle_{\underline{\beta}(x,t)}, \quad \mathsf{j}_i(x,t) = \langle j_i(0,0)\rangle_{\underline{\beta}(x,t)}. \tag{40}$$

Equations (39) are relations for the averages within maximal entropy states, thus giving constraints on the Lagrange parameters. If the quantities are differentiable, they can be re-written in terms of derivatives,

$$\partial_t \mathsf{q}_i(x,t) + \partial_x \mathsf{j}_i(x,t) = 0. \tag{41}$$

There is, however, a conceptual difference between the derivatives appearing in the microscopic conservation equations in (11), and those appearing in (41). Indeed, the latter should be understood as large-scale derivatives, encoding variations amongst fluid cells. They represent the large-scale integral form (39) of the conservation laws obtained under local entropy maximisation.

Recall that the set $\{\mathsf{q}_i(x,t)\}$ can be used to completely characterise the maximal entropy state at the point $x, t$, and recall the equations of state (22) and the flux Jacobian (26). Putting these into (41), we obtain what I will refer to as the **Euler hydrodynamic equations** for the system of interest. In the so-called quasilinear form, by using the chain rule these are

$$\partial_t \mathsf{q}_i(x,t) + \sum_j \mathsf{A}_i{}^j(x,t)\partial_x \mathsf{q}_j(x,t) = 0, \tag{42}$$

where $A_i^{\ j}(x,t) = A_i^{\ j}(\underline{q}(x,t))$ depends on $x,t$ only via the state $\underline{q}(x,t)$. These are "wave equations" for all space-time-dependent coordinates $q_i(x,t)$ characterising the local maximal entropy state. The form of the Euler equations depend on the equations of state of the model and the number of conserved quantities, but nothing else – at the emerging Euler scale, very little of the microscopic dynamics remains. Euler hydrodynamic equations are hyperbolic equations, and a large amount of material exists on their solutions and behaviours [50]. We will see a small part of this below when studying the Riemann problem. We note that the quasi-linear form can also be written in terms of the local Lagrange parameters $\beta^i(x,t)$. Indeed, using (20), (28) and (42), we obtain

$$\partial_t \beta^i(x,t) + \sum_j A_j^{\ i}(x,t)\partial_x \beta^j(x,t) = 0. \tag{43}$$

Note that once the hydrodynamic equations (41), or equivalently (42) or (43), are solved, then we know the exact local state at every space-time point $(x,t)$. In particular, we can evaluate the average of any observable $o(x,t)$ lying within the fluid cell at $x,t$, by using $\langle o(x,t)\rangle = \langle o(0,0)\rangle_{\underline{\beta}(x,t)}$, as per (36) (assuming the Euler scaling limit is taken exactly).

Consider the eigenvalue equation (30). The discussion there means that there is a matrix R which diagonalises A:

$$RAR^{-1} = v^{\mathrm{eff}}, \tag{44}$$

where $v^{\mathrm{eff}} = \mathrm{diag}(v_i^{\mathrm{eff}})$. Since A is a function of $\underline{q}$, so is R. Suppose we can find functions $n_j(\underline{q})$ whose Jacobian is R, that is

$$\frac{\partial n_i}{\partial q_j} = R_i^{\ j}. \tag{45}$$

Then (42) implies

$$\partial_t n_i + v_i^{\mathrm{eff}}\partial_x n_i = 0. \tag{46}$$

As R is invertible, the functions $n_i(\underline{q})$ are invertible. Thus, they can be seen as *new coordinates for the maximal entropy state*. These coordinates are referred to as the **normal modes** of the hydrodynamic system. Having the normal modes is very useful in order to solve specific hydrodynamic problems. Eq. (46) simply means that the $i$th normal mode is convectively transported at the velocity $v_i^{\mathrm{eff}}$. The set of equations (46) does not in general decouple: the velocity $v_i^{\mathrm{eff}}$ depends on all $n_j$s.

By changing coordinates, it is clear that, from the viewpoint of the manifold of maximal entropy states, A does not have a fundamental meaning – only its spectrum does. However, the set of coordinates corresponding to physical conserved densities is special. In this sense, the class of flux Jacobians obtained under constant similarity transform (linear changes of coordinates) is physically meaningful – this is the class of "natural" flux Jacobians. Of course, in general, there does not exist $n_i$ that are linear combinations of $q_j$ such that (45) hold. That is, there does not exist a set of conserved densities that diagonalise the flux Jacobian. If, however, the natural flux Jacobian is *constant*, then there exists a set of conserved densities that diagonalises A. These are very special cases, where a lot of the hydrodynamics simplify. In fact, as we will see later, all free-particle models are of this type, as well as 1+1-dimensional conformal hydrodynamics. These can be referred to as **free hydrodynamic systems** (see the discussion in [95]).

Finally, observe that Euler equations are first-order in derivatives. They are time-reversible, and completely determined by the thermodynamics of the system (encoded by the equations of state of the maximal entropy manifold). In this sense, they are as near to the thermodynamic point as possible in Fig. 2 – they constitute some "scaling limit" towards this point.

The latter observations are given more meaning by looking at the entropy density. Using (14) and the formal definition of the free energy density just below (18), the entropy density

is easily expressible as

$$s = \sum_i \beta^i q_i - f. \tag{47}$$

Avoiding formal expressions, this can simply be taken as a definition of $s$. Using the Euler hydrodynamic equation (41) and the free energy flux (25), it is a simple matter to observe that there exists a conserved entropy current,

$$\partial_t s + \partial_x j_s = 0, \tag{48}$$

with flux

$$j_s = \sum_i \beta^i j_i - g. \tag{49}$$

Thus, Euler hydrodynamics on differentiable fluid fields conserve entropy.

In the above statement, the specification "on differentiable fluid fields", which the derivation assumes, may in fact be broken. Indeed, Euler hydrodynamics is often unstable. It may for instance develop discontinuities – shocks – in finite time; in these cases, the solution to the Euler equation is said to be weak, see [50]. At shocks, entropy is no longer conserved, and a fuller understanding is obtained by considering diffusion, discussed in Section 2.4. Shocks in Euler equations are described more precisely in Section 2.5. Further, in realistic Euler equations fluid fields may also become non-differentiable while remaining continuous, at which loci energy may fail to be conserved, a problem known as Onsager's conjecture [129]. See for instance [130] and [131][6].

**Remark 2.4.** *The hydrodynamic assumption can be made to sound weaker: it is sufficient to assume that averages of local and quasi-local observables may only depend on the average conserved densities at the same space-time point. Making this assumption within a homogeneous state, we deduce that the average currents must be fixed by the equations of state.*

**Example 2.5.** *In conventional Galilean systems, see Example 2.3, it is possible to bring (41) into a form that is more familiar, as discussed in Section 2.1. One defines a "fluid velocity" $v = q_1/q_0$, and combining with the equations of Example 2.3, one finds in particular*

$$\partial_t v + v \partial_x v = -\frac{1}{q_0} P. \tag{50}$$

## 2.4 Constitutive relations and diffusive hydrodynamics

Beyond the Euler scale, there are corrections. These corrections should depend on the variations in space-time of the fluid state, as we no longer take the Euler scaling limit of infinite variation lengths. Variation lengths are still large, but not infinite, and we are looking for the first correction. The way this is taken into account is via the constitutive relations. These are based on two assumptions. First, we assume that it is *still* possible to fully describe the state in space-time by the averages of conserved densities,

$$q_i(x, t) = \langle q_i(x, t) \rangle. \tag{51}$$

More precisely, we say that a time slice – the set $q_i(y, t)$ for all $i$ and all $y \in \mathbb{R}$ for fixed $t$ – gives a full description of the fluid at time $t$. Note how I have adjusted a bit my notation: the local state is not of maximal entropy, but I still use $q_i(y, t)$ for describing state coordinates. From the information at the time slice $t$, the assertion is that it is possible (in principle) to evaluate all averages of other local and quasi-local observables at $x, t$ for any $x$. Of course,

---

[6]I thank H. Spohn for pointing out this phenomenon to me.

a local observable's average at $x, t$ cannot depend on all of $\mathsf{q}_i(y, t)$ for $y \in \mathbb{R}$; by locality, they should just depend on those $y$ near to $x$. The difference with the Euler scale is that it is no longer purely determined by $\mathsf{q}_i(x, t)$. Instead, the second assumption is that *the first derivative* is involved (as this is a first correction). Let me denote the average currents, within this approximation, as

$$\mathcal{J}_i(x, t) = \langle \mathsf{j}_i(x, t) \rangle. \tag{52}$$

Then, local entropy maximisation, (36) with (22), is modified to

$$\mathcal{J}_i(x, t) = \mathsf{j}_i(\underline{\mathsf{q}}(x, t)) - \frac{1}{2} \sum_j \mathcal{D}_i^{\ j}(\underline{\mathsf{q}}(x, t)) \partial_x \mathsf{q}_j(x, t), \tag{53}$$

where the factor $1/2$ is by convention, and $\mathcal{D}_i^{\ j}$ is the **diffusion matrix**. Equation (53) is what is usually referred to as the **constitutive relations**. The microscopic conservation laws,

$$\partial_t \mathsf{q}_i + \partial_x \mathcal{J}_i = 0, \tag{54}$$

then give **diffusive hydrodynamics**. The second-order derivative terms (coming from the derivative in $\mathcal{J}_i$ and that in the conservation law) in the resulting equations are the **Navier-Stokes** terms, using this terminology in a generalised way. Again, the functions $\mathcal{D}_i^{\ j}(\mathsf{q})$ are characteristics of the microscopic model. However, they are no longer part of its thermodynamics (as the equations of state (22) were) – they go beyond thermodynamics, part of hydrodynamics.

Diffusive hydrodynamics is irreversible, and not completely determined by the maximal entropy states of the system. Hence it is truly away from thermodynamics in Fig. 2.

In fact, diffusive hydrodynamics gives rise to an "arrow of time": *entropy production*. In order to see this, consider the **Onsager matrix** $\mathcal{L} = \mathcal{D}\mathsf{C}$. One can show that this matrix is positive (by which I mean non-negative), $\mathcal{L} \geq 0$; I do this below. Positivity of $\mathcal{L}$ then implies that Eq. (48) is modified by a positive non-conservative term. Indeed, let us define the entropy density $\mathsf{s}$ as in (47), but now with the average conserved densities $\mathsf{q}_i$ defined by (51), without assuming local entropy maximisation. Let us also denote by $\mathsf{j}_s$ the entropy flux as a function of those average densities $\mathsf{q}$, as per (49). Using the conservation equation (54) and assuming the derivative expansion (53), we obtain a correction to (48),

$$\partial_t \mathsf{s} + \partial_x \mathsf{j}_s = \frac{1}{2} \sum_{ij} \beta^i \partial_x (\mathcal{D}_i^{\ j} \partial_x \mathsf{q}_j) = -\frac{1}{2} \sum_{ijk} \beta^i \partial_x (\mathcal{D}_i^{\ j} \mathsf{C}_{jk} \partial_x \beta^k). \tag{55}$$

This is equivalent to

$$\partial_t \mathsf{s} + \partial_x \mathcal{J}_s = \frac{1}{2} \sum_{ijk} (\partial_x \beta^i) \mathcal{D}_i^{\ j} \mathsf{C}_{jk} \partial_x \beta^k = \frac{1}{2} \sum_{ik} \partial_x \beta^i \mathcal{L}_{ik} \partial_x \beta^k \geq 0, \tag{56}$$

where the entropy flux receives a correction from the diffusive components of the currents,

$$\mathcal{J}_s = \mathsf{j}_s + \frac{1}{2} \sum_{ik} \beta^i \mathcal{L}_{ik} \partial_x \beta^k = \sum_i \beta^i \mathcal{J}_i - \mathsf{g}. \tag{57}$$

The right-hand side of Eq. (56) is indeed positive if $\mathcal{L}$ is positive.

Positivity of $\mathcal{L}$, much like the statement $\mathsf{C} > 0$, is a consequence of microscopic, statistical-mechanics considerations. It can be shown by using the Green-Kubo formula, which relates it to two-point functions. Considering the definition (53) and its consequences, via linear

response, on two-point functions, one may show that (see for instance [7, 132], and, in the present context, the derivation presented in [123])

$$\mathcal{L}_{ij} = \lim_{t \to \infty} \int_{-t}^{t} ds \left[ (j_i(s), j_j(0)) - D_{ij} \right], \tag{58}$$

where $D_{ij}$ is the Drude weight (35), and we use the notation of the right-hand side of the last equation of (35). By using the relations amongst the matrices $A, B, C, D$ derived in Section 2.2, and recalling that the projection $\mathbb{P}$ on the space of conserved densities gives the Drude weight, Eq. (32), we obtain

$$\mathcal{L}_{ij} = \lim_{t \to \infty} \int_{-t}^{t} ds \left( [1 - \mathbb{P}] j_i(s), [1 - \mathbb{P}] j_j(0) \right). \tag{59}$$

Further, by stationarity,

$$\mathcal{L}_{ij} = \lim_{t \to \infty} \frac{1}{2t} \int_{-t}^{t} ds \int_{-t}^{t} ds' \left( [1 - \mathbb{P}] j_i(s), [1 - \mathbb{P}] j_j(s') \right), \tag{60}$$

and thus $\mathcal{L} \geq 0$ by positive semi-definiteness of the inner product. In fact, the Onsager coefficients can be bounded from below by strictly positive values if there are "quadratically extensive" conserved quantities [133, 134].

Hydrodynamic entropy production can be puzzling: the microscopic model might have deterministic, reversible dynamics, hence preserves its total entropy. Entropy appears to be increasing in hydrodynamics because of the coarse graining. Effectively, the distribution of large structures at the Euler scale contains entropy, and as the fluid moves (convection), these structures become more homogeneous (their entropy decreases), while, because of diffusion, entropy accumulated into small-scale degrees of freedom increases. There is a transfer of entropy from large scales to small scales. The total hydrodynamic entropy $\int dx\, s(x, t)$ does not capture the entropy contained in the Euler-scale structures, just that at small scales in the fluid cells' thermodynamic states.

It is also important to emphasise that although there may be diffusion, there *does not need to be any external noise*: the randomness usually associated to diffusion, in deterministic systems, will come from the random initial condition. This is still true diffusion.

As mentioned, diffusive hydrodynamics gives corrections beyond the exact Euler scaling limit. For instance, if the variation lengths of the initial state are not infinite, then diffusive hydrodynamics will give a more precise evolution than Euler hydrodynamics. The time range in which Euler / diffusive hydrodynamics is appropriate depends on the problem under consideration. As mentioned, the dynamics under Euler hydrodynamics may lead the fluid to be less smooth – such as when shocks develop, discontinuities in the Euler solution. In this case, diffusive hydrodynamics "kicks in" before discontinuities appear in the actual fluid, and the diffusion terms smooth them out. Diffusion explains the associated lack of entropy conservation in weak solutions of Euler equations discussed after Eq. (48). In general, we expect diffusion to smooth out the fluid's state at very long times and eventually, in finite (but large) volume, homogeneity to be reached. Thus, roughly, as time goes forward, the relevant equations are those from left to right in Fig. 2.

**Remark 2.6.** *It is important to remark that there is now a* gauge ambiguity*: the conserved densities $q_i(x, t)$ are fixed by their relation to the total charges $Q_i$, as per (11), only up to total derivatives. But, such total derivatives change the second term in (53) – hence affect the diffusion matrix $\mathcal{D}_i{}^j$. This gauge ambiguity can be fixed by requiring an appropriate parity-time symmetry, see [123].*

**Remark 2.7.** *Although the diffusion matrix $\mathcal{D}_i{}^j$ is gauge dependent – it depends on the exact definition of the conserved densities $q_i(x,t)$ – the Onsager matrix is gauge invariant; see for instance [123]. A gauge invariant way of writing the modified current (53) is $\mathcal{J}_i = \mathsf{j}_i - \frac{1}{2}\sum_j \mathcal{L}_{ij}\partial_x\beta^j$, leading to the gauge-invariant hydrodynamic equation*

$$\sum_j \left[ \mathsf{C}_{ij}\partial_t\beta^j + \mathsf{B}_{ij}\partial_x\beta^j - \frac{1}{2}\partial_x(\mathcal{L}_{ij}\partial_x\beta^j) \right] = 0. \tag{61}$$

*Note that the Lagrange parameters $\beta^i$s are indeed defined in a gauge-invariant way. In order to be fully predictive, this equation has to be supplemented with a map from $\underline{\beta}(x,t)$ to local averages. As part of the hydrodynamic approximation, this map is fully fixed once the averages $\langle q_i(x,t)\rangle$ are fixed as functions of $\underline{\beta}(y,t)$ (for $y$ in a neighbourhood of $x$).*

**Remark 2.8.** *In generic one-dimensional systems, diffusion is often anomalous, and replaced by superdiffusion. In these cases, the derivative expansion is not valid, and one must appeal to the theory of nonlinear fluctuating hydrodynamics [135, 136], or to other constructions [134]. This is however beyond the scope of these notes. For integrable systems, my understanding is that the derivative expansion holds true for conserved currents associated to the integrable hierarchy of conserved charges, where the integrability structure constrains the dynamics. Hence this is the most relevant situation for the present notes. But for conserved currents associated with internal charges, nondiagonal scattering may lead to their being effectively "non-integrable", and superdiffusion may arise [96–98, 137–139]. See also the discussion in the closing remarks, Chapter 5.*

**Example 2.9.** *For the conventional Galilean hydrodynamics discussed in Section 2.1, the Navier-Stokes term comes from the diffusion matrix*

$$\mathcal{D} = 2\frac{\zeta}{\mathsf{q}_0} \begin{pmatrix} 0 & 0 \\ -\mathsf{q}_1/\mathsf{q}_0 & 1 \end{pmatrix}. \tag{62}$$

*This general form is fixed by Galilean invariance.*

## 2.5 The Riemann problem

One of the most iconic problem of hydrodynamics is the Riemann problem. This is the initial-value problem of hydrodynamics where the initial fluid's state is homogeneous on the line except for a single discontinuity at some point, say at the origin $x = 0$ (so that the states on the left and right are different). Here I'm concentrating on the Euler scale, but one can discuss this at the diffusive scale as well.

From the viewpoint of microscopic models, this is a very natural configuration that is used in order to generate non-equilibrium steady states [51–53]: the two half-infinite halves of the system play the role of infinitely large baths that can provide and absorb particles, energies and other conserved quantities, so that a steady flow may develop over time in the central region. This is sometimes called the partitioning protocol.

The initial condition is

$$\langle q_i(x,0)\rangle = \begin{cases} \mathsf{q}_i^{(\mathrm{l})} & (x < 0) \\ \mathsf{q}_i^{(\mathrm{r})} & (x > 0). \end{cases} \tag{63}$$

Clearly, this initial condition is scale invariant: it does not change under $(x,t) \mapsto (\lambda x, \lambda t)$. Since the Euler hydrodynamic equations (42) also are invariant under this scaling, it is natural to assume that the solution will also be, hence that the solution only depends on the *ray* $\xi = x/t$,

$$\mathsf{q}_i(x,t) = \mathsf{q}_i(\xi), \qquad \xi = x/t. \tag{64}$$

SciPost Phys. Lect.Notes 18 (2020)

The initial conditions give the asymptotics

$$\lim_{\xi \to -\infty} q_i(\xi) = q_i^{(l)}, \qquad \lim_{\xi \to +\infty} q_i(\xi) = q_i^{(r)}. \tag{65}$$

Equation (42) then simplifies to a set of ordinary differential equations,

$$\sum_j \left( A_i^{\ j} - \xi \delta_i^j \right) \frac{dq_j}{d\xi} = 0. \tag{66}$$

This is nothing else than the eigenvalue problem for the flux Jacobian, where $\xi$ is the eigenvalue and $dq/d\xi$ the eigenvector.

The natural solutions to this problem depend on the structure of the flux Jacobian. In fact, it depends if the Euler hydrodynamics is *linearly degenerate* or not. Linear degeneracy is the condition that, for every fluid mode $i$, the effective velocity $v_i^{\text{eff}}$ does not depend on the corresponding normal coordinate $n_i$. Integrable systems fall within this category – in addition to admitting infinitely-many conserved quantities. Let me discuss the situation assuming a finite number of conserved quantities (fluid modes).

There are then three types of (possibly "weak", i.e. with discontinuities) solutions to these equations. First, let me ask for $q_j$ to be continuous and differentiable. Then $\partial_\xi q_j$ is an eigenvector for the flux Jacobian $A$ with eigenvalue $\xi$. In fact, the discussion is simplified by going to the normal modes as per (46), in terms of which we obtain

$$(v_i^{\text{eff}} - \xi) \frac{dn_i}{d\xi} = 0. \tag{67}$$

Thus, for every $i$, we either have $v_i^{\text{eff}} = \xi$, or $dn_i/d\xi = 0$. For a given state $q^{(l)}$ on the left, say, there is a discrete set of $\xi$ for which we could have nonzero $dn_j/d\xi$. As a consequence, the state stays constant on the left, from $\xi = -\infty$ until such a value of the ray $\xi$ is reached. At this point, if the velocities are non-degenerate, then, a single normal mode can be set to have a nonzero derivative, say mode $j$, so we have $v_j^{\text{eff}} = \xi$. Then, as $\xi$ varies, $\underline{n}$ changes, to first order, linearly. As $v_i^{\text{eff}}$s are (expected to be) smooth, they stay far from $\xi$ except for $i = j$, thus only $n_j$ can keep having nonzero derivatives. In order for this to happen, we must satisfy the eigenvalue equation at $\xi + d\xi$, so $v_j^{\text{eff}}(\underline{n}(\xi + d\xi)) = \xi + d\xi$. This implies $dn_j/d\xi = 1/\partial_{n_j} v_j^{\text{eff}}$. If the mode is not linearly degenerate, the derivative on the right-hand side is nonzero, hence $n_j$ indeed changes infinitesimally. This means that once we choose the left-state eigenvalue $\xi$ (in a discrete set of choices), the only parameter that determines the state on the right is the length of the $\xi$ interval for which we solve the equation, as only a single normal mode could be modified along this interval. This is a **rarefaction wave**. It has the property that, on every ray, a single normal mode is travelling along this ray.

Crucially, this is, generically, not enough to span all possible states on the right, as we have a single continuous parameter for the rarefaction wave, yet a higher dimensional manifold of possible right-states. Although one can "stack" rarefaction waves for different modes, generically it is still not possible to connect the state on the left to that on the right with a series of rarefaction waves.

The second type of solutions are **shocks**. These are weak solutions, in that they are discontinuities. This means that we have to revert to the integral form of the Euler equations, (39). Suppose the solution is constant in some region of rays, except for a discontinuity at some $\xi$, where the state just to its left / right is $q_i^\pm$. Then integrating over a small rectangle around this discontinuity, as in (39) with $X = \xi T$, we obtain the Rankine-Hugoniot equation

$$\xi(q_i^- - q_i^+) + (j_i^+ - j_i^-) = 0. \tag{68}$$

We insert the equations of state, and obtain algebraic equations that relate the states just on the left and right of the shock. Given a shock velocity $\xi$, this is a map from the left state to the right state, so again we only have a single parameter.

Formally, one can put as many shocks as is desired at different velocities, and thus one obtains a large family of possible solutions, parametrised by the shock velocities. It is in general possible to relate any state in the left reservoir to any state in the right reservoir with such many-shock solutions. However, these solutions are generically unphysical. Indeed, a shock is a weak solution to the Euler equations, which means that the underlying physical system has fast variations – finite variations on scales which are much smaller than the Euler scales – around the shock's ray. Such variations lead to a breaking of the Euler equations, and at shocks, diffusive terms become important. Since, as we have shown, diffusion produce entropy, one must verify that the Ranquine-Hugoniot equation (68) leads, at the shock, to an *increase* of the entropy s from Eq. (47). It turns out that an equivalent condition is that the *characteristics*, the curves which follow the propagations of normal modes, are not "emitted" by the shock, but can only be "absorbed" by it – physically, entropy production looses the information of these modes, but cannot create it. See [50] for an in-depth discussion.

Finally, the third type are **contact discontinuities**. These are discontinuities which do not produce entropy at the Euler scale – that is, which do not give rise to entropy increase that is linear with time. A contact discontinuity is a discontinuity of a mode $n_j$ exactly at the ray $\xi = v_j^{\text{eff}}$. In this case, no characteristic is absorbed or emitted by the discontinuity. In linearly degenerate systems, as $\partial_{n_j} v_j^{\text{eff}} = 0$, the rarefaction wave analysis above leads to the possibility of having $dn_j/d\xi = \infty$, and thus a contact discontinuity. One can stack enough of these in order to connect the left and right states. Physically, it appears as though the system chooses the least entropy production possible, hence in linearly degenerate systems, no shock is produced.

**Remark 2.10.** *The partitioning protocol in a microscopic system does not lead, at early times, to a hydrodynamic description, because of the large variations. However, after a long enough time, the state can be assumed to be described by Euler hydrodynamics, and, scaling it, gives the Riemann problem. In fact, any initial condition with the given left and right asymptotics, approached fast enough, leads to the Riemann problem with these left and right states.*

**Remark 2.11.** *As we saw, the partitioning protocol gives rise, at large times and in any finite regions around $x = 0$, to a state determined by the Riemann problem of hydrodynamics. If this state carries currents, then this is a non-equilibrium steady state. If this happen, then we say that the system admits ballistic transport: transport of quantities unimpeded by diffusive effects. In particular, Fourier's law is broken. Euler hydrodynamics is, indeed, a theory for ballistic transport of hydrodynamic modes.*

## 2.6 Hydrodynamic correlation functions

Hydrodynamic ideas also allow us to go beyond the evaluation of one-point functions (36), and gives exact asymptotic results (in the Euler scaling limit, and diffusive corrections) for multi-point correlation functions as well.

One way to get to this is by considering linear responses of the hydrodynamic flow to perturbations of its initial state. Hydrodynamics will then describe the propagation of small waves on top of the fluid flow. The perturbation may be thought of as a small fluctuation on top of the fluid flow, or as an effect of an external action on the fluid. Assume $q_i(x, t)$ solves the Euler hydrodynamic equation (41). Consider a small perturbation $q_i \mapsto q_i + \delta q_i$. The way the small perturbation propagates is determined by asking that $q_i + \delta q_i$ also satisfies (41). We

obtain

$$\partial_t \delta q_i + \partial_x \Big[ \sum_j A_i^{\ j}(\underline{q}) \delta q_j \Big] = 0 \quad \text{(Euler scale, fluid background)}. \tag{69}$$

For simplicity, from now on in this section, I'll assume that the background flow is homogeneous and stationary – just a fixed maximal entropy state. Then the flux Jacobian is independent of space-time, and

$$\partial_t \delta q_i + \sum_j A_i^{\ j}(\underline{q}) \partial_x \delta q_j = 0 \quad \text{(Euler scale, stationary background)}. \tag{70}$$

One can also consider this at the diffusive level, and the fluid equation (54) gives, again on top of a homogeneous, stationary background,

$$\partial_t \delta q_i + \sum_j A_i^{\ j}(\underline{q}) \partial_x \delta q_j - \frac{1}{2} \sum_j \mathcal{D}_i^{\ j}(\underline{q}) \partial_x^2 \delta q_j = 0 \quad \text{(diffusive scale, stationary background)}. \tag{71}$$

If one considers fluctuations, an additional noise term must be added, a phenomenological representation of the statistical fluctuations that the particles making up the fluid as subjected to. This is connected to the diffusion matrix by the Einstein relation. We get what is referred to as *linear fluctuating hydrodynamics*. See for instance the discussion in [135]. For our present purpose, there is no need to add a noise term.

Suppose the small initial perturbation is local, at least on the Euler scale, or perhaps the lesser diffusive scale. Then the small wave propagating from it will give rise to correlations between the observable representing this local perturbation, and any other observable. One can think of this as the fluid equivalent of the usual relation between fluctuations and correlations in statistical mechanics. If $o(0,0)$ is the observable resulting from the perturbation, then (71) suggests

$$\partial_t \langle q_i(x,t) o(0,0) \rangle^{\text{hyd}} + \sum_j A_i^{\ j} \partial_x \langle q_j(x,t) o(0,0) \rangle^{\text{hyd}} - \frac{1}{2} \sum_j \mathcal{D}_i^{\ j} \partial_x^2 \langle q_j(x,t) o(0,0) \rangle^{\text{hyd}} = 0. \tag{72}$$

Here $\langle \cdots \rangle^{\text{hyd}}$ is some fluid-cell-averaged (connected) correlation function in the stationary background $\underline{\beta}$. Depending on the model, fluid-cell averaging may indeed be necessary for this equation to hold, as the long-wavelength propagation of perturbations that the equation describes cannot encode too much detail of this perturbation. There are many ways of doing fluid-cell averaging, see e.g. [140,141]. One way, under which (72) is expected to be correct, is by Fourier transform, as described below.

One can in fact "derive" (72) more formally as follows. By the microscopic conservation laws,

$$\partial_t \langle q_i(x,t) o(0,0) \rangle^{\text{hyd}} + \partial_x \langle j_i(x,t) o(0,0) \rangle^{\text{hyd}} = 0. \tag{73}$$

Now look at $\langle j_i(x,t) o(0,0) \rangle^{\text{hyd}}$. This can be obtained by considering a space-dependent Lagrange parameter $\beta^o(x,0)$ associated to $o$ in the initial state at time $t = 0$, somewhat as in the left-hand side of (37), and varying it at the fluid cell $x = 0$. The variation must be applied to the fluid solution $\mathcal{J}_i(x,t)$ for the average current under the initial condition with this inhomogeneous Lagrange parameter. Thus

$$\langle j_i(x,t) o(0,0) \rangle^{\text{hyd}} = \frac{\delta \mathcal{J}_i(x,t)}{\delta \beta^o(0,0)} = \sum_j \int \mathrm{d}y \, \frac{\delta q_j(y,t)}{\delta \beta^o(0,0)} \frac{\delta \mathcal{J}_i(x,t)}{\delta q_j(y,t)}, \tag{74}$$

where we used the fact that the fluid solution $\mathcal{J}_i(x,t)$ only depends on the average densities $q_j(y,t)$'s on the time slice $t$ (this is the hydrodynamic approximation), and we used the chain

rule. With the constitutive relations (53) and the relation $\delta q_j(y,t)/\delta\beta^o(0,0) = \langle q_i(y,t)o(0,0)\rangle^{\mathrm{hyd}}$, and specialising to the stationary, homogeneous background, one obtains (72).

At the Euler scale, without the diffusion operator, (72) is time-reversal invariant. However, at the diffusive scale, it is not. Physically, at time goes on, correlations should decay due to diffusion. Therefore, not only (72) is not time-reversal invariant, but also its validity depends on the sign of $t$. Indeed, for positive (negative) $t$, correlation functions should decay as $t$ gets more positive (negative). As written, (72) gives decaying correlation functions towards positive times, hence is valid for $t > 0$. For $t < 0$, the last term on the left-hand side should have a positive sign, instead of negative.

Equation (72) is a linear partial differential equation with constant coefficients. It can be solved by Fourier transform, and it is this language that clarifies the meaning of the hydrodynamic fluid-cell averaged correlation function $\langle\cdots\rangle^{\mathrm{hyd}}$. That is, define

$$S_{i,o}(k,t) = \int \mathrm{d}x\, e^{ikx} \langle q_i(x,t)o(0,0)\rangle^{\mathrm{c}} \tag{75}$$

as the Fourier transform of the true, microscopic connected correlation function. For $k$ small and $t$ large, this is also the Fourier transform of $\langle q_i(x,t)o(0,0)\rangle^{\mathrm{hyd}}$. Hence, accounting for the sign of $t$, then this satisfies

$$\partial_t S_{i,o}(k,t) - \mathrm{i}k \sum_j \mathsf{A}_i^{\ j} S_{j,o}(k,t) + \mathrm{sgn}(t)\frac{k^2}{2}\sum_j \mathcal{D}_i^{\ j} S_{j,o}(k,t) = 0. \tag{76}$$

In addition, realising that at $k = 0$ the time dependence disappears and we can use (15), we have

$$S_{i,o}(0,t) = -\frac{\partial\langle o(0,0)\rangle}{\partial\beta^i} = \mathsf{C}_{i,o}, \tag{77}$$

where the last equality is a definition of $\mathsf{C}_{i,o}$ Hence

$$S_{i,o}(k,t) = \sum_j \exp\left[\mathrm{i}kt\mathsf{A} - k^2|t|\mathcal{D}\right]_i^{\ j} C_{j,o}(k) \tag{78}$$

for some initial condition $C_{j,o}(k)$, which by (77) is constrained to

$$C_{j,o}(0) = \mathsf{C}_{j,o}. \tag{79}$$

Eq. (78) is an expression for the Fourier transform (75) of the exact connected correlation function (in a maximal entropy state), as an expansion in powers of $k$. The leading power corresponds to the Euler scale, and the first subleading, to the diffusive scale. The Euler scale is more precisely obtained by taking the limit $k \to 0$, $t \to \infty$ with $kt$ fixed. Since in this limit $C_{j,o}(k) = \mathsf{C}_{j,o}$ by (79), at the leading power, we can just use $C_{j,o}(0) = \mathsf{C}_{j,o}$ for the initial condition in (78). By choosing the gauge for $q_i(x,t)$ appropriately at the diffusive scale (see Remark 2.6), it is often possible to cancel the first subleading power of $k$ in $C_{j,o}(k)$. In such a gauge, one may simply use $C_{j,o}(k) = \mathsf{C}_{j,o}$ for the initial condition in (78) at both Euler and diffusive order. In particular, for correlation functions of densities $S_{i,j}(k,t) = S_{i,q_j}(k,t)$, this can be done by choosing a PT-symmetric gauge, and we obtain

$$S_{i,j}(k,t) = \left(\exp\left[\mathrm{i}kt\mathsf{A} - k^2|t|\mathcal{D}\right]\mathsf{C}\right)_{ij}. \tag{80}$$

As anticipated, the Fourier analysis clarifies the meaning of (72). It should be understood as an equation for the inverse Fourier transform, of the small-$k$ expansion, of the Fourier

transform of the true correlation function in a maximal entropy state. Taking the small-$k$ expansion between Fourier and inverse Fourier transforms, means that an averaging is made on the fluid cell at space-time point $x, t$. This, in particular, "washes out" any finite-frequency / wavelength oscillation that may appear in the true correlation function. At the Euler scale, the derivation (74) may use the explicit Euler scaling limit (37), where the scaling limit induces such an averaging. In particular, from this, one infers that, appropriately Euler-scale averaged $n$-point connected correlation functions decay, when all distances are scaled by $\lambda \to \infty$, like $\lambda^{1-n}$.

Finally, it is also possible to access correlation functions involving other operators than the densities. This is essentially done for the current observable in the derivation (74). The general form can be expressed by using **hydrodynamic projections**, extending the Drude weight analysis (35). At the Euler scale, in a self-explanatory notation, this takes the form (compare with (32))

$$\lim_{\substack{k\to 0,\, t\to\infty \\ kt \text{ fixed}}} S_{o_1,o_2}(k,t) = \lim_{\substack{k\to 0,\, t\to\infty \\ kt \text{ fixed}}} S_{\mathbb{P}o_1,\mathbb{P}o_2}(k,t)\,, \tag{81}$$

where I use the projection $\mathbb{P}$ on the space of conserved densities, defined in (33). Explicitly, this is

$$\lim_{\substack{k\to 0,\, t\to\infty \\ kt \text{ fixed}}} S_{o_1,o_2}(k,t) = \sum_{ijmn}(o_1,q_i)\mathsf{C}^{ij} S_{j,m}(k,t)\mathsf{C}^{mn}(q_n,o_2) = (o_1,\underline{q})\cdot \mathsf{C}^{-1}\exp\big[ik t\mathsf{A}\big](\underline{q},o_2)\,, \tag{82}$$

where the inner product (19) is used, and $S_{j,m}(k,t) = \big(\exp\big[ik\mathsf{A}t\big]\mathsf{C}\big)_{jm}$ is at the Euler scale. Note that at the Euler scale, everything depends on the product $kt$ only. The meaning of (82) is that the Euler-scale correlation functions between observables $o_1(x,t)$ and $o_2(0,0)$ is obtained by projecting $o_2$ onto the set of conserved quantities, propagating the associated conserved densities from $(0,0)$ to $(x,t)$, and overlapping the result of this propagation with $o_1$. Physically, we are saying that the leading, Euler-scale contribution to the correlation function is obtained by propagating linear waves ("sound waves") between the positions of the two observables, see Fig. 4. As an image, think about the strongest signal that a pond skater will "hear" from a leaf falling on the surface of the water nearby: it will be when the surface wave produced by the leaf reaches its feet. The formula can be written in real space, where the intuitive interpretation is even clearer,

$$\langle o_1(x,t)o_2(0,0)\rangle^{\text{eul}} = (o_1,\underline{q})\cdot \mathsf{C}^{-1}\delta\big[x-\mathsf{A}t\big](\underline{q},o_2). \tag{83}$$

Here the superscript now says that this is valid at the Euler scale, and $\delta\big[x-\mathsf{A}t\big]$ is the matrix $\mathsf{R}^{-1}\text{diag}\big[\delta(x-v_i^{\text{eff}}t)\big]\mathsf{R}$. The general expectation is that along the ballistic rays $x = v_i^{\text{eff}}t$, there are strong correlations, presumably algebraically decaying, while away from these rays, correlations are much weaker, exponentially decaying.

Equation (81) can be shown rigorously under natural conditions, and is essentially sufficient to derive (80) at the Euler scale. Hydrodynamic projections form a more powerful basis for a rigorous analysis of correlation functions than the physical arguments and formal derivation presented above.

**Remark 2.12.** *Fluid cell averaging is a delicate topic. Besides the Fourier transform procedure presented here, other ways are possible. At the Euler scale, one might integrate over cells in space that grow sublinearly with the scale; or one might take cells that grow simultaneously with the scale, and after the Euler scaling limit is taken, take the cells' lengths, in scaled unit, to zero. In fact, there is no guarantee that this be sufficient in order to extract the actual Euler physics: time-averaging may also be needed.*

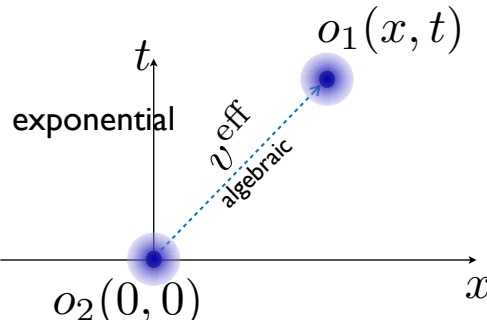

Figure 4: Correlations between observables at the Euler scale are due to propagation of ballistic waves between the two observables.

## Chapter 3  Integrable systems and thermodynamic Bethe ansatz technology

I would now like to apply the various concepts of hydrodynamics introduced in Chapter 2 to integrable systems.

The most basic characteristics of an integrable short-range many-body system is that it admits an *infinite number* of conserved quantities with local densities, of the type (11). Because of this infinity, the notion of maximal entropy state is more delicate – one must address the question as to if the series in the exponential in (14) converges. An answer is proposed in [121]. There, it is shown that one can define states by using state-flow equations similar to (15),

$$-\frac{\partial}{\partial s}\langle o\rangle_s = \int \mathrm{d}x\, \langle oq(x,0)\rangle_s^{\mathrm{c}}. \tag{84}$$

Here $s > 0$ is a parameter, which for definiteness, could be thought of as occuring from replacing $\beta^i$ by $s\beta^i$ in (14) (so that $\partial\langle o\rangle_s/\partial s = \sum_i \beta^i \partial\langle o\rangle_s/\partial\beta^i$ and $q = \sum_i \beta^i q_i$). The flow connects the desired state to the "infinite-temperature" state $s = 0$, which is the trace state (it exists and is unique in many, if not all, situations). The right-hand side of (84) is identified with the inner product (19). Therefore, the set of allowed charges $q$ are those which lie in the Hilbert space constructed "à la Gelfand-Naimark-Segal" from this inner product (moding out the null space, and completing by Cauchy sequences; this is not to be confused with the Hilbert space of the possible underlying quantum model!). It turns out, as shown in [121], that this space is in bijection with the space of linearly extensive conserved quantities, the so-called pseudolocal charges introduced by Prosen [142, 143] and that play a fundamental role in the thermalisation of integrable systems [144]. That is, $\sum_i \beta^i Q_i$ should in fact be identified with a single, Lagrange-parameter-dependent pseudolocal conserved charge. The $Q_i$s form a basis for the Hilbert space of pseudolocal charges, so $\sum_i \beta^i Q_i$ is just a basis decomposition; equivalently, the $q_i$s, occurring on the right-hand side of (15), form a basis for the Hilbert space constructed out of $(\cdot, \cdot)$. Because of the derivative in (15), this Hilbert space is identified with the tangent space to the manifold of maximal entropy states [the full mathematical construction of this manifold is not yet known, though].

In integrable systems, this Hilbert space is infinite-dimensional, hence the manifold of maximal entropy states is infinite-dimensional. One can think of the infinite-dimensionality as the "generalised" bit of generalised Gibbs ensembles (GGEs). The form (14), with an appropriate series of conserved charges, has been seen to arise from relaxation in quench protocols in a great many situations, hence it is quite well established, see the review [144][7].

---

[7]In integrable models there are two types of quasi-local, not strictly local conserved charges: those that can

This means that, in integrable systems, the manifold of maximal entropy states – the GGEs – is much larger than it is in conventional models. Nevertheless, it suggests that the large amount of conserved charges of integrable models *does not* entirely preclude ergodicity: it simply further reduces the manifold within which ergodicity may take place. In a classical mechanics picture, the action variables specify the hyper-torus in phase space on which the system's state is constrained, but ergodicity occurs on the angle variables, as the system covers the available hyper-torus.

The above is a rather formal description. How do we make this more precise and get actual formulae that are practically useful for integrable systems? In this chapter, I explain how the maximal entropy states can be efficiently described by the thermodynamic Bethe ansatz (TBA) technology. I will provide all elements of Section 2.2, and even some of Section 2.3, in the language of TBA. It is important to note that, despite the name, TBA is *not* constrained to Bethe-ansatz solvable quantum models. What I here refer to as the TBA technology is a general formalism, that applies to quantum and classical models alike, and that is based on an understanding of conserved quantities and maximal entropy states using the "asymptotic coordinates" of scattering theory.

## 3.1 Scattering map in integrable systems

The main tool for describing GGE measures is to use the *scattering map*. The idea is very simple. First, we choose a "vacuum" for our physical system. In a spin chain, a canonical choice is all spins up; in a field theory, the zero value of the field; in a gas, the absence of particles. But there are other choices, and the choice of vacuum affects the scattering theory of the model. Second, the system is excited on some region, say of length $L$. That is, the system exists on the infinite line, but, for instance, there are particles on the interval $[0, L]$ and nowhere else. For a finite density system, the total number of particles, quantity of energy or of other charge (with respect to the chosen vacuum) is chosen proportional to $L$, in the eventual large-$L$ limit. But before taking this limit, the question is to describe a distribution (quantum or classical) of configurations on this finite-density excited region. In order to do so, the third step is to simply *change coordinates*, from the canonical ones on the line, to asymptotic coordinates. The latter are obtained in a dynamical fashion, by *letting the system evolve for a very long time on the line*, all the way until the density is null and all emerging excitation units (particles, solitons etc.) are, in some way, very well separated[8]. The description of these excitation units – their velocities or momenta, and a characterisation of their positions ("impact parameters") – is what replace the description of the particles, or the field, or the spins on the interval $[0, L]$. This is the scattering map. See Fig. 5. Finally, we put a measure on the asymptotic coordinates themselves, which, by the scattering map, represents the GGE on the interval $[0, L]$, and we take the large-$L$ limit in order to describe the thermodynamic system. In this description, it was never necessary to fix any explicit boundary: we describe the thermodynamic of a finite-density gas, field theory of chain by using zero-density excitations on the line. Using zero-density objects to describe a finite-density state makes this a powerful technique.

The general idea is of course applicable for any system, integrable or not, but it is with integrability that its full power emerges. Structurally, the main properties of integrable systems is the presence of an extensive number of local conserved quantities. But their most important effects are indeed seen in the scattering theory, and this is what allows us to obtain explicit equations for their thermodynamics.

---

be obtained from local conserved charges by Cauchy completion, and those that are Cauchy completion of local observables which become conserved only in the limit. The latter, I believe, are the extra quasi-local conserved charges associated to strings in the thermodynamic Bethe ansatz language [118].

[8]This is more subtle if there are radiative excitation units, which are not well-delimited energy lumps, but the scattering theory for these can also be constructed.

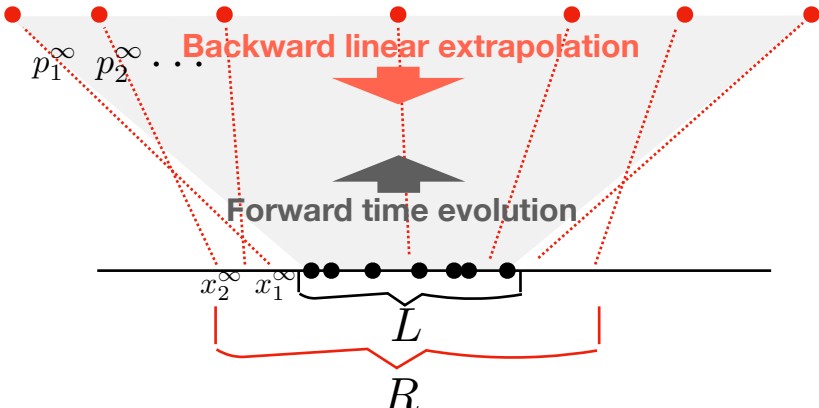

Figure 5: The scattering map. The physical particles (black dots) lie on an interval of length $L$. After a long-time evolution (shaded gray region), they are far apart (red dots). Their velocities $p_n^\infty$, and the impact parameters $x_n^\infty$ – the intercept of the zero-time slice with the asymptotic trajectories' extrapolations (red dotted line) – characterise the asymptotic configuration. The impact parameters lie on an interval of length $R$, generically different from $L$.

Let me illustrate these ideas, in this and the next section, using a simple, explicit example: the Toda model. This is a model of Galilean, labelled particles, with an exponential repulsive potential between particles of neighbouring labels (here with unit mass):

$$H = \sum_n \left( \frac{p_n^2}{2} + e^{-(x_{n+1}-x_n)} \right). \tag{85}$$

One can make the number of particles finite, for instance by requiring that $x_n = \infty$ $(-\infty)$ for $n > N$ $(n < 1)$. The momenta $p_n$ and positions $x_n$ are canonical coordinates, in particular $\{x_n, p_m\} = \delta_{n,m}$. They evolve according to the equations of motion induced by the Hamiltonian flow $H$. I will now describe the scattering map, which maps the particle's coordinates to a new set of coordinates that has a very simple dynamics.

For simplicity, consider the classical case. Because the potential is repulsive, any configuration of particles' positions and velocities, if let to evolve with $H$, will eventually end up with well-separated particles, $x_1 \ll x_2 \ll \ldots \ll x_N$, going away from each other, $p_{n+1} - p_n > 0$. Because they are so well separated, the potential will not be felt, hence the evolution will be free. Thus,

$$p_n(t) = p_n^\infty + O(t^{-\infty}), \quad x_n(t) = p_n^\infty t + x_n^\infty + O(t^{-\infty}). \tag{86}$$

The $O(t^{-\infty})$ corrections come from the exponentially small potential. One can see $p_n^\infty$ and $x_n^\infty$ as functions of the initial conditions, the sets $\{p_m = p_m(0)\}$ and $\{x_m = x_m(0)\}$. Thus, the asymptotic trajectories – determined by the asymptotic momenta $p_n^\infty$ and the "impact parameters" $x_n^\infty$ (see Fig. 5) – give rise to a change of coordinates

$$S : \{p_n\}, \{x_n\} \mapsto \{p_n^\infty\}, \{x_n^\infty\}. \tag{87}$$

This is the **scattering map**. This is the "out" scattering map, $S = S_{\text{out}}$, and there is also an "in" scattering map which maps to the asymptotic coordinates $\{p_n^{-\infty}\}, \{x_n^{-\infty}\}$ obtained by evolution towards negative infinite time. Note that the impact parameters are simply the positions where the linear extensions of the asymptotic trajectories cross the time slice $t = 0$.

The asymptotic coordinates are also dynamical variables, and possess a Poisson bracket induced by that on the original coordinates; indeed $S$ is just a change of coordinates. Because

time evolution is a canonical transformation, it is simple to see that the scattering map *also is a canonical transformation*, and thus

$$\{p_n^\infty, p_m^\infty\} = 0, \quad \{x_n^\infty, p_m^\infty\} = \delta_{nm}, \quad \{x_n^\infty, x_m^\infty\} = 0. \tag{88}$$

Using (86) in the time-evolved form of the Hamiltonian (85), we immediately find, because the positions at large times are separated by very large distances, that the Hamiltonian takes the *free-particle form* when written in terms of asymptotic coordinates,

$$H = \sum_n \frac{(p_n^\infty)^2}{2}. \tag{89}$$

As a consequence, time evolution by time $s$ of the asymptotic coordinates is trivial,

$$p_n^\infty(s) = p_n^\infty, \quad x_n^\infty(s) = p_n^\infty s + x_n^\infty. \tag{90}$$

The asymptotic coordinates are similar to the action-angle variables of integrable systems; however the scattering map exists in general, independently from integrability.

Let me now consider the other conserved charges of the model. Because they are supposed to be local, when particles are far from each other, they are not supposed to be affected by inter-particle interactions. Thus they can only be functions of the asymptotic momenta, that are indeed conserved. One finds that in general, the local conserved charges take the form

$$Q_i = \sum_n (p_n^\infty)^i, \quad i = 1, 2, 3, \ldots \tag{91}$$

[In fact, $i$ might lie in an unbounded subset of $\mathbb{N}$, the set of values of $i$ being a characteristic of the integrable model.] That is, they are obtained from *powers of the asymptotic momenta*. As explained above, we need to complete the set to the space of pseudolocal charges, and this is expected to lead to a parametrisation of conserved charges by a big space of functions $w(p)$ of the momenta,

$$Q_w = \sum_n w(p_n^\infty). \tag{92}$$

Of course, formally, all the charges (92) are well defined and conserved *even if the system is not integrable*. This is similar to the situation in quantum chains, where all projectors onto energy eigenstates are conserved quantities, independently of integrability. However, the main point of integrability is that the charges (92) are local, or pseudolocal – that is, they have appropriate locality properties. This is at the hearth of integrability, and will be used below in one, crucial argument, leading to the very simple structure of the scattering map of integrable systems.

The final ingredient necessary to describe the thermodynamics is the scattering shift. In order to understand this, consider a full scattering process. We have a set of particles which, at large negative times, have free-particle trajectories whose linear extrapolation to the time slice $t = 0$ covers some region of space. This region is assumed to have a length $R$ (see Fig. 5) proportional to the number of particles $N$, so that we are describing a gas of finite density. At large positive times, the particles separate again because of the repulsive potential, and end up in outgoing free-particle trajectories. The scattering question is concerned with determining the outgoing asymptotic coordinates as functions of the ingoing asymptotic coordinates: $S_{\text{out}} \circ S_{\text{in}}^{-1}$.

In integrable models, this question can be answered very explicitly. The clearest argument is from Parke [145] – it was made in the context of quantum field theory, but can be applied more generally, as discussed in [146]. Consider a single *non-trivial* conserved charge, say $Q_3$

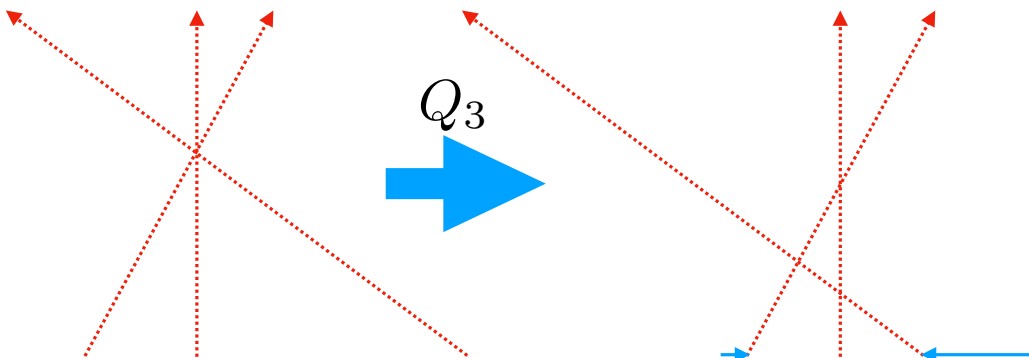

Figure 6: The $Q_3$ flow in integrable models. Because the displacement is proportional to the square of the momentum, a three-body process is factorised into well separated two-body processes.

in (91). Assume that it has appropriate locality property – what this mean will be clear below. Consider its associated Hamiltonian flow $e^{t_3\{\cdot,Q_3\}}$. Let us apply this flow for a "time" $t_3$ on the particles' coordinates at a very negative real-time slice $t \ll 0$, and then for a "time" $-t_3$ at a very positive real-time slice $t \gg 0$. Because the $Q_3$ flow commutes with the $H$ flow, the result does not affect the scattering problem. In math,

$$S_{\text{out}} \circ S_{\text{in}}^{-1} = S_{\text{out}} \circ e^{-t_3\{\cdot,Q_3\}} \circ e^{t_3\{\cdot,Q_3\}} \circ S_{\text{in}}^{-1} = e^{-t_3\{\cdot,Q_3\}} \circ S_{\text{out}} \circ S_{\text{in}}^{-1} \circ e^{t_3\{\cdot,Q_3\}}, \quad (93)$$

where we use commutation of the $Q_3$ flow with the scattering map. Now, *because the charge is local*, and because at large negative (positive) times all trajectories of particles are well separated, one can simply apply the $Q_3$ flow independently on each in (out) asymptotic coordinates themselves. This is trivial, for instance

$$e^{-t_3\{\cdot,Q_3\}}(p_n^\infty) = p_n^\infty, \quad e^{-t_3\{\cdot,Q_3\}}(x_n^\infty) = x_n^\infty - 3t_3(p_n^\infty)^2. \quad (94)$$

The important point it that this is a constant shift by the *square* of the momentum. This is different from real time evolution, where the shift is linear with momentum. As a consequence, for $t_3$ large enough, the $Q_3$-modified scattering process is composed of *well-separated two-body processes*. See Fig. 6.

The above has important consequences. Since in 1+1 dimensions, two-body processes preserve momenta, and since the $Q_3$ flows above don't change the asymptotic momenta, we conclude that *scattering is elastic*: $\{p_n^{\text{in}}\} = \{p_n^{\text{out}}\}$. This in turn implies that the shift due to the $Q_3$ flow on the in-particle of momentum $p$, is equal and opposite to that on the out-particle of the same momentum. Thus, we may evaluate the effect of scattering on the impact parameters simply by adding all **two-body shifts** $\varphi(p_n - p_m)$ (see Fig. 7) incurred as trajectories of momenta $p_n$ and $p_m$ cross (we use Galilean invariance to say it is a function of the difference only) – this is factorised scattering. Here and throughout these notes, we assume the symmetry $\varphi(p) = \varphi(-p)$, which holds in many models.

Let me introduce the concept of **quasiparticle**: this is a "tracer" attached to real particles, but which may jump from particle to particle at collisions, in such a way that it follows a given momentum. The concept of quasiparticle only makes sense with elastic, factorised scattering as above, because we need to be able to trace a given momentum. Labelling quasiparticles using the in-particle labels from left to right, and using $p_n^{\pm\infty}$ and $x_n^{\pm\infty}$ to represent the quasiparticles' asymptotic coordinates, we therefore have $p_n^\infty = p_n^{-\infty}$, the ordering $p_n^{-\infty} > p_m^{-\infty}$ for $n < m$, and

$$x_n^\infty = x_n^{-\infty} - \sum_{m>n} \varphi(p_n^{-\infty} - p_m^{-\infty}) + \sum_{m<n} \varphi(p_n^{-\infty} - p_m^{-\infty}). \quad (95)$$

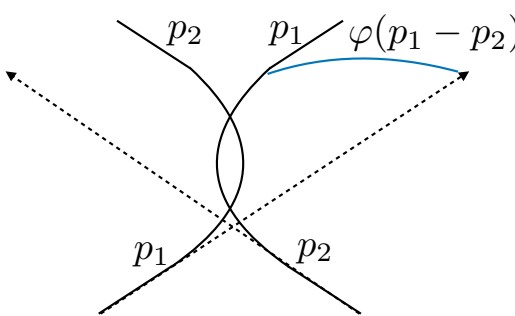

Figure 7: The two-body scattering shift.

It turns out that two-body shifts can also be used to describe the individual in and out scattering maps themselves, up to small corrections that do not affect the thermodynamics. A quasiparticle at position $x_n$ will incur shifts, as it evolves towards positive time, due to quasiparticles on its right that have smaller momenta, and quasiparticles on its left that have higher momenta.

$$x_n^\infty \approx x_n + \Big( \sum_{m<n,\, x_m<x_n} - \sum_{m>n,\, x_m>x_n} \Big) \varphi(p_n^{-\infty} - p_m^{-\infty}). \qquad (96)$$

Similarly,

$$x_n^{-\infty} \approx x_n + \Big( \sum_{m>n,\, x_m<x_n} - \sum_{m<n,\, x_m>x_n} \Big) \varphi(p_n^{-\infty} - p_m^{-\infty}). \qquad (97)$$

In quantum systems, scattering shifts are replaced by scattering phases. The two-body scattering amplitude is a unit-modulus complex number $\mathfrak{S}(p)$, and one defines $\varphi(p) = -\mathrm{i}\,\mathrm{d}\log\mathfrak{S}(p)/\mathrm{d}p$. Integrability implies that the many-body amplitude factorises as a product of two-body amplitudes, and scattering is again elastic, preserving all momenta. The Bethe ansatz wave function can be seen as an explicit description of this factorised scattering.

Asymptotic states, both in the classical and quantum cases, can be much more complicated than the simple freely evolving Toda particles. There can be many quasiparticle types, with internal numbers that can be exchanged under scattering. In chains and field theory, asymptotic states are often described by solitons and radiation modes. In all cases I know, however, the general TBA structure described in the next section holds.

The notion of quasiparticles, and the scattering shift equation (96), are the two important results of this section, and lead to the TBA description of the thermodynamics.

**Remark 3.1.** *The Toda model can be presented in two natural ways. The first is that presented above: it is a gas of particles, interacting via a pairwise potential that depends on their position. Of course, the potential is perhaps a bit "strange", as it only makes particles of* neighbouring labels *interact with each other. Another way is that under which the model is seen as a* chain. *In this way, the labels are the discrete positions were the degrees of freedom lie, say horizontally on the subset $\mathbb{Z}$ of the line, and the local degrees of freedom are particles with their momentum and position, moving, as it were, vertically. In this way of presenting, the interaction is between nearest neighbour on the chain.*

*One difference between these two ways is the notion of space. In a the first, it is the values that $x_n$ take, while in the second, it is the label n itself. In a thermodynamic state, of fixed density, the two notions of space, on large scales, are in proportion to each other. But in general, especially in the hydrodynamics, the difference is more important (see [146]).*

*In fact, another difference, which brings an important point discussed above, is as to the notion of* vacuum. *In the chain viewpoint, if one adds the conserved charge $Q_0 = \sum_n (x_{n+1} - x_n)$ to the*

*Hamiltonian and one considers the infinite chain, the "vacuum" can be seen as the configuration where all vertical positions minimise the interaction potential. Excitations on top of this vacuum are waves propagating along the chain. These excitations are known to be solitons and radiative waves [147, 148]. Thus, in this viewpoint, the most natural set of asymptotic states are solitons and radiation modes. Their two-body scattering shift can be evaluated once the two-body exact solution is known; although this is not as elementary a calculation as in the gas viewpoint. The set of quasiparticles – in the chain, the solitons and radiation modes, and in the gas, the Toda particles themselves – depends on the viewpoint. This illustrates the fact that the choice of vacuum influences the set of asymptotic particles, and thus quasiparticles, of the scattering theory. Different choices lead to different ways of presenting the thermodynamics, but, in principle, are equivalent, possibly, as in the Toda example, up to a redefinition of space.*

**Remark 3.2.** *The asymptotics of the scattering shift at large momentum gives information about the local properties of the gas or field theory. For instance, I note that for the hard rod gas, see Example 3.4 below, the scattering shift is a negative constant, equal to the rods' length a. From the viewpoint of a quantum scattering theory, this corresponds to $\mathfrak{S}(p) = e^{-aip}$. Such a scattering phase is known not to have a well-defined local energy density, which we can interpret by the fact that this is a theory where particles have a finite extent (being otherwise free). Likewise, it is known that certain perturbations of relativistic quantum integrable models, referred to as "T-Tbar" perturbations, which are not UV finite, give rise to an additional factor to the two-body scattering of the form $\mathfrak{S}(\theta_1 - \theta_2) = e^{-aip(\theta_1 - \theta_2)}$ where $\theta_i$ are rapidities and $p(\theta)$ is the momentum. Thus this affects the asymptotic of the scattering phase, and we can interpret again the lack of UV-finiteness by the conjecture that T-Tbar perturbations give a nonzero length to the fundamental particles of the original theory.*

**Example 3.3.** *In the Toda model, the Lax matrix was constructed by Flashka [149]. It is the matrix with structure*

$$L = \begin{pmatrix} \ddots & \vdots & & & \vdots & \\ \cdots & b_{-1} & a_{-1} & 0 & 0 & \cdots \\ & a_{-1} & b_0 & a_0 & 0 & \\ & 0 & a_0 & b_1 & a_1 & \\ \cdots & 0 & 0 & a_1 & b_2 & \cdots \\ & \vdots & & & \vdots & \ddots \end{pmatrix}, \tag{98}$$

*where $a_n = e^{-(x_{n+1} - x_n)/2}$ and $b_m = p_m$. Because this is part of a Lax pair, its traces $\mathrm{Tr}L^k$ are time-independent. In fact, they give rise to the local conserved charges of the Toda model, with in particular $\frac{1}{2}\mathrm{Tr}L^2 = H$. Thus here we have access to all conserved charges, and it is easy to verify that (91) holds. As the spectrum is time invariant, we can evaluate it simply by evolving L for a long time (that is, applying the scattering map). Clearly, at long times $a_n \to 0$ and $b_n \to p_n^\infty$. Thus L becomes diagonal, and the spectrum of the Lax matrix is the set of asymptotic momenta, which are indeed conserved quantities. The associated current observables can also be constructed explicitly [150]. But more importantly for our construction, the crucial ingredient, the scattering shift, is an elementary two-body calculation of classical mechanics, and gives*

$$\varphi(p - p') = 2\log|p - p'|. \tag{99}$$

**Example 3.4.** *The gas of hard rods is a gas composed of elongated particles (rods), all of length say $a > 0$, which move freely except for elastic collisions (we assume they all have unit mass). See Fig. 8. The set of momenta is therefore conserved throughout the evolution, and for our purpose we may assume that all momenta are different. Here the concept of quasiparticles is very simple: the quasiparticle for momentum p can be taken as the rod which has momentum p – that is, it is the velocity tracer. It simply jump from one rod to the other upon elastic collisions. Its exact*

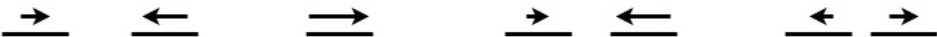

Figure 8: The gas of hard rods.

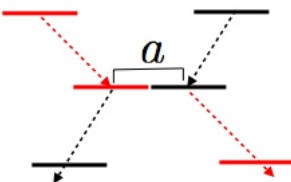

Figure 9: Upon collision, velocities are exchanged. We can think of a quasiparticle as the tracer of a given velocity, which jumps by a distance $a$ upon collisions.

*position can also be taken simply as the centre of the rod that is tracing the given velocity. With this picture, a quasiparticle is affected by a jump forward of distance a upon collision with another quasiparticle. This is the scattering shift, and we conclude that*

$$\varphi(p - p') = -a \tag{100}$$

*(notice the negative sign associated to a forward jump, in our convention). See Fig. 9*

**Example 3.5.** *The quantum Lieb-Liniger model is perhaps the most important model for applications of GHD, as it gives a good description of cold atomic gases seen in many experiments [39]. It is a model for N bosons (where N is taken to infinity in the thermodymic limit), with Hamiltonian*

$$H = -\sum_{n=1}^{N} \frac{1}{2} \partial_{x_n}^2 + g \sum_{n<m} \delta(x_n - x_m), \tag{101}$$

*where g > 0 is the repulsion strength between the bosons. It is Bethe-ansatz integrable, and the study of its thermodynamics led to one of the first formulation of TBA [38, 69]. The differential scattering phase can also be calculated by the elementary construction of the exact two-particle wave function for this Hamiltonian (no need for Bethe ansatz!), giving*

$$\varphi(p - p') = \frac{2g}{(p - p')^2 + g^2}. \tag{102}$$

## 3.2 Quasiparticle description of thermodynamic densities

I now develop the thermodynamics of integrable systems using the scattering formalism. As explained in Chapter 2, this eventually allows us to obtain the Euler hydrodynamics. There is one important remark however: thermodynamics gives us, in a sense, a bit more, which is not required for the Euler hydrodynamic equations, as it gives access to the large-scale fluctuations. Indeed, for instance, the thermodynamic entropy is a large-deviation function for fluctuations of total conserved quantities in thermodynamic systems. The Euler hydrodynamic equations do not necessitate the knowledge of any such fluctuation "spectrum" (although, of course, Euler-scale correlation functions do). Below this will translate into the fact that the statistics of quasiparticles is required for the thermodynamics, but not the Euler hydrodynamics equations. In this sense, it would be possible to go more directly to the Euler hydrodynamic equations, and one way is via the change of metric induced by the scattering map, as explained in Section 4.3. Nevertheless, any complete discussion necessitates the thermodynamics, hence I will follow this route here.

Let me consider again the example of the Toda model. Its partition function in a system of length $L$, for the state (14), is

$$Z = \sum_N \frac{1}{(2\pi)^N N!} \int_{x_n \in [0,L]} \prod_{n=1}^N dx_n \prod_{n=1}^N dp_n \, e^{-\sum_i \beta^i Q_i} = \sum_N \frac{1}{(2\pi)^N N!} Z_{\text{cano}}(N,L), \qquad (103)$$

where the sum is over all possible number of particles $N \in \mathbb{N}$. This is the "grand canonical ensemble", and $Z_{\text{cano}}(N,L)$ is the (or rather, a generalised version of the) partition function for the canonical ensemble. The factor $1/(2\pi)^N$ is introduced for consistency with quantum calculations, where the phase-space cell is divided by $h$, and we take $\hbar = 1$. The change to asymptotic coordinates preserves the measure, as it is canonical. Using (91), but for the in-coordinates $p^{-\infty} = p^{\text{in}}$ and $x^{-\infty} = x^{\text{in}}$ for convenience, and in fact re-writing $\sum_i \beta^i Q_i = Q_w$ with (92), we then obtain

$$Z_{\text{cano}} = \int_{x_n \in [0,L]} \prod_n dx_n^{\text{in}} \prod_n dp_n^{\text{in}} \, e^{-\sum_n w(p_n^{\text{in}})} = \int \prod_n dp_n^{\text{in}} \, e^{-\sum_n w(p_n^{\text{in}})} \int_{x_n \in [0,L]} \prod_n dx_n^{\text{in}}. \quad (104)$$

I emphasise that the function $w(p)$ *does not need to have a Taylor series expansion*, or even to be continuous, as $Q_w$ lies in a certain completion of the space of conserved quantities. What functions $w(p)$ we can put here is a tricky question, but certainly $w(p)$ should grow at large $|p|$ fast enough to make the multiple integral convergent.

Note how, because the GGE measure is only a function of the asymptotic momenta, I have extracted in (104) the integral over impact parameters. What we see is that the partition function takes the form of that of free particles, up to the volume contribution, which I denote $R^N$:

$$R^N = \int_{x_n \in [0,L]} \prod_n dx_n^{\text{in}}. \qquad (105)$$

This volume contribution is nontrivial: the relation between $x_n^{\text{in}} = x_n^{-\infty}$ and $x_n$ is given by (97), and thus momentum dependent. It can be evaluated in the limit of large $N$ and $L$ (at fixed ratio). Consider for instance quasiparticle 1. The full length $R$ it sees in "asymptotic space" is that obtained by scanning from left to right its trajectory, in such a way that its position on time-slice $t = 0$ scans from 0 to $L$. Scanning from left to right, jumps are incurred, and thus the length in asymptotic space is different as that at time 0, a "change of metric", see Fig. 10. At its furthest point on the left, it does not incur any shift, as $p_1^{\text{in}} > p_n^{\text{in}}$ for all $n > 1$: its trajectory is straight from the asymptotic region to time 0. By contrast, when the impact parameter is such that the trajectory reaches at time 0 the furthest point on the right, it must have crossed all other quasiparticles' trajectories exactly once. Therefore, the full length $R$ seen in asymptotic space (the length covered by the impact parameter such that the time-0 position covers the interval $[0, L]$, see Figs. 5 and 10) is

$$R = L + \sum_{m \neq 1} \varphi(p_1^{\text{in}} - p_m^{\text{in}}). \qquad (106)$$

Repeating for all quasiparticles, we obtain

$$R^N = \prod_n \left( L + \sum_{m \neq n} \varphi(p_n^{\text{in}} - p_m^{\text{in}}) \right) = L^N \prod_n \left( 1 + \sum_{m \neq n} \frac{\varphi(p_n^{\text{in}} - p_m^{\text{in}})}{L} \right). \qquad (107)$$

One can then combine (104), (105) and (107), and what remains to do is a large-deviation analysis, $N, L \to \infty$ with density $N/L$ fixed, and then the Legendre transformation to get (103). In the large-deviation analysis, a certain distribution of the values of the integration

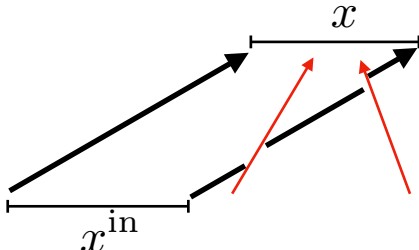

Figure 10: The change of metric due to interactions in integrable models.

momenta $p_n^{\text{in}}$ will give the dominant contribution to the integral. Thus, the integrations over momenta are replaced by the minimisation of an appropriate functional with respect to a distribution of momenta. This analysis is beyond the scope of these notes, but see [146]. Here I simply express the final result.

Before doing so, let me note two things. First, the analysis produces a distribution of momenta, a function I will denote $\rho_{\text{p}}(p)$. By definition, it is such that

$$\frac{1}{L}\sum_n f(p_n^{\text{in}}) \sim \int \mathrm{d}p \, \rho_{\text{p}}(p) f(p) \quad \forall f \text{ smooth enough} \tag{108}$$

in the limit of large $L$ and $N$. It is interpreted as a density of quasiparticles per unit momentum and per unit volume, thus the subset p (for "particle"). The average of any conserved density can be expressed using this, for instance

$$\mathsf{q}_i = \frac{1}{L}\sum_n \left(p_n^{\text{in}}\right)^i \sim \int \mathrm{d}p \, \rho_{\text{p}}(p) p^i. \tag{109}$$

Second, using this density, the quantity $\left(1 + \sum_{m\neq 1} \frac{\varphi(p_1^{\text{in}} - p_m^{\text{in}})}{L}\right)$ from (106) becomes

$$2\pi\rho_{\text{s}}(p_1^{\text{in}}) = 1 + \int \mathrm{d}p \, \rho_{\text{p}}(p)\varphi(p_1^{\text{in}} - p); \tag{110}$$

this defines the function $\rho_{\text{s}}(p)$, and the factor $2\pi$ is by convention. According to the above argument, the quantity (110) is $2\pi\rho_{\text{s}} = R/L$, the ratio of asymptotic-space lengths to zero-time-slice lengths. This takes into account the "perceived" change of length due to the scattering shifts, and can be interpreted as *an effective available space density* in which quasiparticles roam. Thus, really, as mentioned, this is like a metric – an interpretation that will play an important role in Section 4.3. The subscript s is for "space" density.

I will now state the result of the large-deviation analysis by expressing the free energy density $\mathsf{f} = -\lim_{N,L\to\infty} L^{-1}\log Z$; from this, the above quasiparticle and space densities will be obtained. The free energy density takes the form

$$\mathsf{f} = \int \frac{\mathrm{d}p}{2\pi} F(\epsilon(p)). \tag{111}$$

Here the **free energy function** $F(\epsilon)$ encodes the almost-free nature of the form of $Z$, and is the free energy contribution of a free particle of energy $\epsilon$, given by $F(\epsilon) = -\mathrm{e}^{-\epsilon}$. The function $\epsilon(p)$ is not, however, the "bare" contribution $w(p)$ of quasiparticle $p$ to the partition function. Instead, it receives contributions due to the change of volume in asymptotic space. It solves the nonlinear integral equation

$$\epsilon(p) = w(p) + \int \frac{\mathrm{d}p'}{2\pi} \varphi(p - p') F(\epsilon(p')). \tag{112}$$

The function $w(p)$ in this equation is usually referred to as the *source term*.

This gives us in principle the full information about all average conserved densities $q_i$, obtained by differentiation of the free energy density, (18). Suppose the amount of charge $Q_i$ carried by quasiparticle $i$ is the function $h_i(p)$; in (91) and (109), for instance, this is chosen to be $h_i(p) = p^i$, but one may choose a different basis of functions. This means

$$w(p) = \sum_i \beta^i h_i(p). \tag{113}$$

Differentiation then gives

$$q_i = \int \frac{dp}{2\pi} n(p) h_i^{dr}(p), \tag{114}$$

where we define the **occupation function**

$$n(p) = \left. \frac{dF(\epsilon)}{d\epsilon} \right|_{\epsilon = \epsilon(p)}. \tag{115}$$

The factor which we denoted $h_i^{dr}(p)$ is simply $h_i^{dr}(p) = \partial \epsilon(p) / \partial \beta^i$. Applying the derivative to (112), we find that this satisfies the *linear* integral equation

$$h_i^{dr}(p) = h_i(p) + \int \frac{dp'}{2\pi} \varphi(p - p') n(p') h_i^{dr}(p'). \tag{116}$$

For any function of momentum, we define the **dressing operation** $h \mapsto h^{dr}$ by the solution to (116) for $h_i$ replaced by $h$.

In order to go further, let me show the following *symmetry formula*:

$$\int dp\, g(p) n(p) h^{dr}(p) = \int dp\, g^{dr}(p) n(p) h(p). \tag{117}$$

The most straightforward way of showing this is by introducing the integral-operator notation. We write

$$(Th)(p) = \int \frac{dp'}{2\pi} \varphi(p - p') h(p') \tag{118}$$

and write $n$ for the diagonal integral operator that is multiplication by the function $n(p)$. Usually, by convention, one also introduces the $T(p - p')$ kernel,

$$T(p - p') = \frac{1}{2\pi} \varphi(p - p'). \tag{119}$$

Then,

$$h^{dr} = h + Tnh^{dr} \Rightarrow h^{dr} = (1 - Tn)^{-1} h. \tag{120}$$

With the dot-product $h \cdot g = \int dp\, h(p) g(p)$, we find

$$\int dp\, g(p) n(p) h^{dr}(p) = g \cdot n(1 - Tn)^{-1} h = (1 - nT)^{-1} ng \cdot h = n(1 - Tn)^{-1} g \cdot h \tag{121}$$

and the last expression is exactly the right-hand side of (117) (here we used the symmetry of $T$).

Using this symmetry formula, we obtain, from (114),

$$q_i = \int \frac{dp}{2\pi} 1^{dr}(p) n(p) h_i(p), \tag{122}$$

where $1(p) = 1$ is the constant unit function. This is of the form (109), that is

$$\mathsf{q}_i = \int \mathrm{d}p\, \rho_{\mathrm{p}}(p) h_i(p), \tag{123}$$

which allows us to identify

$$\rho_{\mathrm{p}}(p) = \frac{1}{2\pi} 1^{\mathrm{dr}}(p) n(p) \tag{124}$$

as the **density of quasiparticles**. Note that by (116)

$$1^{\mathrm{dr}}(p) = 1 + \int \frac{\mathrm{d}p'}{2\pi} \varphi(p - p') n(p') 1^{\mathrm{dr}}(p') \tag{125}$$

and inserting (124) on the right-hand side, we compare with the definition (110) and obtain

$$\rho_{\mathrm{s}}(p) = \frac{1}{2\pi} 1^{\mathrm{dr}}(p). \tag{126}$$

As argued above, this is interpreted as the **density of space**. In particular, the occupation function takes the form

$$n(p) = \frac{\rho_{\mathrm{p}}(p)}{\rho_{\mathrm{s}}(p)}, \tag{127}$$

the ratio of the density of quasiparticles to the available space density – it is indeed, then, an "occupation" ratio.

As a technical note, the symmetry formula (117) can be used to obtain a different form of the dressing operation. Applying it to the second term on the right-hand side of (116) gives

$$h^{\mathrm{dr}} = (1 + T^{\mathrm{dr}} n) h, \tag{128}$$

where $T^{\mathrm{dr}}$ is the integral operator whose kernel,

$$T^{\mathrm{dr}}(p, p') = \left[(1 - Tn)^{-1} T(\cdot - p')\right](p), \tag{129}$$

is the **dressed scattering kernel**, the dressing of $T(p - p')$ as a function of its first argument $p$. [Note that even if $T(p - p')$ is a function the difference of momenta, such as in the Galilean model here considered the result, in general, is no longer a function of $p - p'$: the state is not necessarily Galilean invariant.] That is, once $T^{\mathrm{dr}}(p, p')$ is evaluated, the dressing of any function can be evaluated simply by integrating – no need to solve an integral equation. Note that if $T$ is symmetric, then $T^{\mathrm{dr}}(p, p') = T^{\mathrm{dr}}(p', p)$, as $(T^{\mathrm{dr}})^{\mathrm{T}} = \left[(1 - Tn)^{-1} T\right]^{\mathrm{T}} = T(1 - nT)^{-1} = (1 - Tn)^{-1} T = T^{\mathrm{dr}}$, a relation which we have used in order to obtain (128).

Finally, in order to complete the description of the thermodynamics of conserved densities, let me look at the entropy density, defined in (47). It can be calculated explicitly in terms of the above objects:

$$\begin{aligned}
\mathsf{s} &= \sum_i \beta^i \mathsf{q}_i - \mathsf{f} \\
&= \int \mathrm{d}p \left[ \sum_i \beta^i h_i(p) \rho_{\mathrm{p}}(p) - \frac{1}{2\pi} F(\epsilon(p)) \right] \\
&= \int \mathrm{d}p \left[ w(p) \rho_{\mathrm{p}}(p) - \frac{1}{2\pi} F(\epsilon(p)) \right] \\
&= \int \mathrm{d}p \left[ \epsilon(p) \rho_{\mathrm{p}}(p) - \frac{1}{2\pi} \left( \int \mathrm{d}q\, \rho_{\mathrm{p}}(p) \varphi(p - q) F(\epsilon(q)) + F(\epsilon(p)) \right) \right] \\
&= \int \mathrm{d}p\, \epsilon(p) \rho_{\mathrm{p}}(p) - \frac{1}{2\pi} \left( \int \mathrm{d}q\, (2\pi \rho_{\mathrm{s}}(q) - 1) F(\epsilon(q)) + \int \mathrm{d}p\, F(\epsilon(p)) \right) \\
&= \int \mathrm{d}p\, \rho_{\mathrm{s}}(p) \left[ \epsilon(p) n(p) - F(\epsilon(p)) \right]. \tag{130}
\end{aligned}$$

Using $F(\epsilon) = -e^{-\epsilon}$, this becomes $\mathsf{s} = \int \mathrm{d}p\, \rho_s(\epsilon + 1)e^{-\epsilon} = -\int \mathrm{d}p\, \rho_s n(\log(n) - 1)$, which we recognise as the usual form the entropy density takes in a system of free classical particles.

## 3.3 Ingredients of the TBA technology, classical and quantum

The formulation given in Section 3.2 forms the basic ingredients of the "TBA technology", which I complete in the following section of this Chapter. Although the example of the Toda model was taken, which is a Galilean invariant classical model of interacting particles, the formulation is directly adaptable to other models, quantum and classical, Galilean and relativistic (or with any other dispersion relation), field theories or chains.

In fact, as far as I am aware, the TBA was actually first obtained in quantum models, in the Bose gas with $\delta$-function repulsion potential, by Yang and Yang [69]. See [70, 71] for the derivations in quantum chains and quantum field theory. In these cases, $\rho_s$ is usually interpreted as a **density of states** – the density of available states that can be filled by quasiparticles (thus the subscript s still makes sense). The Yang-Yang thermodynamic equations were verified experimentally in cold atom gases [151].

Classical limits of quantum results can be taken in order to get the classical TBA, see [152], and in particular [153–156] for the Toda model. Independent, purely classical calculations are also available, such as the one outlined in Section 3.2 based on scattering of classical particles; the picture is suggested in [154] and worked out in [146]. More involved derivations using the classical inverse scattering method were also done or proposed for various classical models [157, 158, 158, 159].

Generalised hydrodynamics based on the TBA, discussed in Chapter 4, was first introduced for quantum field theories [1] and quantum chains [2], and later shown [160] to agree with equations found before for the classical hard rod gas [77] and soliton gases [73–76], and to reproduce the dynamics of classical field theories [141]. It was also verified experimentally in cold atom gases [80].

In any case, from observation of the results in quantum and classical cases, it appears as though the TBA technology is extremely general. I observe that all numbered equations from (110) to (130) hold in generality, with the following ingredients:

**I.** The **spectral space**. This is part of the specification of the model. If there are many quasiparticle types, then one must augment the momentum $p$ with an extra index, $(p, \ell)$, labelling the types, and replace $\int \mathrm{d}p \to \sum_\ell \int \mathrm{d}p$. For instance, these may be soliton types in field theory or chains. In some cases, it is also possible that the momenta may only take values within a finite region (Brillouin zone). In general, one must determine the full set of parameters that parametrise the asymptotic excitations (which, in integrable systems, become quasiparticles) of scattering theory.

**II.** The scattering shift $\varphi(p - p')$. This is again part of the model specifications. It can be evaluated simply by looking at the two-body scattering of asymptotic excitations. It may in general be a function not of the momentum difference, but of two independent momenta, $\varphi(p, p')$; and it might not be symmetric, in which case some of the equations above have to be adjusted. With an appropriate re-parametrisation, for instance $p(\theta) = m \sinh \theta$ with the rapidity $\theta$ in relativistic models, it may simplify again to a function of the difference, $\varphi(\theta - \theta')$. There is a theory of reparametrisation of the spectral space. See [123].

**III.** The free energy function $F(\epsilon)$. This is also part of the model specifications. It describes

the *statistics* of the quasiparticles involved. For instance:

Classical particles:
$$F(\epsilon) = -e^{-\epsilon}, \; n = e^{-\epsilon}$$
$$s = -\int dp \, \rho_s n (\log n - 1). \tag{131}$$

Classical radiation:
$$F(\epsilon) = \log \epsilon, \; n = 1/\epsilon$$
$$s = \int dp \, \rho_s (\log n + 1). \tag{132}$$

Quantum fermions:
$$F(\epsilon) = -\log(1 + e^{-\epsilon}), \; n = (1 + e^\epsilon)^{-1}$$
$$s = -\int dp \, \rho_s [n \log n + (1-n) \log(1-n)]. \tag{133}$$

Quantum bosons:
$$F(\epsilon) = \log(1 - e^{-\epsilon}), \; n = (e^\epsilon - 1)^{-1}$$
$$s = -\int dp \, \rho_s [n \log n - (1+n) \log(1+n)]. \tag{134}$$

**IV.** The functions $h_i(p)$. These specify the conserved quantities $Q_i$. In the classical case, $h_i(p)$ is obtained from evaluating $Q_i$ as a function of asymptotic momenta, as $Q_i = \sum_n h_i(p_n^\infty)$. In the quantum case, it is the one-particle eigenvalue of the operator $Q_i$; on a $N$-particle state, this operator acts as $Q_i | p_1, \ldots, p_N \rangle = \sum_n h_i(p_n) | p_1, \ldots, p_N \rangle$.

**V.** The function $w(p)$. This specifies the state (14). Again, in the classical case, it is the function obtained when evaluating $\sum_i \beta^i Q_i$ as a function of asymptotic momenta, and in the quantum case, it is the one-particle eigenvalue of the operator $\sum_i \beta^i Q_i$. Clearly, $w(p) = \sum_i \beta^i h_i(p)$.

It is also worth noting that we can now take a variety of different "coordinate systems" for the maximal entropy state. Recall that the set of Lagrange parameters $\{\beta^i\}$ is such a system of coordinates. As mentioned above, because of the relation (113), once we've fixed our choice of basis charges $Q_i$, hence of basis functions $h_i$, then the Lagrange parameters fix $w(p)$. Thus, $w(p)$ specifies the state. Once we know $w(p)$, we may evaluate $\epsilon(p)$ by solving (112). The inverse is also immediate: knowing $\epsilon(p)$, we obtain $w(p)$ easily. But once $\epsilon(p)$ is known, then so is the occupation function $n(p)$, via (115) (and the model-given form of the free-energy function $F(\epsilon)$). Then we know $\rho_p(p)$ by (124), and this also can be easily inverted: knowing $\rho_p(p)$ we obtain $n(p)$ by evaluating $\rho_s(p)$ by (110) and then calculating the right-hand side of (127). Finally, all $q_i$ are then fixed by, for instance, (123) [the inversion problem, getting $\rho_p(p)$ from the $q_i$s, requires $h_i(p)$ to be complete in an appropriate sense, see the discussion in Remark 3.11]. Thus, there are various bijections between various systems of coordinates:

$$\{\beta^i\} \overset{(113)}{\longleftrightarrow} w(p) \overset{(112)}{\longleftrightarrow} \epsilon(p) \overset{(115)}{\longleftrightarrow} n(p) \overset{(124),(110),(127)}{\longleftrightarrow} \rho_p(p) \overset{(123)}{\longleftrightarrow} \{q_i\}. \tag{135}$$

**Example 3.6.** *All models with quadratic Hamiltonians – free classical and quantum particles and radiations – can be worked out explicitly, and one can check that the above formulation works, with $\varphi(p) = 0$. More explicitly, let me consider various single-specie free models (spectral*

space $\mathbb{R}$), of various statistics (see Point III above) for free classical particles the partition function $Z \sim \mathrm{e}^{-L\mathrm{f}}$ is

$$Z = \sum_{N=1}^{\infty} \frac{1}{(2\pi)^N N!} \int \prod_{n=1}^{N} \mathrm{d}p_n \int_{x_n \in [0,L]} \prod_{n=1}^{N} \mathrm{d}x_n \exp\left[-\sum_n w(p_n)\right] \tag{136}$$

and this can be calculated explicitly by elementary integration. For free radiation, we instead do a classical field theory, for instance, for a thermal state,

$$Z = \int \mathcal{D}\Pi\mathcal{D}\Phi \exp\left[-\frac{\beta}{2}\int_0^L \left(\Pi(x)^2 + (\partial_x \Phi(x))^2\right)\right]. \tag{137}$$

The fields can be Fourier-transformed, the integral in the exponential is diagonalised, and the functional integral can be evaluated. In the quantum case, we may formulate the problem either as a field theory or as a particle system. For instance, in terms of for free particles, we evaluate the trace

$$Z = \mathrm{Tr}_{\mathcal{H}_L} \exp\left[-\frac{1}{2}\sum_{n=1}^{N} \hat{p}_n^k\right] \tag{138}$$

on the Hilbert space $\mathcal{H}_L$ of $N$-particle wave functions, with given statistics (fermionic or bosonic). Again by Fourier transform, this factorises the trace into a product of traces over the fillings of the Fourier modes, and the allowed filling – either 0 or 1 for fermions, or $0, 1, 2, 3, \ldots$ for bosons, are determined by the statistics. Anyonic cases can be done by restricting the allowed fillings to a finite set $0, 1, 2, 3, \ldots, m$.

**Example 3.7.** *The partition function for the hard rod gas of Example 3.4 can be written explicitly,*

$$Z = \sum_{N=1}^{\infty} \frac{1}{(2\pi)^N N!} \int \prod_{n=1}^{N} \mathrm{d}p_n \int_{x_n \in [0,L]} \prod_{n=1}^{N} \mathrm{d}x_n \prod_{m,n} \Theta(|x_n - x_m| - a) \exp\left[-\sum_n w(p_n)\right], \tag{139}$$

*where $\Theta(\cdot)$ is Heavyside's step function. That is, it is like a free particle model, but with the constraint that the particles must be pairwise separated by a distance at least $a$. In this case, $\varphi(p - p') = -a$ as mentioned. The partition function is more difficult to evaluate fully rigorously, however the arguments presented in Section 3.2 apply, and thus results in the statistics of a classical particle, $F(\epsilon) = -\mathrm{e}^{-\epsilon}$.*

**Example 3.8.** *The partition function for the Lieb-Liniger model, Example 3.5, based upon the Hamiltonian (211), is evaluated by the quantum TBA calculation. Here, it turns out, there are two ways of describing the result: either in terms of fermionic quasiparticle, or bosonic quasiparticles, see the discussion in [152]. This points to the ambiguity in the set of asymptotic states; this set depends on the choice of vacuum, see Remark 3.1. The fact that fermionic quasiparticles can occur is intuitively understood by the delta-function repulsion: this essentially forbids particles to be at the same point, a bit like the fermoinic statistics does. Indeed, at infinite interaction $g \to \infty$, the model becomes that of free fermions. The bosonic description, on the other hand, is more natural when taking the limit $g \to 0$.*

*The elementary two-body calculation of the scattering shift, giving (102), actually corresponds to fermionic quasiparticles, with $F(\epsilon) = -\log(1 + \mathrm{e}^{-\epsilon})$.*

**Example 3.9.** *Another class of important examples are the quantum spin chains, such as the Heisenberg chain*

$$H = J \sum_{n\in\mathbb{Z}} \vec{\sigma}_n \cdot \vec{\sigma}_{n+1}, \tag{140}$$

*where $\vec{\sigma}_n$ is the vector of Pauli matrices acting nontrivially on site $n$. I refer to [70] for the TBA of this and related model, which, it turns out, fits within the above general "TBA technology".*

## 3.4 Equations of state

In the Section 3.2, I have described how to obtain the free energy density f of the thermodynamic description. This, however, does not give the *equations of state* (22), the relation between the average currents and densities, which are essential in order to develop the hydrodynamic theory. Since currents know about the definition of time, one extra ingredient is needed:

- The energy function $E(p)$. This is $E(p) = h_i(p)$ for $i$ such that $Q_i = H$ the Hamiltonian, i.e. the generator of time evolution.

For the Hamiltonian (85), this is $E(p) = p^2/2$.

A guess for the equations of state may be obtained as follows. Recall that in relativistic quantum field theory, there is *crossing symmetry*: a symmetry under the exchange of space and time, $(x, t) \to (-it, ix)$. Under this change, the relativistic rapidities change as $\theta \to i\pi/2 - \theta$. This means with momentum and energy being $p(\theta) = m \sinh\theta$ and $E(\theta) = m \cosh\theta$, crossing symmetry gives $(p, E) \to (iE, -ip)$. Naturally, under crossing symmetry, densities and currents are likewise exchanged. Therefore, if we know how to evaluate densities, we simply have to apply crossing symmetry to the state in order to obtain currents. This means, we have to change momentum to energy.

The simplest proposition is to write the *free energy flux* (25), simply by taking the TBA expression for the free energy density (111), and replacing the momentum measure by the "energy measure",

$$g = \int \frac{dp}{2\pi} E'(p) F(\epsilon(p)), \tag{141}$$

where $E'(p) = dE(p)/dp$. Applying the derivative as per (25), we obtain, by a similar calculation as in Section 3.2,

$$j_i = \int \frac{dp}{2\pi} E'(p) n(p) h_i^{dr}(p) \tag{142}$$

and then

$$j_i = \int \frac{dp}{2\pi} (E')^{dr}(p) n(p) h_i(p). \tag{143}$$

We can therefore define the **effective velocity**

$$v^{eff}(p) = \frac{(E')^{dr}(p)}{1^{dr}(p)} \tag{144}$$

and we obtain

$$j_i = \int dp\, v^{eff}(p) \rho_p(p) h_i(p). \tag{145}$$

Notice that $1 = dp/dp$, thus the effective velocity is $(E')^{dr}(p)/(p')^{dr}(p)$. This expression holds under more general spectral parametrisations. In this form, the effective velocity appears as a version of the group velocity which takes into account, via the dressing operation, the interaction with the quasiparticles in the gas.

Expression (145) has a very natural interpretation: the average current is obtained by evaluating the quantity $h_i(p)$ of charge carried by the quasiparticles of bare momentum $p$, times the density of quasiparticles $\rho_p(p)$ at this momentum, multiplied by the velocity $v^{eff}(p)$ at which they are going within the gas.

In order make this interpretation even clearer, and to get additional intuition about the effective velocity, I will now show that it satisfies the following linear integral equation:

$$v^{eff}(p) = E'(p) + \int dp'\, \varphi(p - p') \rho_p(p')(v^{eff}(p') - v^{eff}(p)). \tag{146}$$

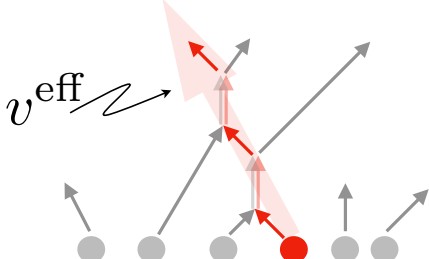

Figure 11: The effective velocity emerges as a tagged quasiparticle travels through the gas. In this cartoon, the scattering shifts arise from time delays due to quasiparticles sticking together for a while.

One can see this equation as defining the "**effectivisation**" $v(p) \to v^{\text{eff}}(p)$ of the group velocity $v(p) = E'(p)$. To show this, bring one term on the right-hand side of (146) to the left-hand side, this is

$$\left[1 + \int \mathrm{d}p' \, \varphi(p-p')\rho_{\text{p}}(p')\right]v^{\text{eff}}(p) = E'(p) + \int \mathrm{d}p \, \varphi(p-p')\rho_{\text{p}}(p')v^{\text{eff}}(p'). \tag{147}$$

According to (110) and using (127), this becomes

$$2\pi\rho_s(p)v^{\text{eff}}(p) = E'(p) + \int \frac{\mathrm{d}p}{2\pi} \, \varphi(p-p')n(p)2\pi\rho_s(p')v^{\text{eff}}(p'). \tag{148}$$

Therefore, with (116), we identify

$$2\pi\rho_s(p)v^{\text{eff}}(p) = (E')^{\text{dr}}(p). \tag{149}$$

With (126), we finally find (144).

Eq. (146) is useful because it leads to a very natural interpretation of the effective velocity: it is the *large-scale effective velocity of a quasiparticle as it travels through the gas, obtained by taking into account the scattering shifts it accumulates at collisions with the other quasiparticles of the gas*, see Fig. 11.

In order to see this, consider a quasiparticle of momentum $p$, which we tag and follow. Its displacement $\Delta x$ within the gas, as it travels for a time $\Delta t$, is

$$\Delta x = E'(p)\Delta t + \sum_n \varphi(p-p_n) \times \begin{cases} -1 & \text{(tagged quasiparticle hits quasiparticle } n \text{ on its right)} \\ +1 & \text{(tagged quasiparticle hits quasiparticle } n \text{ on its left)} \end{cases}, \tag{150}$$

where the sum is over all quasiparticles it crosses in the time $\Delta t$. This formula is interpreted as the linear displacement $p\Delta t$ due to its "free" propagation between collisions at the group velocity $v(p) = E'(p)$, plus the accumulation of the shifts $\varphi(p-p_n)$ it incurs at every collision. This is just an interpretation, as, of course, in general, in a finite density gas, the quasiparticle's trajectory is never actually linear. This interpretation is valid on large scales only: the quasiparticle doesn't actually "jump" at collisions, rather it overall accumulates shifts. There is one subtlety in the formula: it is important that the direction of the "jump", at every collision, depends on the direction in which the collision occurs, *not on the velocity difference* $v(p) - v(p_n)$. That is, if the tagged quasiparticle hits a quasiparticle on its right (left), then it jumps to the left (right) for $\varphi(p-p_n) > 0$ [and the opposite direction for $\varphi(p-p_n) < 0$] – meaning that the contribution of the collision to the tagged quasiparticle's overall displacement is to the left

(right). Because the tagged quasiparticle's trajectory is different from a linear displacement of velocity $v(p)$, it may be that, even for $v(p) > v(p_n)$, the quasiparticle of momentum $p_n$ collide with the tagged quasiparticle (of momentum $p$) on its left, instead of its right as would be suggested by their group velocities.

The exact displacement $\Delta x$ depends on the precise configuration, but let us take its expectation $\langle \Delta x \rangle$, over the distributions of the quasiparticles in the gas for a given state. Equivalently, by self-averaging, we can think of a large trajectory in a typical configuration of the gas. The result gives the effective velocity:

$$\langle \Delta x \rangle = E'(p)\Delta t + \int dq\, s(p,q)\omega(p,q)\varphi(p-q) = v^{\mathrm{eff}}(p)\Delta t\,, \tag{151}$$

where $s(p,q)$ is the direction of the jump with a quasiparticle of momentum $q$, and $\omega(p,q)$ is the probability that it hits a quasiparticle of momentum $q$. The fact that the direction and probability only depend on the momenta, and their exact expressions, is argued for as follows. For the direction, it is clear that, on large scales, if $v^{\mathrm{eff}}(p) > v^{\mathrm{eff}}(q)$, then the quasiparticles will have to have collided exactly one more time as $p$ (left) $\leftrightarrow$ $q$ (right), then as $q$ (left) $\leftrightarrow$ $p$ (right). Thus the sign of the direction can be taken as $-1$. Hence we have

$$s(p,q) = \mathrm{sgn}(v^{\mathrm{eff}}(q) - v^{\mathrm{eff}}(p)). \tag{152}$$

The probability of colliding is proportional to both the density of quasiparticles $\rho_{\mathrm{p}}(q)$, and to a geometric contribution due to the angle at which, on large scales, the quasiparticles propagate, $|v^{\mathrm{eff}}(p) - v^{\mathrm{eff}}(q)|$ (e.g. if they are parallel, they will never meet). Thus we obtain

$$\omega(p,q) = \rho_{\mathrm{p}}(q)|v^{\mathrm{eff}}(p) - v^{\mathrm{eff}}(q)|\Delta t. \tag{153}$$

Putting things together, we obtain (146).

Expression (141), on which all was based, was of course just a guess. In relativistic QFT, it can be derived more precisely [1] using crossing symmetry. The semi-classical interpretation via the accumulation of jumps [160] helps justify it. The expression (144) for an effective velocity appeared in [161] in the study of propagations of excitations in the context of quantum quenches. The formula (145) for the average current has been verified numerically in many ways both in quantum [2] and classical [162] models. There is a proof in QFT [108] using a form factor expansion. Following similar arguments, expressions valid in any Bethe state of quantum chains were obtained [112], with a simpler derivation using certain boost operators [113]. Other proofs were recently found in [114, 115] using the symmetry of the B-matrix (27), applicable in certain situations such as Galilean and relativistic gases, and the XXZ spin chain. In a large family of quantum integrable models, a full algebraic construction of the currents has been given [116]. The hydrodynamic equations obtained from formulae (145) (see Chapter 4) in fact appeared earlier than all works on GHD, proven rigorously in [77] for the hard rod gas, and derived from the inverse scattering method in [74] in the context of soliton gases.

Finally, the discussion of the currents is completed by considering the entropy flux (49). Again, this can be calculated (I leave it as an exercise to figure out which of the equations

we've derived until now is used at every step!):

$$
\begin{aligned}
j_s &= \sum_i \beta^i j_i - g \\
&= \int dp \left[ \sum_i \beta^i h_i(p) v^{\text{eff}}(p) \rho_p(p) - \frac{1}{2\pi} E'(p) F(\epsilon(p)) \right] \\
&= \int dp \left[ w(p) v^{\text{eff}}(p) \rho_p(p) - \frac{1}{2\pi} E'(p) F(\epsilon(p)) \right] \\
&= \int dp \left[ \epsilon(p) v^{\text{eff}}(p) \rho_p(p) - \frac{1}{2\pi} \left( \int dq\, v^{\text{eff}}(p) \rho_p(p) \varphi(p-q) F(\epsilon(q)) + E'(p) F(\epsilon(p)) \right) \right]
\end{aligned}
$$

and

$$
\begin{aligned}
&= \int dp \left[ \epsilon(p) v^{\text{eff}}(p) \rho_p(p) - \frac{1}{2\pi} \left( \int dq\, v^{\text{eff}}(q) \rho_p(p) \varphi(p-q) F(\epsilon(q)) + v^{\text{eff}}(p) F(\epsilon(p)) \right) \right] \\
&= \int dp\, \epsilon(p) v^{\text{eff}}(p) \rho_p(p) - \frac{1}{2\pi} \left( \int dq\, v^{\text{eff}}(q) (2\pi \rho_s(q) - 1) F(\epsilon(q)) + \int dp\, v^{\text{eff}}(p) F(\epsilon(p)) \right) \\
&= \int dp\, v^{\text{eff}}(p) \rho_s(p) \left[ \epsilon(p) n(p) - F(\epsilon(p)) \right].
\end{aligned}
\tag{154}
$$

Thus, the entropy current is, again quite naturally, obtained by multiplying the integrand of the entropy density (130) – which is the entropy density per unit phase-space element – by the effective propagation velocity of the quasiparticles, $v^{\text{eff}}(p)$.

**Remark 3.10.** *The semi-classical interpretation of $v^{\text{eff}}$ discussed above has been explains the Euler-scale hydrodynamic equations found in classical soliton gases [72–74]. It can also be used in the simpler hard rod gas [77], where $\varphi(p) = -a$ is a negative constant, see Example 3.4. There, the tagged quasiparticle is subject to exact linear displacements punctuated by actual jumps. The picture of exact linear displacements punctuated by actual jumps is at the source of the flea gas algorithm, discussed in Section 4.4, valid for any $\varphi(p)$.*

## 3.5 Hydrodynamic matrices, Drude weights and normal modes

Finally, we complete the description of the thermodynamic state by deriving the expressions for the matrices $A, B, C, D$ discussed in Section 2.2.

The matrices $B$ and $C$ are obtained by differentiation of $j_i$ and $q_i$, respectively, following (27) and (20). Let us do the case of $C$, as the case of $B$ is similar. A direct way is to start with expression (114), in the form

$$
q_i = \frac{1}{2\pi} 1 \cdot n (1 - Tn)^{-1} h_i
\tag{155}
$$

(see (118) and (120)). We observe that

$$
-\frac{\partial n(p)}{\partial \beta^i} = -\frac{d^2 F(\epsilon)}{d\epsilon^2} \bigg|_{\epsilon = \epsilon(p)} \frac{\partial \epsilon(p)}{\partial \beta^i} = f(p) n(p) h_i^{\text{dr}}(p),
\tag{156}
$$

where we used the equation in the paragraph above (116) and Eq. (115), and we introduced the **statistical factor**

$$
f(p) = -\frac{d^2 F(\epsilon)/d\epsilon^2}{dF(\epsilon)/d\epsilon} \bigg|_{\epsilon = \epsilon(p)}.
\tag{157}
$$

This takes the form $f(p) = 1$, $f(p) = n(p)$, $f(p) = 1 - n(p)$ and $f(p) = 1 + n(p)$ for classical particle, classical radiation, fermions and bosons, respectively. Then

$$
\begin{aligned}
\mathsf{C}_{ij} &= -\frac{\partial \mathsf{q}_i}{\partial \beta^j} \\
&= \frac{1}{2\pi} \, 1 \cdot (n(1 - Tn)^{-1}T + 1)(-\partial_{\beta^j} n)(1 - Tn)^{-1} h_i \\
&= \frac{1}{2\pi} \, 1 \cdot ((1 - nT)^{-1}nT + 1)(f n h_j^{\mathrm{dr}}) h_i^{\mathrm{dr}} \\
&= \frac{1}{2\pi} \, 1 \cdot (1 - nT)^{-1} \, n f h_i^{\mathrm{dr}} h_j^{\mathrm{dr}} \\
&= \frac{1}{2\pi} \, n(1 - Tn)^{-1} 1 \cdot f h_i^{\mathrm{dr}} h_j^{\mathrm{dr}} \\
&= \frac{1}{2\pi} \, n 1^{\mathrm{dr}} \cdot f h_i^{\mathrm{dr}} h_j^{\mathrm{dr}} \\
&= \int \mathrm{d}p \, \rho_{\mathrm{p}}(p) f(p) h_i^{\mathrm{dr}}(p) h_j^{\mathrm{dr}}(p).
\end{aligned}
\tag{158}
$$

A similar calculation gives

$$
\mathsf{B}_{ij} = -\frac{\partial \mathsf{j}_i}{\partial \beta^j} = \int \mathrm{d}p \, v^{\mathrm{eff}}(p) \rho_{\mathrm{p}}(p) f(p) h_i^{\mathrm{dr}}(p) h_j^{\mathrm{dr}}(p),
\tag{159}
$$

which indeed satisfies the expected symmetry (27).

For the matrix $\mathsf{A}_i{}^j$, it can be obtained again by differentiation as per (26). One may obtain its integral-operator kernel, acting on functions of $p$, by using this formula, with the replacement $\mathsf{q}_j \mapsto \rho_{\mathrm{p}}(p)$, which is justified by (123). Let me instead directly use the results obtained above, along with the general structure that we have in (27). The main assumption is that *the set of functions $h_i^{\mathrm{dr}}$ form a complete set with respect to the inner product $h_i^{\mathrm{dr}} \bullet h_j^{\mathrm{dr}} = \mathsf{C}_{ij}$ given by (158)*. This is equivalent to the assumption of completeness of the set of conserved densities in the Hilbert space based on the inner product (19). Then

$$
\mathsf{B}_{ij} = \sum_k \mathsf{A}_i{}^k \mathsf{C}_{kj}
\tag{160}
$$

implies

$$
\int \mathrm{d}p \, v^{\mathrm{eff}}(p) \rho_{\mathrm{p}}(p) f(p) h_i^{\mathrm{dr}}(p) h_j^{\mathrm{dr}}(p) = \int \mathrm{d}p \, \rho_{\mathrm{p}}(p) f(p) \sum_k \mathsf{A}_i{}^k h_k^{\mathrm{dr}}(p) h_j^{\mathrm{dr}}(p)
\tag{161}
$$

and by completeness on the index $j$, we deduce

$$
\sum_j \mathsf{A}_i{}^j h_j^{\mathrm{dr}}(p) = v^{\mathrm{eff}}(p) h_i^{\mathrm{dr}}(p).
\tag{162}
$$

This is the eigenvalue equation (30), where we identify the index $\ell$ parametrising the spectrum with the momentum $p$. That is, the flux Jacobian has a continuous spectrum given by the allowed effective velocities of the quasiparticles.

This immediately gives the Drude weight by using (35):

$$
\begin{aligned}
\mathsf{D}_{ij} = \mathsf{A}^2 \mathsf{C} &= \int \mathrm{d}p \, \rho_{\mathrm{p}}(p) f(p) \sum_{k,l} \mathsf{A}_i{}^k \mathsf{A}_k{}^l h_l^{\mathrm{dr}}(p) h_j^{\mathrm{dr}}(p) \\
&= \int \mathrm{d}p \, \rho_{\mathrm{p}}(p) f(p) \left[ v^{\mathrm{eff}}(p) \right]^2 h_i^{\mathrm{dr}}(p) h_j^{\mathrm{dr}}(p),
\end{aligned}
\tag{163}
$$

which is one of the most physically important results of the analysis up to now.

It is convenient to introduce integral operators, acting on a space of functions of $p$, in order to represent the hydrodynamic matrices, avoiding the use of any explicit basis of functions $h_i(p)$. For this purpose, let me define integral operators $A$, $B$, $C$ and $D$ via

$$\sum_j \mathsf{A}_i{}^j h_j(p) = (A^{\mathrm{T}} h_i)(p), \quad \mathsf{B}_{ij} = h_i \cdot B h_j, \quad \mathsf{C}_{ij} = h_i \cdot C h_j, \quad \mathsf{D}_{ij} = h_i \cdot D h_j. \tag{164}$$

Note how I had to treat covariant and contravariant indices differently. In particular, the definitions make sure that we keep the relation $B = AC$ for integral operators, as $h_i \cdot B h_j = \mathsf{B}_{ij} = \mathsf{A}_i{}^k \mathsf{C}_{kj} = A^{\mathrm{T}} h_i \cdot C h_j = h_i \cdot A C h_j$. The first equation in (164), combined with (162), gives

$$v^{\mathrm{eff}}(p) h_i^{\mathrm{dr}}(p) = \sum_j \mathsf{A}_i{}^j h_j^{\mathrm{dr}}(p) = \left(\sum_j \mathsf{A}_i{}^j h_j\right)^{\mathrm{dr}}(p) = (A^{\mathrm{T}} h_i)^{\mathrm{dr}}(p),$$

thus

$$A = (1 - nT)^{-1} v^{\mathrm{eff}} (1 - nT). \tag{165}$$

The others can be directly read off the expressions obtained above,

$$B = (1 - nT)^{-1} v^{\mathrm{eff}} \rho_{\mathrm{p}} f (1 - Tn)^{-1} \tag{166}$$

$$C = (1 - nT)^{-1} \rho_{\mathrm{p}} f (1 - Tn)^{-1} \tag{167}$$

$$D = (1 - nT)^{-1} (v^{\mathrm{eff}})^2 \rho_{\mathrm{p}} f (1 - Tn)^{-1}. \tag{168}$$

In the next Chapter, I discuss the hydrodynamics of integrable models, called generalised hydrodynamics, based on the TBA technology developed in the present Chapter. However, even before discussing aspects specific to hydrodynamics, we already have the elements in order to build the normal modes of generalised hydrodynamics, as defined in (45) and (46). Indeed, we have the necessary hydrodynamic matrices! Clearly, (165) is the integral-operator form of the main equation (44) defining the matrix change-of-basis $\mathsf{R}$. It is then immediate that we can take, for the kernel $R$ associated to the matrix $\mathsf{R}$ (in the same way as $A$ is related to $\mathsf{A}$ in (164)),

$$R = \frac{1}{\rho_{\mathrm{s}}} (1 - nT) \tag{169}$$

(from its definition, this can be premultiplied by any state-dependent diagonal operator, but for the specific results below, we choose $1/\rho_{\mathrm{s}}$). Equation (45), which defines the normal modes, can be written as

$$-\frac{\partial n_i}{\partial \beta^j} = \sum_k \mathsf{R}_i{}^k \mathsf{C}_{kj}. \tag{170}$$

In terms of integral kernels, the right-hand side is (omitting the explicit dependence on $p$ of the integrand)

$$\sum_k \mathsf{R}_i{}^k \mathsf{C}_{kj} = \int \mathrm{d}p \, h_i R C h_j = \int \mathrm{d}p \, h_i nf h_j^{\mathrm{dr}} = h_i \cdot nf h_j^{\mathrm{dr}}. \tag{171}$$

As $\mathsf{q}_i = h_i \cdot \rho_{\mathrm{p}}$ (Eq. (123)), likewise the normal modes take the form $n_i = h_i \cdot \tilde{n}$ for a normal-mode function $\tilde{n}(p)$. Thus we must have

$$-\frac{\partial \tilde{n}}{\partial \beta^j} = nf h_j^{\mathrm{dr}}. \tag{172}$$

We compare this with the result (156) for the derivative of $n$, and conclude that we can take

$$\tilde{n}(p) = n(p). \tag{173}$$

That is, we have found that the **occupation function gives the normal modes of generalised hydrodynamics**. There is one normal mode for every value of momentum $p$, in agreement with the continuous spectrum $v^{\text{eff}}(p)$ that we found in Section 3.4.

As a final calculation, using what we have just established, we can show that a property of *linear degeneracy*, as introduced in Section 2.5, holds in integrable models. This, as I explained in that Section, has strong consequences on the type of solutions hydrodynamic problems can display, which we will see in particular in Section 4.2. This is the property according to which the effective velocity of a normal mode does not depend on this normal mode (although it may depend on all other normal modes). We should keep in mind that in integrable models, as there is a continuum of effective velocities, the consequences of this property are not as clear-cut. For instance, the modes associated to different effective velocities do not all separate well after large times, a notion that is sometimes made use of in conventional fluids. This subtlety does not seem to affect the application of linear degeneracy to the Riemann problem, however.

Let me then calculate

$$\frac{\delta v^{\text{eff}}(p)}{\delta n(p')}. \tag{174}$$

For this purpose, I will use the expression (144), the ratio of the dressed quantities $(E')^{\text{dr}}(p)$ and $1^{\text{dr}}(p)$. There is a nice general formula:

$$
\begin{aligned}
\frac{\delta h^{\text{dr}}(p)}{\delta n(p')} &= \frac{\delta}{\delta n(p')}\big[(1 - Tn)^{-1}h\big](p) \\
&= \Big[(1 - Tn)^{-1}T\frac{\delta n(\cdot)}{\delta n(p')}(1 - Tn)^{-1}h\Big](p) \\
&= T^{\text{dr}}(p, p')h^{\text{dr}}(p'),
\end{aligned} \tag{175}
$$

where I used $\delta n(p)/\delta n(p') = \delta(p - p')$ and the dressed scattering kernel (129). Hence,

$$
\begin{aligned}
\frac{\delta v^{\text{eff}}(p)}{\delta n(p')} &= \frac{\delta}{\delta n(p')}\frac{(E')^{\text{dr}}(p)}{1^{\text{dr}}(p)} \\
&= T^{\text{dr}}(p, p')\frac{(E')^{\text{dr}}(p')1^{\text{dr}}(p) - 1^{\text{dr}}(p')(E')^{\text{dr}}(p)}{\big[1^{\text{dr}}(p)\big]^2} \\
&= T^{\text{dr}}(p, p')\frac{1^{\text{dr}}(p')}{1^{\text{dr}}(p)}\big[v^{\text{eff}}(p') - v^{\text{eff}}(p)\big].
\end{aligned} \tag{176}
$$

We immediately observe that $\delta v^{\text{eff}}(p)/\delta n(p) = 0$, which is the linear degeneracy property sought.

**Remark 3.11.** *The question of a complete space of functions $h_i(p)$ which would correspond to the complete set of conserved charges of a model is a tricky one. The basic answer lies within the theory of hydrodynamic projections and pseudolocal charges. As explained at the beginning of Chapter 3, the natural object is the Hilbert space built from the inner product (19). The subspace of this Hilbert space composed of all elements that are invariant under time evolution is the subspace of conserved densities. If this subspace, is countable dimensional (which is the case, for instance, in quantum spin chains), then the static covariance matrix (158) is the sesquilinear form to be used. Therefore, the Hilbert space of conserved quantities should be identified with that of functions $h(p)$ of finite norm, under the inner product*

$$(h, g) = \int \mathrm{d}p\, \rho_{\text{p}}(p)f(p)\big[h^{\text{dr}}(p)\big]^* g^{\text{dr}}(p). \tag{177}$$

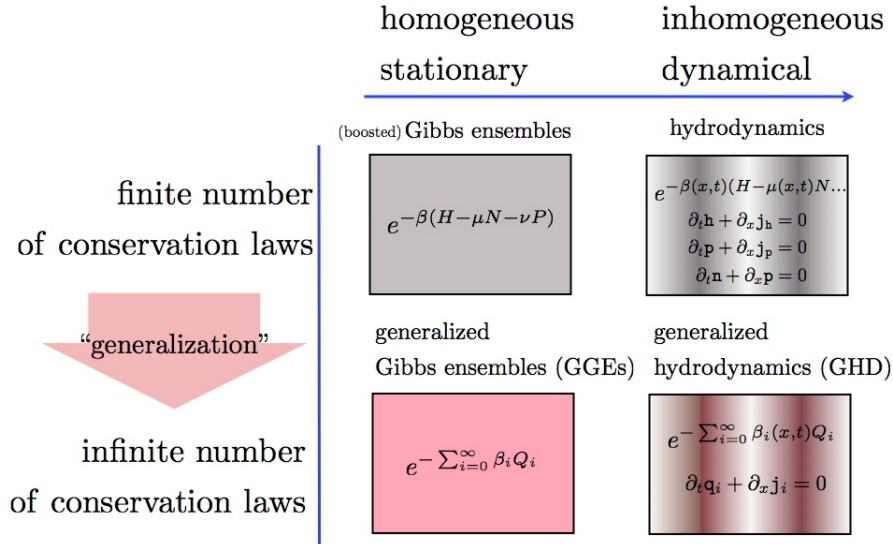

Figure 12: Where GHD fits.

*This Hilbert space depends on the state $\rho_p(p)$, in accordance with the general theory of pseudolocal charges [121].*

*However, the change of coordinates (135) between $\{\rho_p(p)\}$ and $\{q_i\}$ requires, instead, the inversion of (123). Here we would like to see $q_i = q[h_i]$ with $q[\cdot]$ a map on the space of functions just defined. If we think of the state as the integration along the path (84) on the manifold of states starting from the infinite-temperature state, as proposed in [121], and if the Hilbert spaces (the tangent spaces) at each point satisfy an appropriate inclusion property (as may be achieved if the path is directed by the strength of clustering of the states visited), then the result is a continuous linear functional on h, and by the Riesz representation theorem, is representable by an element of the Hilbert space. It is perhaps in this sense that we should understand (123). More research on these aspects would be needed.*

## Chapter 4   Generalised hydrodynamics

In Chapter 3, I developed the TBA technology, which expresses all elements of the statistical mechanics of maximal entropy states introduced in Section 2.2 in terms of asymptotic states of the underlying microscopic model. We now have all the ingredients necessary in order to work out the hydrodynamics, Sections 2.3 (Euler scale), 2.4 (diffusive scale), 2.5 (Riemann problem) and 2.6 (correlation functions). As this hydrodynamic theory is based not on Gibbs ensembles, but on generalised Gibbs ensembles (GGEs), it has been dubbed generalised hydrodynamics (GHD). See Fig. 12.

I won't follow exactly this order of Sections 2.3 to 2.6, instead concentrating on the Euler scale as until now most of the theory is developed at this scale, and only at the end discuss diffusion. So I'll start with the Euler hydrodynamic equations, the basic equations of GHD; then I'll develop various applications, including the Riemann problem, and explain additional concepts which were not discussed above such as force terms and numerical solutions; then I'll discuss correlation functions, again only at the Euler scale; then I'll finish with diffusion.

## 4.1 Fundamental equations

Repeating the ideas of Chapter 2, the fundamental assumption of GHD is that, when the variation scales of the state become large enough in space-time, then we may assume that on every "fluid cell" – large enough to contain a thermodynamic amount of miscrocopic particles, but small enough so that their state vary slowly in space-time – the state has maximised entropy with respect to all available linearly extensive conserved quantities. This is expressed in the approximation (36), where averages of observables at $(x, t)$ are evaluated by calculating their average in a maximal entropy state whose Lagrange parameters depend on $(x, t)$.

In Chapter 3, I explained how such maximal entropy states can be represented using the asymptotic states of scattering theory. In fact, from this, we obtained various systems of coordinates which can be used in order to represent a given maximal entropy state, see (135). Hence, according to (36), we must introduce space-time dependence in all these coordinate systems [although, of course, the way they are related to each other is not dependent on space-time]. In particular, we replace

$$\rho_{\rm p}(p) \to \rho_{\rm p}(p, x, t), \quad n(p) \to n(p, x, t). \tag{178}$$

The quasiparticle density, per unit of length and momentum, now depends not just on the momentum, but on space-time as well. This function of $p$ tells us in what state the fluid cell at space-time point $(x, t)$ is.

Once we have this, we simply write the Euler hydrodynamic equation (41), and replace in it the expressions for $\mathsf{q}_i$ and $\mathsf{j}_i$ from the TBA technology, (123) and (145) respectively. Bringing the space-time derivatives inside the momentum integral,

$$\int \mathrm{d}p \, h_i(p)\Big(\partial_t \rho_{\rm p}(p, x, t) + \partial_x\big[v^{\rm eff}(p, x, t)\rho_{\rm p}(p, x, t)\big]\Big) = 0. \tag{179}$$

Using completeness of the set of functions $\{h_i\}$, we then extract the bracket, and obtain the Euler-scale **GHD equations**:

$$\partial_t \rho_{\rm p}(p, x, t) + \partial_x\big[v^{\rm eff}(p, x, t)\rho_{\rm p}(p, x, t)\big] = 0. \tag{180}$$

This is the most important result from our analysis, and is at the basis of much of the development in GHD.

Recall that the $x, t$-dependence of $\rho_{\rm p}(p, x, t)$ is because this function represents the state at the fluid cell $x, t$. The $x, t$-dependence of $v^{\rm eff}(p, x, t)$ arises because $v^{\rm eff}(p, x, t)$ is the effective velocity evaluated in the state at $x, t$: it is a functional of $\rho_{\rm p}(\cdot, x, t)$. For instance, in (146), the quasiparticle density in the integrand becomes $x, t$-dependent, $\rho_{\rm p}(p', x, t)$, hence the solution $v^{\rm eff}(p, x, t)$ also is. Likewise, in (144), the *dressing operation introduces an $x, t$-dependence* even if the function it dresses is not $x, t$-dependent, i.e. we write $h^{\rm dr}(p, x, t)$ for $h(p)$, because (116) involves the occupation function, $n(p, x, t)$.

Equation (180) is a integro-differential equation, which is first-order in derivatives. It is highly nonlinear, because $v^{\rm eff}(p, x, t)$ depends nonlinearly on the solution $\rho_{\rm p}(p, x, t)$ that we seek. The equation is, however, homogeneous in space-time, in that it does not depend on $x, t$ except through the solution sought $\rho_{\rm p}(p, x, t)$. In order to "solve" it, one would need to set an initial condition $\rho_{\rm p}(p, x, 0)$, then use this in order to evaluate $v^{\rm eff}(p, x, 0)$, and, with a finite-element discretisation of (180), solve for $\rho_{\rm p}(p, x, \mathrm{d}t)$, etc. The initial condition depends on the problem, but it may be obtained by the "local density approximation", which is the hydrodynamic approximation as applied to the initial state. For instance, given space-dependent Lagrange parameters $\beta^i(x)$, such as a space-dependent chemical potential, then one evaluates $\rho_{\rm p}(p, x, 0)$ by applying, at every $x$, the TBA technology based on the Lagrange parameters $\beta^i(x)$.

Continuing our analysis, we can write the GHD equations in a quasilinear form, as in (42). Using the integral-operator representation of the flux Jacobian (164), (165), we obtain (omitting the explicit $p, x, t$, dependence)

$$\partial_t \rho_p + A \partial_x \rho_p = 0. \tag{181}$$

This is not, however, the most useful equation, because it involves an integral operator. Instead, we have already found in Section 3.5 that the **occupation function gives the GHD normal modes**, diagonalising this operator, as defined in (45) and (46), see Eq. (173). Therefore, the hydrodynamic equations they satisfy, (46), take the form

$$\partial_t n(p, x, t) + v^{\text{eff}}(p, x, t) \partial_x n(p, x, t) = 0. \tag{182}$$

Compared with (180), the effective velocity is now *outside the derivative*. Eq. (182), and its generalisations and specialisations, is by far the most useful equation in GHD.

There was quite a lot of theory involved in Section 3.5 in order to deduce that the occupation function gives the GHD normal modes. But it is possible to derive (182) much more directly, in a perhaps less conceptual way, as was originally done [1,2]. We can simply use the forms (122) and (143) for the average densities and currents, and apply the time and space derivatives, recalling that all space-time dependence is in the occupation function $n$. In general, the result of a derivative with respect to some parameter $u$ (which will be $x$ or $t$) of $\int \mathrm{d}p \, hng^{\text{dr}}$ is

$$\partial_u \int \mathrm{d}p \, hng^{\text{dr}} = \partial_u \int \mathrm{d}p \, hn(1 - Tn)^{-1} g = \int \mathrm{d}p \, h(1 - nT)^{-1} \partial_u n \, g^{\text{dr}}. \tag{183}$$

Therefore, again using completeness of the set of functions $\{h_i\}$ we have

$$\partial_t n \, 1^{\text{dr}} + \partial_x n \, (E')^{\text{dr}} = 0, \tag{184}$$

which is indeed equivalent to (182) by using (144).

In words, Equation (182) means that the occupation function – which is the density of quasiparticle per unit available space in asymptotic coordinates – is convectively transported by the GHD flow. That is, the value of $n(p, x, t)$ along the *characteristic curve* for $p$, the curve whose tangents at every point $x, t$ are $v^{\text{eff}}(x, t, p)$, is constant along the curve. This picture will be very useful in Section 4.3, where the characteristics will be studied.

Equations (180) and (182) are the most important results of this section.

An immediate consequence of these two equations is that *any function $r(n)$ gives rise to a conservation law*:

$$\partial_t \big[ \rho_p r(n) \big] + \partial_x \big[ v^{\text{eff}} \rho_p r(n) \big] = 0. \tag{185}$$

This is a consequence of combining (180) with (182). Hence, in particular, taking $r(n) = 1/n$ and using (127), we find that the *state density also is a conserved density*:

$$\partial_t \rho_s + \partial_x \big[ v^{\text{eff}} \rho_s \big] = 0. \tag{186}$$

We also see, using the expressions (130) and (154), that, as expected, the entropy density and flux satisfy a continuity equation,

$$\partial_t \mathsf{s} + \partial_x \mathsf{j}_s = 0. \tag{187}$$

**Remark 4.1.** *The derivation presented here has the drawback that we need to assume appropriate completeness of the set of functions $\{h_i(p)\}$. There is also the question as to the correctness of assuming that locally, generalised thermalisation happens, even though the GGEs occurring may be based on pseudolocal charges with quite weak locality properties. These problems are circumvented in a different derivation, based directly on the scattering theory, outlined in Section 4.3.*

## 4.2 The Riemann problem

Let me now consider the Riemann problem of hydrodynamics, as applied to GHD [1, 2, 132, 163]. This is the problem of solving the Euler equation with the initial condition where on the left- and right-hand side of space, the fluid is set to different, otherwise homogeneous, states, see Eq. (63). As explained in Section 2.5, it is convenient to use the normal modes, which are the occupation functions $n(p, x, t)$. We therefore set the initial state by considering two maximal entropy states, say as determined by the Lagrange parameters $\beta_l^i$ and $\beta_r^i$, and set

$$n(p, x, 0) = \begin{cases} n_l(p) = n(p)\Big|_{w = \sum_i \beta_l^i h_i(p)} & (x < 0) \\ n_r(p) = n(p)\Big|_{w = \sum_i \beta_r^i h_i(p)} & (x > 0) \end{cases}, \tag{188}$$

where we use the transformations described in (135).

We must now solve the problem (67) with the asymptotic conditions (65). This translates into

$$\left[v^{\text{eff}}(p, \xi) - \xi\right] \frac{\partial n(p, \xi)}{\partial \xi} = 0, \quad \lim_{\xi \to -\infty} n(p, \xi) = n_l(p), \quad \lim_{\xi \to +\infty} n(p, \xi) = n_r(p). \tag{189}$$

We have established in Section 3.5 that GHD is a linearly degenerate hydrodynamics. Hence, according to the discussion in Section 2.5, the solution to the Riemann problem must be composed of contact discontinuities: jumps of the normal modes exactly at the rays corresponding to their velocities. This means that we can just naively solve (189), by requiring $n(p, \xi)$, for every $p$, to be constant in $\xi$ except at the value $\xi = \xi_*(p)$ solving the equation

$$\xi_*(p) = v^{\text{eff}}(p, \xi_*(p)). \tag{190}$$

There, it must have a jump, and assuming that there is a single solution $\xi_*(p)$, there is a single jump for this mode, hence this jump is determined by requiring that it connects the left and right state in order to satisfy the asymptotic conditions in (189). With this, we then set

$$n(p, \xi) = \begin{cases} n_l(p) & (\xi < \xi_*(p)) \\ n_r(p) & (\xi > \xi_*(p)). \end{cases} \tag{191}$$

We thus have a *coupled system of nonlinear integral equations*: given $n(p, \xi)$, representing, as a function of $p$, the state at the ray $\xi$, we evaluate $v^{\text{eff}}(p, \xi)$, a nonlinear functional of $n(\cdot, \xi)$. From this, we set $\xi_*(p)$ by solving (190), and this in turn determines $n(p, \xi)$. In practice, we solve by recursion: we may start with approximating $v^{\text{eff}}(p, \xi)$ by the bare group velocity $E'(p)$, and set $\xi_*(p) = E'(p)$. Inserting this into (191), we get an approximation for the solution $n(p, \xi)$. Using this approximation, we evaluate $v^{\text{eff}}(p, \xi)$, from which we get a new $\xi_*(p)$, from which we evaluate a new $n(p, \xi)$. We repeat until convergence.

Surprisingly, an exact analytic solution was found to this system of nonlinear integral equations in the XXZ quantum chain with a particular initial state [163], and a general solution was obtained in the hard rod gas [132]. However, in general it appears to be necessary to solve the system numerically.

The solution is a continuum of contact discontinuities, and thus gives rise to a continuum of states, one for each ray $\xi$, where typical observables would change in a continuous fashion. See Fig. 13

The picture behind the solution (191), (190) is very simple: the quasiparticle of momentum $p$ carries, on the ray $\xi$, the information of the left (right) initial reservoir if its effective velocity $v^{\text{eff}}(p, \xi)$ on this ray is positive (negative). This can be understood by looking at the *characteristics* associated to $p$, the continuous curve in space-time that is tangent to $v^{\text{eff}}(p, \xi)$ at

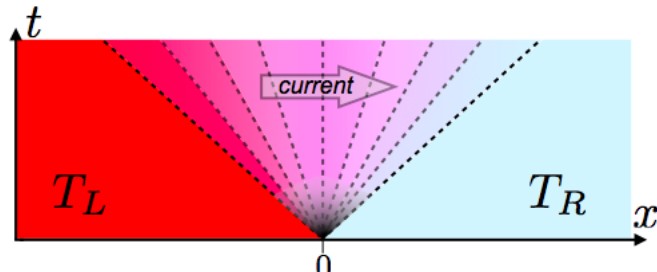

Figure 13: A continuum of states, parametrised by the ray $\xi = x/t$, emerges in the partitioning protocol for integrable systems, where ballistic currents exist.

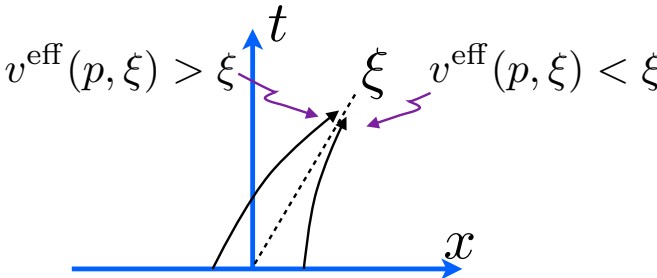

Figure 14: The solution to the partitioning protocol. The state for quasiparticle $p$ is determined by the direction this quasiparticle comes from when it reaches the ray $\xi$.

every point. This characteristics represents the trajectory of a test particle. As the occupation function is convectively transported, it is constant, for fixed $p$, on this curve, thus this curves carries its value from the initial reservoir. The characteristics for a given $p$ cannot cross twice a given ray $\xi$: doing so there would have to be two different values of effective velocities for a given $p$ and $\xi$, but as the state depends only on the ray, a single value may occur. Thus, if the characteristics crosses the ray $\xi$ from the left (right), it must come from the left (right). See Fig. 14.

In fact, in many cases $v^{\text{eff}}(p, \xi)$ is monotonic in $p$, and it is more convenient to write the solution as

$$n(p, \xi) = \begin{cases} n_l(p) & (p > p_\star(\xi)) \\ n_r(p) & (p < p_\star(\xi)), \end{cases} \qquad \xi = v^{\text{eff}}(p_\star(\xi), \xi). \tag{192}$$

The problem of the non-equilibrium steady state is that of looking at the state at $\xi = 0$, and thus for this purpose we may omit the ray $\xi$ and write the condition as $(E')^{\text{dr}}(p_\star) = 0$.

Numerically, in many models a single solution to (190) is found, and the recursive process converges quite fast. The Riemann problem is one of the most successful applications of GHD. A more in-depth analysis of the Riemann problem for GHD is beyond the scope of these notes, see the original papers [1, 2], as well as [140] and especially the extensive analysis done in [104].

## 4.3 Geometric interpretation and solution by characteristics

The characteristic curves have been discussed in Sections 4.1 and 4.2 in order to interpret the equations obtained there. Let me now study these curves in a bit more detail [84]. I will use them in order to obtain a *geometric picture* behind the GHD equations (180) and (182), a *solution by characteristics* to the initial value problem for these equations which generalises

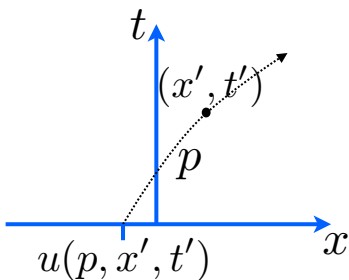

Figure 15: The function $u(p, x, t)$.

the solution presented in 4.2 to arbitrary initial conditions, and an *alternative derivation* of the GHD equations themselves, based on the scattering picture and which does not explicitly use the local entropy maximisation approximation and the completeness of the set of conserved quantities.

The characteristic curve for the quasiparticle of momentum $p$ (the $p$-characteristics), starting at position $u$, in a generically inhomogeneous, non-stationary fluid, is the curve $t \mapsto x(p, u, t)$ tangent to the effective velocity $v^{\text{eff}}(p, x, t)$ at every point:

$$\frac{\partial x(p, u, t)}{\partial t} = v^{\text{eff}}(p, x(p, u, t), t), \quad x(p, u, 0) = u. \tag{193}$$

By the GHD equation (182), the occupation function is constant along this curve:

$$\frac{\partial n(p, x(p, u, t), t)}{\partial t}\bigg|_{p,u} = \left[ \partial_t n(p, x, t) + v^{\text{eff}}(p, x, t) \partial_x n(p, x, t) \right]_{x=x(p,u,t)} = 0. \tag{194}$$

Let me now consider the starting point $u$ of the characteristic curve, and invert the function $x(p, u, t)$ as

$$u = u(p, x, t), \quad x(p, u(p, x, t), t) = x. \tag{195}$$

[By abuse of notation, I use the same symbol for the function $x(\cdots)$ and the variable $x$.] The function $u(p, x, t)$ is the starting point of the $p$-characteristics which crosses the spacetime point $x, t$, see Fig. 15. In order to find it, starting at $x, t$ we go backwards along the $p$-characteristics until we reach time 0. As a consequence of the invariance of $n$ along the characteristic curve, we have

$$n(p, x, t) = n(p, u(p, x, t), 0). \tag{196}$$

Thus, if we can determine $u(p, x, t)$, this provides a solution to the initial value problem for GHD. This is the solution by characteristics.

The function $u(p, x, t)$ satisfies the same equation as $n(p, x, t)$. This is simple to see from (196), just imposing (182). One can also derive this from the definition of $u(p, x, t)$ itself. Indeed using (193), we find

$$0 = \frac{\partial x(p, u(p, x, t), t)}{\partial t}\bigg|_{p,x} = \left[ \partial_t u(p, x, t) \partial_u x(p, u, t) + v^{\text{eff}}(p, x(p, u, t), t) \right]_{u=u(p,x,t)} \tag{197}$$

and

$$1 = \frac{\partial x(p, u(p, x, t), t)}{\partial x}\bigg|_{p,t} = \partial_x u(p, x, t) \partial_u x(p, u, t)\big|_{u=u(p,x,t)}. \tag{198}$$

Combining these,

$$\partial_t u(p, x, t) + v^{\text{eff}}(p, x, t) \partial_x u(p, x, t) = 0. \tag{199}$$

Its initial condition is

$$u(p, x, 0) = x. \tag{200}$$

In free models, the effective velocity does not depend on the state, and is equal to the group velocity $v(p) = E'(p)$. Hence it is constant in space-time, and $u(p, x, t) = x - v(p)t$. The solution $n(p, x, t) = n(p, x - v(p)t, 0)$ is just that of a freely propagating, non-dispersive wave. In interacting fluids, however, it is in general not possible to explicitly solve for $u(p, x, t)$.

Surprisingly, in GHD, there is an *integral equation* that determines $u(p, x, t)$. Suppose that the state of the fluid asymptotically very far on the left does not depend on space-time (the left is just a choice, I could use the right as well) – it is asymptotically homogeneous, hence does not evolve. Let me define

$$\hat{v}(p) = 2\pi \rho_s(p, -\infty, 0) v^{\text{eff}}(p, -\infty, 0). \tag{201}$$

Then for $u$ and $x$ related as in (195), we have

$$2\pi \int dy \left[ \rho_s(p, y, 0) \Theta(u - y) - \rho_s(p, y, t) \Theta(x - y) \right] + \hat{v}(p) t = 0. \tag{202}$$

Recall that $2\pi \rho_s(p, x, t) = 1^{\text{dr}}(p, x, t)$. For free models, $\hat{v}(p) = v(p)$ and $\rho_s$ is independent of $x, t$, so that (202) reproduces $u - x + v(p)t = 0$. In a homogeneous, stationary state, the solution is also simply $u = x - \hat{v}(p)t$, representing the linear propagation of quasiparticles. In inhomogeneous, interacting models, however, (202) is a nontrivial integral equation relating $u$ and $x$.

Before showing (202), I note that *the integral equation (202) combined with the characteristic curve equation (196) provide a system of equations solving the GHD initial value problem*. Indeed, in (202), only the state at $t$ and at 0 is required, and the time $t$ appears explicitly as a parameter. Thus, in principle, one can solve for $n(p, x, t)$ by recursion. For instance, set as a first approximation $n(p, x, t) \approx n(p, x - v(p)t, 0)$, then use this in (202), which must be solved numerically, in order to obtain an approximation for $u(p, x, t)$, and insert this in (196) to obtain the next approximation for $n(p, x, t)$. Repeating, the recursive process has been observed to converge in the cases studied [84].

In order to show (202), I will simply show that it leads to (199) with initial condition (200). The initial condition is in fact immediate. In order to show (199), I use the fact that $\rho_s$ satisfies the conservation law (186). Differentiating (202) with respect to $t$ at $p, x$ fixed, this gives

$$2\pi \rho_s(p, u, 0) \partial_t u + 2\pi v^{\text{eff}}(p, x, t) \rho_s(p, x, t) = 0, \tag{203}$$

where I used (186) and performed the $y$-integral of the total derivative that arose, cancelling the term $\hat{v}(p)$. Differentiating with respect to $x$ at $p, t$ fixed,

$$2\pi \rho_s(p, u, 0) \partial_x u - 2\pi \rho_s(p, x, t) = 0. \tag{204}$$

Combining, I indeed get (199).

What is the meaning of (202)? In order to understand, let me define, for every $p$, a new space coordinate $\hat{x}$ related to $x$ as

$$d\hat{x} = 2\pi \rho_s dx. \tag{205}$$

The change of coordinate $x \to \hat{x}$ depends, in general, both on the momentum $p$ of the quasiparticle considered, and on time $t$. Integrating (205), we will write $\hat{x}(p, t)$. With this, Eq. (202) is

$$\hat{u}(p, 0) - \hat{x}(p, t) + \hat{v}(p)t = 0. \tag{206}$$

That is, in this new coordinate system, forgetting about the special $t$ dependence of the transformation, $\hat{u} - \hat{x} + \hat{v}t = 0$: this is of the form of the free-particle solution for the characteristic curve! In fact, these coordinates *trivialise* the fluid equation. Defining $\hat{n}(p, \hat{x}, t)$ by

$$\hat{n}(p, \hat{x}(p, t), t) = n(p, x, t), \tag{207}$$

we have

$$\partial_t \hat{n}(p, \hat{x}, t) + \hat{v}(p) \partial_{\hat{x}} \hat{n}(p, \hat{x}, t) = 0. \tag{208}$$

Indeed, $\hat{n}(p, \hat{x}(p, t), t) = n(p, x, t) = n(p, u, 0) = \hat{n}(p, \hat{u}(p, 0), 0) = \hat{n}(\hat{x}(p, t) - \hat{v}(p)t, 0)$. The occupation function, written in the $\hat{x}$ coordinates, evolves trivially.

The change of coordinates induces a *change of metric* between $x$ and $\hat{x}$. How do we interpret this change of metric? For this purpose, let me go back to the discussion just after Equation (110). There, it was observed that $2\pi\rho_s = R/L$ is the ratio of the lengths perceived in the space of asymptotic impact parameters, to the actual lengths where the real particles lie. The ratio is the effective available space density in which quasiparticles move. Because they are affected by scattering shifts, this effective space density is different from one. Thus, the coordinate $\hat{x}$ should be interpreted as the coordinate for the space perceived by the quasiparticles. Equivalently, it is the coordinate where lie the impact parameters of the scattering map.

The fact that the change of coordinates $x \mapsto \hat{x}$ trivialises the GHD equation is then interpreted as the fact that the asymptotic coordinates of scattering theory satisfy trivial evolution equations (90), that of free particles. Indeed, the fluid equations for free particles are nothing else but the *Liouville equations*, or collisionless Boltzmann equations, for their phase-space density, which state that phase-space elements are preserved. And the occupation function (127) is, by construction, the density of quasiparticle per unit available quasiparticle space, so it is the density in asymptotic phase space. Thus, GHD is the trivial fluid equations for the freely evolving "asymptotic particles", whose coordinates are the asymptotic momenta and impact parameters.

This gives a very powerful interpretation of GHD. In two statements: (1) *GHD is the fluid equation obtained by applying the inverse of the scattering map (87) to the Liouville equations*; and (2) *in integrable systems, the inverse scattering map is a change of metric, which takes into account the accumulation of scattering shifts*. In a "meta-equation", this is

$$\text{GHD} = S_{\text{in}}^{-1} \circ \text{Liouville equation}. \tag{209}$$

In this sense, it is perhaps appropriate to refer to the GHD equations as "Bethe-Liouville equations" (the name "Bethe-Boltzmann equations" has been proposed [48], before the present understanding came to light). This understanding of GHD gives an alternative derivation of the GHD equations, which does not require the assumption of local maximisation of entropy towards GGEs. One starts with the Liouville equation in asymptotic space, and simply applies the change of coordinates corresponding to the accumulation of scattering shifts. The above results show that the GHD equations are obtained in this way. The full thermodynamics, including the Lagrange parameters, can likewise be obtained in this way. Interestingly, I believe this is a blueprint for a *rigorous derivation* of GHD; indeed it is essentially such ideas, expressed in different terms, that are used in order to show the fluid equations in the hard rod gas [77].

**Remark 4.2.** *In (202), it appears as though we could use $\rho_s r(n)$ for any positive function $r(n)$ of the occupation function, in place of $\rho_s$, without changing the result for the differential equation (thanks to (185)). However, the resulting integral equation may not be as well defined, because $n$ may vanish [but this has not been analysed in any depth yet]. Likewise, the metric interpretation also would fail with this replacement.*

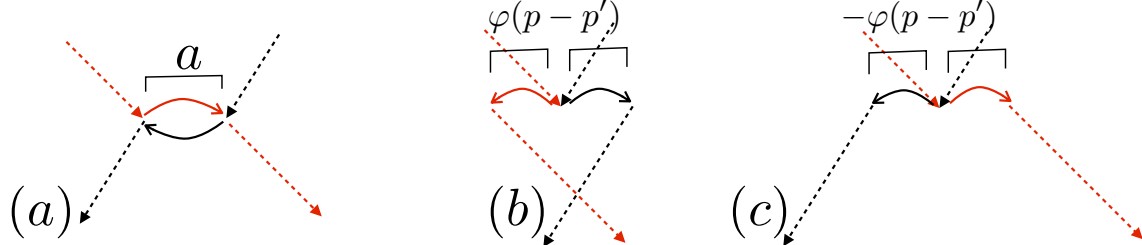

Figure 16: (a) The quasiparticle jump in the hard rod case, see Fig. 9. (b) The flea-gas quasiparticle jump, for $\varphi(p-p') > 0$ (backward jump). (c) The flea-gas quasiparticle jump, for $\varphi(p-p') < 0$ (forward jump).

## 4.4 The flea gas algorithm

I have explained in the previous section how the viewpoint of the accumulation of phase shifts, advocated in Sections 3.1 and 3.2 and at the basis of the TBA technology, gives a simple geometric meaning to the GHD equations themselves. This provides a set of coupled integral equations which "solve" the initial value problem.

In this section, I use this viewpoint in order to construct an algorithm that solves GHD [160]. The algorithm is a *classical molecular dynamics*, that is, a classical gas of interacting particles whose trajectories we directly implement on the computer. There is such a gas for any GHD equation (180), no matter the form of the scattering shift $\varphi(p)$. That is, this algorithm actually shows that there are *classical gases that have the same Euler hydrodynamics as quantum gases*. Because of its particular features, it is referred to as the **flea gas**.

The point of the algorithm – of the molecular dynamics – is to reproduce the displacement equation (150) which we argued, in Section 3.4, could be used as an interpretation of the formula for the effective velocity. The idea is a generalisation of old ideas used for the hard rod gas (see Example 3.4), where the displacements are now made momentum-dependent. However, the algorithm is not a hard-rod type of algorithm, and does not specialise to it in the case of a constant scattering shift (momentum-independent displacements). Nevertheless, it reproduces the same Euler hydrodynamics (there is a lot of universality at large scales).

Let me imagine a gas of particles, each with a momentum label $p$. I wish to identify these particles with the quasiparticles constructed abstractly in Section 3.1 as the velocity tracers.

Suppose each particle travels freely, between collisions, at a velocity $v(p)$. This part of the algorithm reproduces the linear displacements, the first term on the right-hand side of (150). Here, contrary the case of real quasiparticles, from a real model such as the Toda model, there is true linear displacement between collisions; but this is just an imagined gas, an algorithm.

In order to reproduce the jumps that quasiparticles should undergo, I indeed impose the particles of the gas to jump, instantly, by the required distance, in the required direction, every time there is collision. The distance and direction is specified in the second term on the right-hand side of (150). See Fig. 16.

At this point, however, two subtleties arise. First, if $\varphi(p) > 0$, then the jump "adds space", i.e. is backward, towards the directions where the particles come from. If from there the particles continue travelling linearly, as they should, they will meet soon again, and, naively, would jump backward again, never getting through each other. This is not right – a single jump should occur at each crossing. So, we let the particle have a "memory": they exchange a card to say they've already met, and when they meet again, they go through each other without jumping.

The second subtlety is more delicate. When a particle $p$ tries to jump by the amount $\varphi(p-p')$ after meeting another particle $p'$, this jump might be large enough to cross *other*

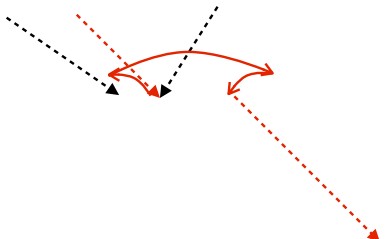

Figure 17: In a flea-gas jump trajectory, another particle is met. A second jump must then be executed, before finishing the first one. Here all scattering shifts are positive.

*particles* on its way. Not considering these, there would be crossings without a scattering shifts, breaking (150). Hence, if the set of particles $p_1, p_2, \ldots p_N$ is on the path of the jump of particle $p$ after it met $p'$, then each crossing $(p, p_n)$ $(n = 1, 2, \ldots, N)$ must be considered a collision. In order to understand these collisions, imagine that, even if the jump is instantaneous, we stop time, and look at the trajectory the particle $p$ takes during its jump. This trajectory has a specific direction, coming from where the original collision was; these are the curly arrows in Fig. 16. Then, the meeting $(p, p_1)$ of particle $p$ with the closest of particle (say $p_1$) on its jump path should be understood as a collision, of the same type as above. Thus, this occasions another jump to be executed, of a distance and direction again determined by (150). Note how, crucially, the direction of this "inside-jump" jump is *not* determined by the direction of linear travel, $v(p)$, but rather by the direction the particle is taking during its jump. This guarantees that the directions of the jumps are determined by those of crossings, as it should for (150). See Fig. 17.

Naturally, any "inside-jump" jump may also cross other particles, and require further jumps. Thus, we arrive at a *recursive procedure*, in which the jump procedure may call itself in order to execute "inside-jump" jumps.

The result is a chain reaction of jumps at every collision, reminiscent of fleas jumping around and encouraging their neighbours to jump as the pass above (I do not know if this is an actual socio-biological effect in the flea).

**Remark 4.3.** *It is important to note that the flea gas algorithm is* not time-reversal invariant*, and that it* does not immediately reproduce the thermodynamics *with arbitrary free energy function $F(\epsilon)$ (see Section 3.3). In particular, it does not reproduce, in general, the correlation functions or the diffusion operator discussed in Section 4.6. It is, for now, limited to solving the Euler-scale equation. It* does*, however, work with simple external force terms, see Remark 4.4. In more general situations, the algorithm still needs to be studied.*

## 4.5 External force

Equations (180), and equivalently (182), describe how the fluid of a many-body integrable system evolves in space-time. This is for a microscopic evolution according to some homogeneous Hamiltonian $H$, for instance that of the Toda gas (85).

However, in many situations of high interest experimentally, the microscopic evolution includes an *external force field*, which is often inhomogeneous. The most important example is the quantum Lieb-Liniger model in an external potential,

$$H = -\frac{1}{2} \sum_n \partial_{x_n}^2 + g \sum_{n<m} \delta(x_m - x_n) + \sum_n V(x_n), \tag{210}$$

see Example 3.5. Written in second quantisation, this takes the form

$$H = \int dx \left[ \frac{1}{2} \partial_x \Psi^\dagger(x) \partial_x \Psi(x) + \frac{g}{2} \Psi^\dagger(x) \Psi^\dagger(x) \Psi(x) \Psi(x) + V(x) \Psi^\dagger(x) \Psi(x) \right] \tag{211}$$

for a canonical Bosonic field, $[\Psi(x), \Psi^\dagger(y)] = \delta(x - y)$. The function $V(x)$ is the external potential, and the associated force is $-V'(x)$.

The conserved number of particles is

$$Q_0 = \int dx \, \Psi^\dagger(x) \Psi(x). \tag{212}$$

However, with an external potential that is inhomogeneous, the momentum charge is broken because there is no translation invariance anymore, and in fact all higher conserved charges, beyond the Hamiltonian, are broken. That is, *the system is not integrable anymore*! Can we still use the theory of GHD in order to describe what happens at large scales in such a situation?

The answer is yes, under the condition that the potential $V(x)$ vary on large enough scales. The situation is not so different from that of conventional hydrodynamics in an external force field, such as gravity or an external electric or magnetic field. There, although momentum is broken by an external inhomogeneous field, we can still write Euler equations (9) for local densities of particles and momentum, modified to include a force term. The momentum conservation equation, the second line of (10), receives a correction due to Newton's equation telling us how momentum changes. That is, locally, we still have fluid cells that maximise entropy with respect to all the usual conserved charges including momentum (that is, the fluid cells can still have a nontrivial velocity); it is the large-scale equations that break momentum.

Hence, we apply the same principles to GHD, and we assume that the breaking of conserved charges (be it the momentum, or the higher conserved charges) occurs only on large enough scales, and hence *does not* preclude local entropy maximisation to GGEs. That is, on the fluid cell at position $x$, the evolution with the microscopic Hamiltonian (211) is equivalent to that of the homogeneous Hamiltonian, modified by the presence of the chemical potential $-V(x)$ [the force potential is the negative of the chemical potential, in the usual conventions],

$$H(x) = H_{\text{LL}} + V(x) Q_0, \tag{213}$$

where

$$H_{\text{LL}} = \int dy \left[ \frac{1}{2} \partial_y \Psi^\dagger(y) \partial_y \Psi(y) + \frac{g}{2} \Psi^\dagger(y) \Psi^\dagger(y) \Psi(y) \Psi(y) \right] \tag{214}$$

is the homogeneous part of the Lieb-Liniger Hamiltonian. We thus have to figure out how to evolve the locally entropy-maximised states in time with a Hamiltonian that contains space-dependent couplings to the conserved charges.

Of course, $H(x)$ is part of the integrable hierarchy for any value of $V(x)$. That is, it has a local energy function, now $x$-dependent, which we can write within our formalism,

$$E(p, x) = \frac{p^2}{2} + V(x). \tag{215}$$

In general, we can do the same with couplings to all other conserved charges, for instance

$$H(x) = H_{\text{LL}} + \sum_i V^i(x) Q_i. \tag{216}$$

This will be associated to a local energy function

$$E(p, x) = \frac{p^2}{2} + \sum_i V^i(x) h_i(p). \tag{217}$$

We can therefore simply assume a local energy function that depends on both $p$ and $x$, in some given way.

How is the Euler-scale hydrodynamic equation modified by such a space-dependent Hamiltonian? Certainly, we can calculate the currents in the local fluid cells, and they have the same form as before, (145), but with an effective velocity that knows about the $x$-dependence of the Hamiltonian, as it uses the local energy function at $x$,

$$v^{\text{eff}}(p, x, t) = \frac{(E'(\cdot, x))^{\text{dr}}(p, x, t)}{1^{\text{dr}}(p, x, t)}. \tag{218}$$

But this does not tell us how, at large scales, the charges are broken by the presence of a force; e.g. how the momentum is not conserved anymore. For this, we need an additional *force term* in the GHD equations (180) and (182). It turns out that this force term can be written explicitly [47], although the calculation is quite involved, and I don't have a simple, direct argument leading to it. The result is simple, however, and takes the form

$$\partial_t \rho_{\text{p}} + \partial_x \big[ v^{\text{eff}} \rho_{\text{p}} \big] + \partial_p \big[ a^{\text{eff}} \rho_{\text{p}} \big] = 0 \tag{219}$$

in terms of quasiparticle densities, and

$$\partial_t n + v^{\text{eff}} \partial_x n + a^{\text{eff}} \partial_p n = 0 \tag{220}$$

in terms of the occupation function, where the **effective acceleration** is the "effectivisation" of the acceleration,

$$a^{\text{eff}}(p, x, t) = \frac{\big( -\partial_x E(\cdot, x) \big)^{\text{dr}}(p, x, t)}{1^{\text{dr}}(p, x, t)}. \tag{221}$$

It turns out that it satisfies an equation similar to (146), but with the acceleration as source term (here omitting the $t$ dependence),

$$a^{\text{eff}}(p, x) = -\partial_x E(p, x) + \int \mathrm{d}p' \, \varphi(p - p') \rho_{\text{p}}(p', x)(a^{\text{eff}}(p', x) - a^{\text{eff}}(p, x)). \tag{222}$$

It is noteworthy that for the simple and most physical case of (215), the effective acceleration *simplifies to the usual acceleration*,

$$a^{\text{eff}}(p, x) = -\partial_x V(x) \qquad \text{(case (215))}. \tag{223}$$

More general force terms were proposed in [164, 165] in order to describe space-time variations of the *interaction terms* in the Hamiltonian (that is, space-time dependent, integrable Hamiltonians that do not stay within a given integrable hierarchy), where special effects such as sudden entropy increase can occur due to the features of the spectrum of quasiparticles.

**Remark 4.4.** *We note that, with an external potential associated to the number of particles and with (215), the flea gas algorithm described in Section 4.4 can be implemented, simply by adding an acceleration to the flea's linear evolution, without changing the jump procedure.*

## 4.6 Correlation functions and diffusion

Finally, it is possible to apply the general ideas of Sections 2.4 and 2.6 to the cases of integrable models, as done in [125, 140] and [49, 123, 166].

I start with the discussion of correlation functions, Section 2.6. The general result for the two-point function of conserved densities is expressed in (80). At the Euler scale, we

can set $\mathcal{D}_i^{\ j} = 0$ in this equation, and from Section 3.5, we already know all the necessary hydrodynamic matrices in the TBA framework. Putting things together, one finds

$$S_{ij}(k,t) = \int dp\, \rho_p(p) f(p) e^{iktv^{eff}(p)} h_i^{dr}(p) h_j^{dr}(p) \tag{224}$$

and, Fourier transforming back to real space,

$$\begin{aligned}
\langle q_i(x,t) q_j(0,0)\rangle^{eul} &= \int dp\, \rho_p(p) f(p) \delta(x - v^{eff}(p)t) h_i^{dr}(p) h_j^{dr}(p) \\
&= t^{-1} \sum_{p \in P(x/t)} \frac{\rho_p(p) f(p) h_i^{dr}(p) h_j^{dr}(p)}{\left| \partial v^{eff}/\partial p \right|},
\end{aligned} \tag{225}$$

where $P(x/t) = \{p : v^{eff}(p) = x/t\}$ is the set of momenta for which the mode propagates along the ray $x/t$.

It is also possible to go further and work out the formula (82) from hydrodynamic projections. The necessary ingredients are the overlaps $(q_i, o)$. These are known as soon as the averages $\langle o \rangle_{\underline{\beta}}$ are known as functions of all Lagrange parameters $\beta^i$. For convenience, assume that this is known, and write it in the form (see [140])

$$-\frac{\partial \langle o \rangle_{\underline{\beta}}}{\partial \beta^i} = (q_i, o) = \int dp\, \rho_p(p) f(p) h_i^{dr}(p) V^o(p) \tag{226}$$

for some function $V^o(p)$. This parallels the form (158) for $(q_i, q_j)$. Then the result is

$$\langle o_1(x,t) o_2(0,0)\rangle^{eul} = \int dp\, \rho_p(p) f(p) \delta(x - v^{eff}(p)t) V^{o_1}(p) V^{o_2}(p). \tag{227}$$

In particular, for the currents, $V^{j_i}(p) = v^{eff}(p) h_i^{dr}(p)$, as per (159).

The quantities $V^o(p)$ can be evaluated for various fields, see [140]. The very general result (227) was recently shown from a thermodynamic spectral decomposition [110, 167].

It is important to note that these correlation functions decay generically as $1/t$ in the region of rays $x/t$ where the effective velocity ranges. This is a particularity of integrable systems: instead of a smaller power law (slower decay) at specific rays and an exponential decay otherwise, one finds that the continuum of normal modes give a larger power law (1, or it may be larger along certain rays for certain observables) within the full range of the effective velocity. Of course, the specialisation of the Euler-scale correlation function to the Drude weight is then immediate: one simply needs to integrate the current-current correlation function, as per (35), re-obtaining (163).

Finally, the diffusion operator can also be evaluated. This is much more involved, as it requires a more in-depth understanding of the structure of correlation functions in integrable models. The calculation in [49, 123] uses a spectral decomposition, which can be framed within the thermodynamic spectral decomposition of [110, 167]. The result can be written for the Onsager matrix in the form

$$\begin{aligned}
\mathcal{L}_{ij} &= \int \frac{dp_1 dp_2}{2} \rho_p(p_1) f(p_1) \rho_p(p_2) f(p_2) |v^{eff}(p_1) - v^{eff}(p_2)| T^{dr}(p_1, p_2) \times \\
&\quad \times \left[ \frac{h_i^{dr}(p_2)}{\rho_s(p_2)} - \frac{h_i^{dr}(p_1)}{\rho_s(p_1)} \right] \left[ \frac{h_j^{dr}(p_2)}{\rho_s(p_2)} - \frac{h_j^{dr}(p_1)}{\rho_s(p_1)} \right].
\end{aligned} \tag{228}$$

This is obviously a non-negative, symmetric matrix, as it should. Although this form was derived in the quantum context, when specialised to classical systems, with the Boltzmann

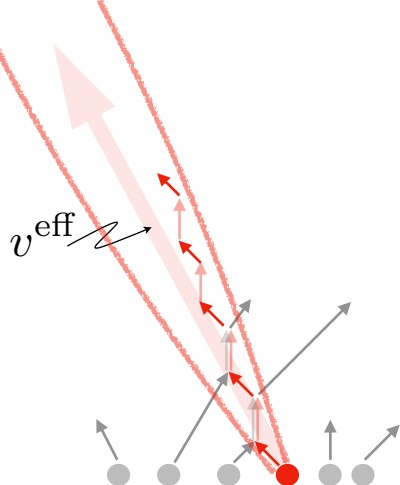

Figure 18: The trajectory of a test quasiparticle within the gas spreads diffusively around its velocity $v^{\text{eff}}$.

statistical factor $f(p) = 1$, it correctly specialises to the form derived rigorously in the hard rod gas [78]. This is therefore expected to be completely general.

The proposed physical picture behind diffusion in integrable systems is that the trajectory of a test quasiparticle within a gas of quasiparticles is not exactly the line at $v^{\text{eff}}$, but rather spreads diffusively because of the random collisions, see Fig. 18. This picture is well known from the hard rod model. It was first linked with the (diagonal elements of the) diffusion constant in [166], and has now been made quite precise thanks to the understanding of ballistic wave scattering in hydrodynamics [134, 168]. It is important to emphasise that although there is diffusion, there *is no external noise* (usually) in integrable system: the randomness normally associated with diffusion comes from the random initial condition.

## Chapter 5   Closing remarks

In these notes, I have tried to provide a pedagogical overview of the subject of generalised hydrodynamics (GHD). Although GHD grew out of describing many-body quantum systems out of equilibrium, it is now seen as applicable to a very wide array of many-body systems, both quantum and classical. It connects nicely many previous studies, extracting the main physical principles and their structures. In fact, one important outcome of GHD has been to push the study of the general theory of hydrodynamics. It has put the emphasis on the general relations between conserved quantities and the emerging large-scale physics, in contrast with the particular hydrodynamic notions based on density of particles, momentum and energy traditionally used.

The GHD equations are similar to the Liouville equations, or collisionless Boltzmann equations. I have explained the origin of this in Section 4.3: the GHD equations come from applying a scattering map to the Liouville equations, using the special properties of integrable systems. It is important to note that the GHD equations are not based on molecular chaos, as Boltzmann equations typically are, but local entropy maximisation at large wavelengths, as Euler equations are. The GHD equations were also extended to include diffusive contributions, Section 4.6.

It is likely that GHD can be pushed further than the diffusive level, possibly using form factor techniques as in [49, 123], or the more general techniques of [134, 168]. A possible picture

is that as finer and finer hydrodynamic scales are accessed (the expansion is pushed to higher and higher derivatives), the quasiparticle density $\rho_{\rm p}(p)$ morphs into the real-particle phase space density of the Boltzmann equation. This is to be contrasted with generic, conventional hydrodynamics, where it is usually believed that beyond the diffusive scale, the hydrodynamic approximation, under which the number of degrees of freedom is reduced to the conserved quantities, fails. In integrable models, one might indeed expect that the derivative expansion up to infinite order stays meaningful, at least as an asymptotic expansion. The reduction of the number of degrees of freedom should stay valid at all inverse powers of the variation lengths, because the scattering map gives a position of quasiparticles up to imprecisions of order of the scattering length, that is, of order 1 with respect to the thermodynamic limit.

Another aspect of GHD which would be interesting to investigate is the importance of the choice of vacuum in the scattering description. The choice of the vacuum affects the spectrum of asymptotic excitations, see the beginning of Section 3.1 and Remark 3.1. Thus, for any given system and state, there may be many different-looking TBA formulations. One example is the dual bosonic and fermionic descriptions of the Lieb-Liniger model. How does this affect the physical intuition behind GHD?

Although GHD pertains to the domain of integrable systems, in one dimension of space, one can argue (see e.g. [146]) that generic, non-integrable systems at low density follow to a good approximation the equations of GHD. Indeed, at low densities, two-body scattering is sufficient in order to describe the dynamics, and in one dimension, energy and momentum are conserved under two-body scattering. Thus, the notion of quasiparticle makes sense, and the general arguments leading to the TBA technology apply. This is probably an important reason for which integrable systems are so successful in reproducing many of the observations made in quasi-one-dimensional experimental setups. It is also the fundamental reason behind the Fermi-Pasta-Ulam-Tsingou results [169]: when a classical field, or anharmonic chain, is excited only with large-wavelength configurations, then it has a low density of asymptotic excitations [note that this is *different* from exciting a classical field with a large-wavelength distribution of configurations that may themselves be very rough; this leads to hydrodynamics].

The above in fact points to the crucial, more general problem of understanding how perturbations of integrability affect the non-equilibrium dynamics; this is essential, it can be argued, in order to have full theoretical understanding of experiments on near-integrable systems.

One area of research in GHD that needs further studies is that of classical integrable field theory. This has, for instance, potential application to low-temperature cold atomic systems, where an integrable classical field theory (the nonlinear Schrödinger equation) describes the pseudo-condensate wave function. Results are also much more easily compared to numerical simulations. In classical field theory, there is the well-known UV catastrophe. However, as emphasised in [141], higher conserved charges in one dimension cure this. It would be interesting to analyse the physical consequences further.

In many integrable models, GHD, as currently framed, is not the full story. As GHD concentrates on the commuting Hamiltonian flows of integrable systems, it does not account very well for the internal symmetries that may be available. It would be important to elucidate how these can be described efficiently, complementing the GHD framework. In particular, it is likely that, in models where scattering is non-diagonal (internal degrees of freedom are mixed in scattering events), there is some emerging conventional, non-integrable type of hydrodynamics for these degrees of freedom, on top of the GHD hydrodynamics for quasiparticles. This would connect with some recent work on spin degrees of freedom in spin chains [139].

As noted, see Section 3.5, GHD is a linearly degenerate hydrodynamic system. In such systems, shocks do not develop. An important question is as to the connection between GHD, and in particular the TBA technology, and generic linearly degenerate hydrodynamic equations. Are the hydrodynamic properties of integrable systems actually entirely properties of

linear degeneracy? Could we adapt the GHD geometric picture, or the exact results on correlations and diffusion, to more general linearly degenerate hydrodynamics?

Finally, one potential area of extension is to higher-dimensional systems. One may devise higher-dimensional gases with appropriate factorisation properties as in the hard rod gas or flea gas algorithm. It would be interesting to see how this may connect to, and inform, the theory of higher-dimensional integrability.

## Problems

Chap. 2. A relativistic, conformal fluid in $d$ dimensions of space ($d + 1$ space-time dimensions) can be restricted to a one-dimensional fluid if we only look at unidirectional flows. Two conserved quantities are relevant: the energy $H = Q_1$, and the momentum $P = Q_2$ in the flow's direction. The maximal entropy states are characterised by a rest-frame temperature $T$ and a boost $\theta$, as $\rho = e^{-(H \cosh \theta - P \sinh \theta)/T}$. The densities and currents take the form

$$\mathsf{q}_1 = aT^{d+1}(d \cosh^2 \theta + \sinh^2 \theta)$$
$$\mathsf{j}_1 = \mathsf{q}_2 = a(d+1)T^{d+1} \cosh \theta \sinh \theta$$
$$\mathsf{j}_2 = aT^{d+1}(\cosh^2 \theta + d \sinh^2 \theta),$$

where $a$ is a model-dependent constant.

a. Evaluate the matrices A, B and C, and the effective velocities for this fluid. Show that the effective velocities are given by $\pm 1/\sqrt{d}$ when the fluid is at rest, $\theta = 0$. How does A simplify in the case $d = 1$?

b. Show that $n_\pm = Te^{\pm \theta/\sqrt{d}}$ are normal modes.

c. Show that the free energy flux is $\mathsf{g} = -aT^d \sinh \theta$. Calculate the free energy density f.

Chap. 3. a. By directly using the equations of motion, and going to the centre-of-mass frame, show that the scattering shift of the Toda model is $\varphi(p) = 2 \log |p|$.

b. In the general framework of TBA with a Galilean invariant, single quasiparticle spectrum, show that the entropy flux $\mathsf{j}_s = \sum_i \beta^i \mathsf{j}_i - g$ takes the form

$$\int dp \, v^{\text{eff}}(p)\rho_s(p)\big[\epsilon(p)n(p) - F(\epsilon(p))\big].$$

You can try without looking at the proposed derivation in Eq. (154); if you use this derivation, simply give the justification for each of the steps shown there.

c. Consider again the framework of TBA with a Galilean invariant, single quasiparticle spectrum. What functions $h_0(p)$ and $h_1(p)$ should be used in order to represent the density of particles, and the density of momentum, respectively? Show that $\int dp \, p\rho_p(p) = \int dp \, v^{\text{eff}}(p)\rho_p(p)$ and interpret this result in view of your answer to this question.

d. Consider a Galilean invariant quantum integrable model with a single quasiparticle type of fermionic statistics, with scattering shift $\varphi(p) = 4/(p^2 + 4)$. Suppose the state is determined by $w(p) = w_T(p) = (p^2/2 - 1)/T$ where $T > 0$ is the temperature. Evaluate numerically (using Mathematica, or Python, or any other computer language), the average density and the average energy for various temperatures (say 5 values) in the range $T \in [0, 2]$. Also plot the occupation function $n(p)$. What happens to the latter when $T \to 0$?

Chap. 4.    a. The hard rod gas is a classical gas of rods, each of length $a > 0$, with elastic colli­sions. With quasiparticles being velocity tracers as usual, the TBA framework and GHD can be applied, with $\varphi(p) = -a$ and the classical particle statistics. Show that the effective velocity simplifies to

$$v^{\text{eff}}(p, x, t) = \frac{p - a\rho(x,t)u(x,t)}{1 - a\rho(x,t)},$$

where $\rho(x, t) = \int \mathrm{d}p\, \rho_{\mathrm{p}}(p, x, t)$ is the total density per unit length at the fluid cell $x, t$, and

$$u(x, t) = \frac{1}{\rho(x,t)} \int \mathrm{d}p\, p\rho_{\mathrm{p}}(p, x, t)$$

is the average velocity at the fluid cell $x, t$. Then, consider the solution to the par­titioning protocol, with left and right initial states determined by the occupation functions $n_{\mathrm{l}}(p)$ and $n_{\mathrm{r}}(p)$, respectively. Recall that the solution $n(p, \xi)$ has a dis­continuity, as a function of $p$ for $\xi$ fixed, at $p = p_*(\xi)$. Show that the condition for the momentum at which discontinuity occurs, $v^{\text{eff}}(p_*(\xi), \xi) = \xi$, reduces to

$$p_*(\xi) = g^{-1}(\xi), \quad g(p) = p + a \int_p^\infty \mathrm{d}p'(p - p')n_{\mathrm{r}}(p') + a \int_{-\infty}^p \mathrm{d}p'(p - p')n_{\mathrm{l}}(p').$$

b. Consider a Galilean invariant quantum integrable model with a single quasiparti­cle type of fermionic statistics, with scattering shift $\varphi(p) = 4/(p^2 + 4)$, as in Part d., Chapter 3 above. Suppose an initial state of the partitioning protocol is deter­mined by $w_{T_{\mathrm{l}}}(p)$ and $w_{T_{\mathrm{r}}}(p)$, for $T_{\mathrm{l}} > T_{\mathrm{r}}$ the temperatures on the left and the right, where $w_T(p) = (p^2/2 - 1)/T$. Using the solution method that we obtained in the class, evaluate numerically, and plot, the energy current and the energy density as functions of the ray $\xi$ for, say $T_{\mathrm{l}} = 2$ and $T_{\mathrm{r}} = 0.5$.

c. Show that the equations for GHD in a force fields in terms of the quasiparticle density (219), and in terms of the occupation function (220), are equivalent. For this, it is sufficient to start with (219), replace in it the expressions for the effective velocity and acceleration in terms of dressed quantities, and apply the derivatives on this.

d. Consider a system with a Galilean invariant, single particle spectrum, in some ex­ternal force field. There is thus some energy function $E(p, x)$, whose form I leave undetermined. Consider a *statrionary* solution (independent of time) for the GHD equation with acceleration, and consider the source term $w(p, x)$ associated to this solution. Show that it satisfies

$$\frac{\left(\partial_x w(\cdot, x)\right)^{\mathrm{dr}}(p, x)}{\left(\partial_x E(\cdot, x)\right)^{\mathrm{dr}}(p, x)} = \frac{\left(w'(\cdot, x)\right)^{\mathrm{dr}}(p, x)}{\left(E'(\cdot, x)\right)^{\mathrm{dr}}(p, x)}$$

(where, as usual, $' = \partial/\partial p$ is the momentum derivative). From this, deduce that $w(p, x) = \beta E(p, x)$ is a solution for any $\beta$, and interpret this result.

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
