# Peer review of "Lecture Notes On Generalised Hydrodynamics"

_SciPost Physics Lecture Notes, doi:SciPost Phys. Lect. Notes 18 (2020)_

## Round 1 · Referee Report · Anonymous (Referee 1) · 2020-6-9

Report
This manuscript contains a very valuable introduction to the theory of generalized hydrodynamics (GHD). The exposition is very well organized, focusing first on hydrodynamics, then on Bethe Ansatz, and finally on GHD, which draws from both previous subjects. This logical organization is made explicit in the text, helping the reader to follow the development of the theory more easily.
Overall, I find the manuscript to be remarkably well written. The exposition benefits from a steady pace, never delving too deep into technical details, and frequently summarizing the latest paragraphs. The technical jargon is clearly defined, (which adds to the usefulness of the manuscript as a reference,) and the notation is consistent. I praise the Author for discussing how the main ideas originated in the literature; where his exposition differs, for pedagogical reasons, from the historical development; and how more recent literature reinterpreted previous results in a different light. Finally, I really appreciate the paragraphs where the Author highlights the crucial logical point underlying the development of the theory.
I have just a couple of observations, which the Author might take into consideration, mostly stemming from my personal background in condensed matter physics.
First, in the last four years, the hydrodynamic flow of electrons in ultra-clean layered conductors has been theoretically investigated and experimentally demonstrated, leading to several high-impact publications. Mention of these results might find a place in the introductory part of the manuscript.
Second, I find that more space could be devoted to the motivation for the development of the theory, both in terms of an actual physical system to model, and in terms of results that showcase how the theory is (or could be) superior to other approaches. In other words, in the spirit of making the manuscript more interesting to a reader who is not already focused on integrable systems, the Authors could try to address the big "So what?" question: What changes in a theoretician's capabilities, if he/she learns GHD? What new systems can be modeled, that he/she could not model before? How do the results for the physical observables improve? For example, if available, I would like to see how GHD models the quantum Newton's cradle experiment (how many equations do we have to solve?, how do they look?), or how it improves upon the Gross-Pitaevskii equation in any cold-atoms setup. I think that, if the Author would go the extra mile in showcasing some results of the theory applied to a physical, experimentally realizable setup, this already excellent manuscript could be palatable to an even broader readership.
Overall, I find the manuscript to be remarkably well written. The exposition benefits from a steady pace, never delving too deep into technical details, and frequently summarizing the latest paragraphs. The technical jargon is clearly defined, (which adds to the usefulness of the manuscript as a reference,) and the notation is consistent. I praise the Author for discussing how the main ideas originated in the literature; where his exposition differs, for pedagogical reasons, from the historical development; and how more recent literature reinterpreted previous results in a different light. Finally, I really appreciate the paragraphs where the Author highlights the crucial logical point underlying the development of the theory.
I have just a couple of observations, which the Author might take into consideration, mostly stemming from my personal background in condensed matter physics.
First, in the last four years, the hydrodynamic flow of electrons in ultra-clean layered conductors has been theoretically investigated and experimentally demonstrated, leading to several high-impact publications. Mention of these results might find a place in the introductory part of the manuscript.
Second, I find that more space could be devoted to the motivation for the development of the theory, both in terms of an actual physical system to model, and in terms of results that showcase how the theory is (or could be) superior to other approaches. In other words, in the spirit of making the manuscript more interesting to a reader who is not already focused on integrable systems, the Authors could try to address the big "So what?" question: What changes in a theoretician's capabilities, if he/she learns GHD? What new systems can be modeled, that he/she could not model before? How do the results for the physical observables improve? For example, if available, I would like to see how GHD models the quantum Newton's cradle experiment (how many equations do we have to solve?, how do they look?), or how it improves upon the Gross-Pitaevskii equation in any cold-atoms setup. I think that, if the Author would go the extra mile in showcasing some results of the theory applied to a physical, experimentally realizable setup, this already excellent manuscript could be palatable to an even broader readership.

---

## Round 2 · Author Response

I would like to thank the referee for their report and the helpful recommendations made. I have made all changes suggested.
First point, mention of hydrodynamics flow of electrons: indeed this was missing. I have added references and brief mention in the introduction (top of page 6). I had already ref 18 (formerly [13]) about electrons in graphene, but I added more refs and a mention in the text.
Second point, showcasing the theory: this is indeed a very good point. I opted to add a subsection in the introduction of 3-4 pages, "Invitation: the hydrodynamics of a quantum Newton’s cradle experiment", where I briefly describe a quantum Newton's cradle type of experiment. I have included a discussion of the experiment itself (without, of course, going into any experimental detail, of which I am not an expert and which is beyond the scope of the notes), the microscopic description in terms of Hamiltonians and observables we would want to evaluate, the technical difficulties in naively trying to solve this problem either by the direct methods of quantum mechanics or by using integrability, the physics - that of standard hydrodynamics - one might naively expect, the principles behind what is actually happening with the simplified picture of the Newton's cradle, and finally how generalised hydrodynamics solves the problem, with all necessary equations for the full initial value problem. I believe this gives a good intuition as to why GHD is important, and what it consists of at the most basic level. I think going over how it improves upon the Gross-Pitaevskii equation in any cold-atoms setup would go beyond the scope of these notes, as I would like not to assume the reader is familiar with the particular physics of cold atomic gases; instead I have referred to a book about cold atom methods, and papers where their relation to GHD is discussed, at the beginning of this new section. I think there is still research to do in the direction of comparing GHD to other methods.
First point, mention of hydrodynamics flow of electrons: indeed this was missing. I have added references and brief mention in the introduction (top of page 6). I had already ref 18 (formerly [13]) about electrons in graphene, but I added more refs and a mention in the text.
Second point, showcasing the theory: this is indeed a very good point. I opted to add a subsection in the introduction of 3-4 pages, "Invitation: the hydrodynamics of a quantum Newton’s cradle experiment", where I briefly describe a quantum Newton's cradle type of experiment. I have included a discussion of the experiment itself (without, of course, going into any experimental detail, of which I am not an expert and which is beyond the scope of the notes), the microscopic description in terms of Hamiltonians and observables we would want to evaluate, the technical difficulties in naively trying to solve this problem either by the direct methods of quantum mechanics or by using integrability, the physics - that of standard hydrodynamics - one might naively expect, the principles behind what is actually happening with the simplified picture of the Newton's cradle, and finally how generalised hydrodynamics solves the problem, with all necessary equations for the full initial value problem. I believe this gives a good intuition as to why GHD is important, and what it consists of at the most basic level. I think going over how it improves upon the Gross-Pitaevskii equation in any cold-atoms setup would go beyond the scope of these notes, as I would like not to assume the reader is familiar with the particular physics of cold atomic gases; instead I have referred to a book about cold atom methods, and papers where their relation to GHD is discussed, at the beginning of this new section. I think there is still research to do in the direction of comparing GHD to other methods.

---

## Round 2 · List of Changes

references added according to the referee's comments and to other comments received by colleagues
typos corrected and some clarifications here and there
subsection 1.1 added, as described above.
typos corrected and some clarifications here and there
subsection 1.1 added, as described above.

---

## Editorial Decision

published